# Private Zeroth-Order Optimization with Public Data

**Xuchen Gong**  **Tian Li**
University of Chicago
{xuchengo,litian}@uchicago.edu

## Abstract

One of the major bottlenecks for deploying popular first-order differentially private (DP) machine learning algorithms (e.g., DP-SGD) lies in their high computation and memory cost, despite the existence of optimized implementations. Zeroth-order methods have promise in mitigating the overhead, as they leverage function evaluations to approximate the gradients, hence significantly easier to privatize. While recent works have explored zeroth-order approaches in both private and non-private settings, they still suffer from relatively low utilities compared with DP-SGD, and have only been evaluated in limited application domains. In this work, we propose to leverage public information to guide and improve gradient approximation of private zeroth-order algorithms. We explore a suite of public-data-assisted zeroth-order optimizers (PAZO) with minimal overhead. We provide theoretical analyses of the PAZO framework under an assumption of the similarity between public and private data. Empirically, we demonstrate that PAZO achieves superior privacy/utility tradeoffs across vision and text tasks in both pre-training and fine-tuning settings, outperforming the best first-order baselines (with public data) especially in highly private regimes, while offering up to $16\times$ runtime speedup.

## 1 Introduction

Differentially private (DP) is a widely-used framework to protect sensitive information so that adversaries cannot infer if any user or sample participates in the computation. When applied to machine learning tasks, popular DP algorithms based on privatizing first-order gradients (such as DP-SGD [1]) fundamentally rely on per-sample gradient clipping, which can be computationally expensive and impractical in large-scale settings. While there exist optimized implementations of DP-SGD, they are limited in their generality to handle all model architectures and often incur other overheads, such as trading extra memory for computation [2, 3].

To tackle this, zeroth-order optimization offers an attractive alternative for DP training, as it leverages function queries (scalar values) to approximate the gradients and is hence inherently amenable to privatization [4, 5]. However, randomly searching in a potentially high-dimensional space based on function query feedback can be rather inefficient [4]. Prior work has demonstrated competitive performance of (private) zeroth-order methods only in the limited context of language model fine-tuning with prompts [6, 7, 8, 9, 10] or models with extreme sparsity [11]. In addition, there is still a utility gap between private zeroth-order and first-order approaches on challenging tasks [8].

In this work, we aim to narrow the gap between zeroth-order and first-order methods in private training leveraging public data. Zeroth-order outputs are high-variance estimators of the first-order gradients and suffer from slow convergence in terms of the total number of iterations. However, there usually exists non-sensitive public data, whose batch gradients provide informative guidance on perturbing the parameter space. We thus introduce PAZO, a suite of zeroth-order DP algorithms that leverage a small amount of public data with similar distributions as private data along with their first-order gradients to guide or augment the zeroth-order outputs. In particular, we explore (1) PAZO-M, a mix (convex combination) of private zeroth-order estimates and public first-order gradients, (2) PAZO-P,

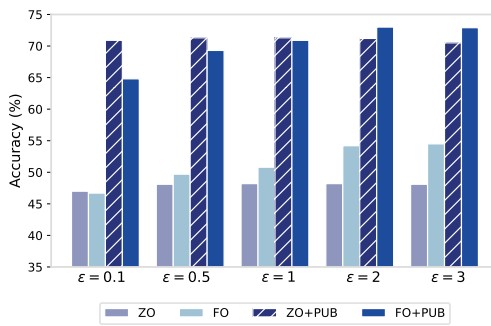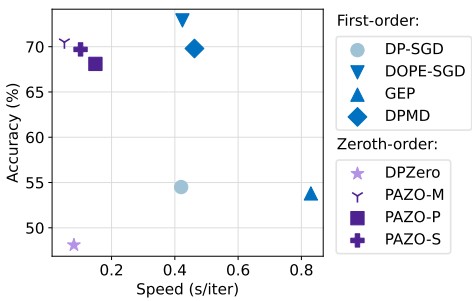

Figure 1: Results of CIFAR-10 with NFResNet18 trained from scratch under privacy budget $\varepsilon = 3$. *Left:* Zeroth-order methods demonstrate consistent accuracies under various privacy budgets compared with the best first-order method with public data. *Right:* Proposed zeroth-order approaches (PAZO-*) are more accurate than vanilla DPZero, and significantly more efficient than all the public data augmented first-order baselines.

constraining the sampling of random directions in the public gradient subspace, and (3) PAZO-S, selecting the best public gradient based on function queries on private data. When designing PAZO, we ensure that privatization only operates on top of function evaluations to preserve the efficiency of zeroth-order approaches, while still satisfying desired privacy guarantees.

Unlike recent zeroth-order work that mostly focuses on language model tuning with prompts, we investigate both image and text domains, and both pre-training and fine-tuning scenarios. We show that without access to public data, DP zeroth-order methods may underperform DP first-order approaches (e.g., DP-SGD [1]), whereas even modest amounts of public data can significantly close the gap, especially in highly private regimes. In particular, the best zeroth-order method with public data can match or even outperform the best public-data-assisted first-order counterpart, while being significantly faster to train. Our results highlight the broader potential of zeroth-order methods for DP training with public data: enabling improved privacy/utility tradeoffs, applicability across diverse domains, and achieving up to $16\times$ speedup compared to traditional first-order methods. Our contributions are summarized as follows:

1. **Algorithm design.** We propose the first set of private zeroth-order optimization algorithms (PAZO-{M,P,S}) augmented with public data (gradients) to construct better gradient estimates in a more constrained space. PAZO helps close the gap between zeroth- and first-order methods in the settings where zeroth-order approaches underperform first-order ones.

2. **Theoretical analysis.** We present the privacy and utility guarantees for each method, all with improved convergence rate in terms of model dimension $d$. PAZO-M improves the vanilla zeroth-order method by a factor of $\log d$, and PAZO-{P,S} obtain $d$-independent rates.

3. **Empirical validation.** We evaluate our methods on both image and text domains and in both pre-training and fine-tuning scenarios. We find that zeroth-order methods are robust across various privacy budgets whereas first-order methods are sensitive. Our methods consistently have superior privacy/utility tradeoffs and outperform the best public-augmented first-order method in highly privacy regimes, while achieving up to $16\times$ speedup.

## 2 Related Work and Preliminaries

**Differential privacy.** In this work, we focus on the popular definition of sample-level DP [1, 12].

**Definition 2.1** (Differential privacy [12]). *A randomized algorithm $\mathcal{M}$ is $(\varepsilon, \delta)$-differentially private if for all neighboring datasets $D, D'$ differing by one element, and every possible subset of outputs $O$,*

$$\Pr(\mathcal{M}(D) \in O) \le e^{\varepsilon} \Pr(\mathcal{M}(D') \in O) + \delta.$$

We follow the classic DP model where the neighboring datasets $D$ and $D'$ differ by adding/removing one training sample. Typically, noise is added to ensure DP scales with the model dimensions, resulting in degraded and unusable model utilities [13]. Extensive prior research has been proposed

to improve privacy/utility tradeoffs, including increasing the batch size [14, 15], using public or side information [16, 17, 18], and reducing the dimensionality of gradients [19]. Another bottleneck of deploying DP algorithms at scale lies in the computation (or memory) cost [2]. For example, vanilla DP-SGD computes and stores per-sample clipped gradients, leading to memory consumption $O(bd)$ where $b$ is the private batch size and $d$ is the model dimension. Existing methods, such as ghost-clipping/bookkeeping [20], reduce layer-wise gradient storage to $\min\{2bp, bd\}$, where $d$ is the layer dimension and $p$ is the feature dimension of this layer, i.e., sequence length for text data. In this work, we propose to mix zeroth-order (on sensitive private data) and first-order oracles (on public data) to mitigate these two challenges at once.

**Zeroth-order optimization.** Zeroth-order approaches use (stochastic) function queries to estimate the true gradients. They are particularly suitable for applications where gradient information is difficult to obtain, such as adversarial attacks and defenses [21, 22, 23], hyperparameter tuning [24], and data-driven science workloads [25]. One fundamental challenge of zeroth-order methods is the need for a large number of function queries to reduce the variance of the estimate [4]. Existing work has explored various techniques to improve the estimate, such as incorporating the previous estimated gradient directions [26] and sparsifying gradients [11]. Our work focuses on private training, and the proposed techniques can be combined with those prior methods. Given the current model parameter $x \in \mathbb{R}^d$ and loss function $f : \mathbb{R}^d \to \mathbb{R}$, the widely used two-point zeroth-order gradient estimator [4], involves two evaluations of function values:

$$g_\lambda(x; \xi) := \frac{f(x + \lambda u; \xi) - f(x - \lambda u; \xi)}{2\lambda} u, \tag{1}$$

where $\xi$ is a randomly sampled training data point, $u \in \mathbb{R}^d$ is uniformly sampled from the Euclidean sphere $\sqrt{d}\mathbb{S}^{d-1}$, and $\lambda > 0$ is the smoothing parameter. Let $v$ be uniformly sampled from the Euclidean ball $\sqrt{d}\mathbb{B}^d = \{x \in \mathbb{R}^d | \|x\| \le \sqrt{d}\}$. Define the smoothed version of $f(\cdot)$ as $f_\lambda(x) := \mathbb{E}_v[f(x + \lambda v)]$. We have that (1) $f_\lambda(x)$ is differentiable and (2) $\mathbb{E}_u[g_\lambda(x; \xi_i)] = \nabla f_\lambda(x)$ [4, 8]. It indicates that by using the zeroth-order gradient estimator, we are asymptotically optimizing a smoothed version of the original objective $f(x)$, where the smoother is a ball with radius $\lambda\sqrt{d}$.

**Differentially private zeroth-order optimization.** The desired private gradients are expensive to obtain in DP training, because gradients have to be generated and privatized at a granularity of samples as opposed to mini-batches. Therefore, recent work has considered privatizing zeroth-order algorithms [8, 7, 27, 28] by first clipping the function queries and then adding proper Gaussian noise. Specifically, based on the non-private two-point estimator on one sample (Eq. (1)), the private zeroth-order gradient $\tilde{g}_\lambda(x; B)$ is computed by

$$\tilde{g}_\lambda(x; B) := \left( \frac{1}{b} \sum_{\xi \in B} \text{clip}_C \left( \frac{f(x + \lambda u; \xi) - f(x - \lambda u; \xi)}{2\lambda} \right) + z \right) u, \tag{2}$$

where $b = |B|$ is batch size, $z \sim \frac{1}{b}\mathcal{N}(0, C^2\sigma^2)$ is privacy noise, and $u$ is a random direction, e.g., sampled uniformly from a sphere $\sqrt{d}\mathbb{S}^{d-1}$. We can query the raw data multiple times per iteration by sampling multiple $u$'s to improve the estimate (Section 3). Prior private zeroth-order work mostly focuses on language model tuning with prompts, and additionally there still exists a big performance gap between zeroth- and first-order methods [8, 7, 27]. In PAZO, we use public information to guide the gradient estimate on private data, as discussed in the next section.

## 3 PAZO: Public-Data-Assisted Private Zeroth-Order Optimization

Given zeroth-order oracles on private data and first-order oracles on public data, we aim to blend public gradients into the private zeroth-order framework to improve privacy/utility tradeoffs, while retaining the efficiency benefits of vanilla zeroth-order updates. In this section, we propose three approaches using this public prior that significantly outperform zeroth-order baselines without public data and result in competitive/superior performance relative to DP-SGD with public data. We analyze their convergence properties in Section 4.

## 3.1 PAZO-M: Mixing Zeroth-Order Estimates and First-Order Gradients

PAZO-M linearly combines the public gradient with the private two-point estimator (Eq. (2)). At each iteration $t$, we sample a public batch, obtain its batch gradient, and mix it with the private two-point gradient estimate. We run private two-point estimation $q$ times to reduce its variance. Since we query the same raw private mini-batch $q$ times, we need to add more privacy noise ($q$ times more variance) to ensure the same DP as if querying once. The updating rule is summarized in Algorithm 1 below.

We note that the norm of two-point gradient estimates is approximately $d$ times that of the true private gradient [6], so it is important to align their norms so that tuning the mixing coefficient can be easier. To achieve this, we sample $u$ uniformly from the sphere $r\mathbb{S}^{d-1}$ with radius $r = d^{\frac{1}{4}}$ so that $\mathbb{E}_{u_t}[\|g_\lambda(x)\|^2] \approx \|\nabla f(x)\|^2$. The proof is detailed in Appendix A. The mixing coefficient $\alpha$ can be adjusted to change the emphasis on the public gradient. Although $\alpha$ is an introduced hyperparameter, as shown in experiments (Section 5), PAZO-M is robust to a wide range of $\alpha$ values in $(0, 1)$ as well as the public batch size $b'$, as long as the $L_2$ norms of $g_{\text{pub}}$ and $\tilde{g}/q$ are aligned.

Despite its simplicity, PAZO-M demonstrates competitive performance among all three PAZO variants (Section 5). While prior work has explored mixing gradients and zeroth-order estimates for memory efficiency in non-private settings [29], PAZO-M differs from this work in terms of the effective optimization objectives, bias-variance tradeoffs, analyses, and application settings.

---

**Algorithm 1** PAZO-M

---

1: **Input:** $T$, noise multiplier $\sigma$, clipping threshold $C$, stepsize $\eta$, smoothing parameter $\lambda$, mixing coefficient $\alpha$, initialization $x_0 \in \mathbb{R}^d$, number of queries $q$, private and public batch sizes $b$ and $b'$
2: **for** $t = 0, \cdots, T - 1$ **do**
3:     Sample a mini-batch $B$ ($|B| = b$) of private training data $\{\xi_1, ..., \xi_b\}$
4:     Sample a mini-batch $B'$ ($|B'| = b'$) of public data and obtain its gradient $g_{\text{pub}}$
5:     $\tilde{g} \leftarrow 0^d$
6:     **for** each of the $q$ queries **do**
7:         Sample $u$ uniformly from the sphere $d^{\frac{1}{4}}\mathbb{S}^{d-1}$
8:         $\tilde{g} \leftarrow \tilde{g} + \left( \frac{1}{b} \sum_{i=1}^{b} \text{clip}_C \left( \frac{f(x_t + \lambda u; \xi_i) - f(x_t - \lambda u; \xi_i)}{2\lambda} \right) + z \right) u$, where $z \sim \frac{1}{b}\mathcal{N}(0, qC^2\sigma^2)$
9:     **end for**
10:    $x_{t+1} \leftarrow x_t - \eta(\alpha g_{\text{pub}} + (1 - \alpha)\tilde{g}/q)$
11: **end for**

---

## 3.2 PAZO-P: Sampling in Public Gradient Subspace

Recall that the two-point estimator samples perturbations $u$ in the sphere $\sqrt{d}\mathbb{S}^{d-1}$. Such random exploration along two directions $\lambda u$ and $-\lambda u$ can result in a loose estimation of the real gradients in high-dimensional settings. In this section, we assume the true gradient on private data is close to the space formed by public gradients. Based on this assumption, we constrain the private gradient estimates to lie in the subspace spanned by the public gradients, and *use function queries to learn the coefficients* associated with the components of the public gradient subspace (named PAZO-P). This gives us a much lower-dimensional optimization problem.

Formally, suppose we have access to $k$ ($k \ll d$) mini-batch stochastic gradients obtained on public data. Denote a concatenation of them as a matrix $G \in \mathbb{R}^{d \times k}$. Let $u \in \mathbb{R}^k$ be a random vector that is uniformly sampled from the sphere $\sqrt{k}\mathbb{S}^{k-1}$. We propose the following update rule (sampling only one $u$ as an example) in the non-private case:

$$g_\lambda^G(x; \xi) := \frac{f(x + \lambda Gu; \xi) - f(x - \lambda Gu; \xi)}{2\lambda} Gu,$$

which can be interpreted as learning the coefficient $u \in \mathbb{R}^k$ to linearly combine the public gradients. Further, if we orthonormalize the columns of $G$, $g_\lambda^G(x; \xi)$ estimates the orthogonal projection of the true gradient onto the public gradient subspace when $\lambda \to 0$, i.e.,

$$\mathbb{E}_u[g_\lambda^G(x; \xi)] = \mathbb{E}_u[\nabla f(x)^\top GuGu] = \text{Proj}_G(\nabla f(x)).$$

We compare the visualization of sampling in the full-dimensional space and public gradient subspace in Figure 7. For private training, we privatize each estimate (in the public gradient subspace) using the standard subsampled Gaussian mechanism, described in Algorithm 2.

---

**Algorithm 2** PAZO-P

1: **Input:** Same as Algorithm 1, and number of public batches $k \ll d$
2: **for** $t = 0, \cdots, T-1$ **do**
3:     Sample a mini-batch $B(|B| = b)$ of private training data $\{\xi_1, ..., \xi_b\}$
4:     Sample $k$ batches of public data and obtain their (ortho)normalized gradients $\{g_1, ..., g_k\}$
5:     $G \leftarrow [g_1, \ldots, g_k], \; \tilde{g} \leftarrow 0^d$
6:     **for** each of the $q$ queries **do**
7:         Sample $u$ uniformly from the sphere $\sqrt{k}\mathbb{S}^{k-1}$
8:         $\tilde{g} \leftarrow \tilde{g} + \left(\frac{1}{b}\sum_{i=1}^{b}\text{clip}_C\left(\frac{f(x_t + \lambda G u; \xi_i) - f(x_t - \lambda G u; \xi_i)}{2\lambda}\right) + z\right)Gu$, where $z \sim \frac{1}{b}\mathcal{N}(0, qC^2\sigma^2)$
9:     **end for**
10:    $x_{t+1} \leftarrow x_t - \eta\tilde{g}/q$
11: **end for**

---

PAZO-P is conceptually related to the idea of model soup, where extensive research has shown that a simple convex combination of the model parameters can result in a souped model that generalizes well even in out-of-distribution tasks [30, 31].

Previous work proposes constraining the random search to the principal components of surrogate gradients [32]. PAZO-P differs from theirs in allowing to use non-orthonormalized $G$. Section 5 presents the performance of PAZO-P with orthonormalization, and the complete results in Tables 2-5 demonstrate the competitive performance of PAZO-P without orthonormalization.

### 3.3 PAZO-S: Select the Best Public Gradient

PAZO-P offers ways to better combine public gradients via zeroth-order function evaluations, while in this section, we take an alternative approach by optimizing an approximation of the problem. Note that for a convex function $f$, for any probability distribution $\alpha \in \Delta_k$, $k$ public gradients $\{g_1, \ldots, g_k\}$, and model parameter $x \in \mathbb{R}^d$, we have that

$$\min_{\alpha \in \Delta_k} f\left(x - \eta \sum_{j=1}^{k} \alpha_j g_j\right) \leq \min_{\alpha \in \Delta_k} \sum_{j=1}^{k} \alpha_j f(x - \eta g_j) = \min_{j \in [k]} f(x - \eta g_j), \tag{3}$$

where the upper bound $\min_{j \in [k]} f(x - \eta g_j)$ can be easily optimized and privatized (as long as $k$ is small) with access to queries of $f(\cdot)$ evaluated on private data. Inspired by this observation, we propose PAZO-S, a method that selects the best public gradients based on loss values on private data, i.e., solving $\min_{j \in [k]} f(x - \eta g_j)$ (Line 5-8 in Algorithm 3). Considering the residual error between

---

**Algorithm 3** PAZO-S

1: **Input:** Same as Algorithm 2, and perturbation scale $\epsilon$
2: **for** $t = 0, \cdots, T-1$ **do**
3:     Sample a mini-batch $B(|B| = b)$ of private training data $\{\xi_1, ..., \xi_b\}$
4:     Sample $k$ mini-batches of public data and obtain their gradients $\{g_1, ..., g_k\}$
5:     **for** $j = 1, ..., k$ **do**
6:         $f_j \leftarrow \frac{1}{b}\sum_{i=1}^{b} \text{clip}_C\left(f(x_t - \eta g_j; \xi_i)\right) + z$ where $z \sim \frac{1}{b}\mathcal{N}(0, (k+1)C^2\sigma^2)$
7:     **end for**
8:     $\hat{j} \leftarrow \arg\min_{j \in [k]} f_j$
9:     $g_{k+1} \leftarrow g_{\hat{j}} + z'$ where $z' \sim \mathcal{N}(0, \epsilon^2 I_d)$
10:    $f_{k+1} \leftarrow \frac{1}{b}\sum_{i=1}^{b} \text{clip}_C\left(f(x_t - \eta g_{k+1}; \xi_i)\right) + z$ where $z \sim \frac{1}{b}\mathcal{N}(0, (k+1)C^2\sigma^2)$
11:    $j^* \leftarrow \arg\min_{j \in [k+1]} f_j$
12:    $x_{t+1} \leftarrow x_t - \eta g_{j^*}$
13: **end for**

---

the public and private subspace, we create an additional noise vector $z'$ (Line 9), add it to the best public gradient (indexed with $\hat{j}$), and perform another comparison between private $f(x - \eta g_{\hat{j}})$ and private $f(x - \eta(g_{\hat{j}} + z'))$ (Line 11). While PAZO-S is motivated by the arguments under a convex $f$ (Eq. (3)), we apply it to all the tasks and models that are non-convex.

### 3.4 Privacy Guarantees of PAZO

The privacy guarantees of all three methods can be analyzed in the same way. At each iteration, we guarantee the $L_2$ sensitivity of the sum of the function queries by $C$, and we add Gaussian noise with variance $qC^2\sigma^2$ where $q$ is the number of queries on the sampled data. Therefore, the privacy bound per iteration is the same for any $q$, following the $n$-fold composition corollary of the Gaussian mechanism [33]. Applying standard moments accountant method [1] to compose across $T$ rounds with sampling ratio $b/n$, we have that there exist constants $c_1$ and $c_2$ such that for any $\varepsilon < c_1 b^2 T/n^2$, all three Algorithms 1-3 are $(\varepsilon, \delta)$-differentially private for any $\delta > 0$ if $\sigma \geq c_2 \frac{b\sqrt{T\log(1/\delta)}}{n\varepsilon}$.

## 4 Convergence Analysis

In this section, we study the convergence properties of three PAZO algorithms. We first define the similarity between public and private data through the distance between the full gradients as follows.

**Definition 4.1** ($\gamma$-similarity). *Denote $\nabla f'(x_t)$ and $\nabla f(x_t)$ as the gradient for model $x_t$ at time step $t$ under the full public and private data, respectively. We call public and private data $\gamma$-similar if $\|\nabla f'(x_t) - \nabla f(x_t)\| \leq \gamma$ for all $t$.*

We note that such similarity is defined on top of the full gradients, a weaker requirement than defining on the stochastic gradients. There are previous similarity metrics based on coordinate-wise gradient norm alignment [16]. Together with their assumption on the bounded gradient norm, their similarity condition implies ours and is thus a stronger assumption. Next, we present additional assumptions.

**Assumption 1.** *$f(x; \xi)$ is $M$-Lipschitz for any $x \in \mathbb{R}^d$ and any subset data $\xi$.*

**Assumption 2.** *$f(x; \xi)$ is $L$-smooth for any $x \in \mathbb{R}^d$ and any subset data $\xi$.*

**Assumption 3.** *The variance of private stochastic gradients is bounded, i.e., $\mathbb{E}[\|\nabla f(x_t; \xi_i) - \nabla f(x_t)\|^2] \leq \sigma_1^2$ for any private sample $\xi_i$ and any $t$.*

**Assumption 4.** *The variance of public stochastic gradients is bounded, i.e., $\mathbb{E}[\|\nabla f'(x_t; \xi_i') - \nabla f'(x_t)\|^2] \leq \sigma_2^2$ for any public sample $\xi_i'$ and any $t$.*

**Theorem 4.1** (Convergence of PAZO-M). *Assume public and private data are $\gamma$-similar. Let Assumptions 1-4 hold. For possibly non-convex $f(\cdot)$, running Algorithm 1 under a fixed learning rate for $T$ rounds gives*

$$\frac{1}{T}\sum_{t=0}^{T-1} \mathbb{E}[\|\nabla f(x_t)\|^2] \leq O\left(\frac{1}{T}\right) + O\left(\gamma^2 + \frac{\sigma_1^2}{b} + \frac{\sigma_2^2}{b'} + \frac{\sigma^2}{b^2}\right). \tag{4}$$

*Additionally, let $c_1$ and $c_2$ be the constants that make PAZO-M satisfy $(\varepsilon, \delta)$-differential privacy for any $\varepsilon < c_1 b^2 T/n^2, \delta > 0$. Then PAZO-M obtains the error rate*

$$O\left(\frac{1-\alpha}{\alpha}\sqrt{d}\right) + O\left(\gamma^2\frac{\alpha\sqrt{d}}{2(1-\alpha)+\alpha\sqrt{d}} + \frac{\sigma_1^2}{b}\frac{(1-\alpha)\sqrt{d}}{(1-\alpha)\sqrt{d}+\alpha} + \frac{\sigma_2^2}{b'}\frac{\alpha^2\sqrt{d}}{(1-\alpha)^2\sqrt{d}+\alpha(1-\alpha)}\right)$$

*by choosing the parameters*

$$\eta = \frac{2(1-\alpha)+\alpha\sqrt{d}}{4L((1-\alpha)^2\sqrt{d}+\alpha(1-\alpha))}, \quad \lambda \leq \frac{1}{Ld^{\frac{5}{4}}}, \quad C = 1 + \sqrt{2}d^{\frac{1}{4}}M, \quad and$$

$$T = \frac{4n\varepsilon[(1-\alpha)\sqrt{d}+\alpha]}{c_2 C[2(1-\alpha)+\alpha\sqrt{d}]}\sqrt{\frac{2L[f(x_0)-f(x_*)]}{\sqrt{d}\log(1/\delta)}}.$$

We present several discussions on the results. First, we see that the first term in the error rate has dependence $O(\frac{1-\alpha}{\alpha}\sqrt{d})$, which saves a factor of $\log d$ compared to DPZero, together with a constant improvement if $\alpha > \frac{1}{2}$. Due to the usage of biased public gradients, we additionally have an error $O(\gamma^2 \alpha \sqrt{d} + \sigma_2^2 \alpha^2 \sqrt{d}/b')$, which decreases to 0 as $\alpha$ decreases to 0. Second, there is a term related to the variance of the stochastic gradients $\sigma_1^2/b$, which is standard when we assume constant learning rates [34] and would reduce as the batch size $b$ increases. Third, we provide a conservative upper bound by choosing the clipping threshold $C$ larger than needed. We can also naturally extend our current analysis to incorporate more advanced clipping analysis [8].

**Theorem 4.2** (Convergence of PAZO-P). *Let assumptions in Theorem 4.1 hold. For possibly non-convex $f(\cdot)$, running Algorithm 2 under a fixed learning rate for $T$ rounds gives*

$$\frac{1}{T}\sum_{t=0}^{T-1}\mathbb{E}[\|\nabla f(x_t)\|^2] \leq O\left(\frac{1}{T}\right) + O\left(\sqrt{\gamma^2 + \frac{\sigma_2^2}{b'}} + \frac{\sigma_1^2}{b} + \frac{\sigma^2}{b^2}\right). \tag{5}$$

*Additionally, let $c_1$ and $c_2$ be the constants that make PAZO-P satisfy $(\varepsilon,\delta)$-differential privacy for any $\varepsilon < c_1 b^2 T/n^2, \delta > 0$. Then PAZO-P obtains the error rate*

$$O(k) + O\left(\sqrt{\gamma^2 + \frac{\sigma_2^2}{b'}} + \frac{\sigma_1^2}{b}\right)$$

*by choosing the parameters*

$$\eta = \frac{1}{2Lk}, \quad \lambda \leq \frac{1}{Lk^{\frac{3}{2}}}, \quad C = 1 + \sqrt{2k}M, \quad and \quad T = \frac{n\varepsilon}{c_2 C}\sqrt{\frac{8Lk[f(x_0) - f(x_*)]}{\log(1/\delta)}}.$$

This shows that we have $d$-independent error rate $O(k)$, with the dimension of the subspace $k$ being small a constant $k \ll \log d$ in practice. We additionally have the error term $O(\gamma^2 + \sigma_2^2/b')$ from the biased stochastic public gradients and $O(\sigma_1^2/b)$ from the stochastic private gradients.

**Theorem 4.3** (Convergence of PAZO-S). *Let assumptions in Theorem 4.1 hold. For possibly non-convex $f(\cdot)$, running Algorithm 3 under a fixed learning rate for $T$ rounds gives*

$$\frac{1}{T}\sum_{t=0}^{T-1}\mathbb{E}[\|\nabla f(x_t)\|^2] \leq O\left(\frac{1}{T}\right) + O\left(\gamma^2 + \frac{\sigma_2^2}{b'} + \epsilon^2\right). \tag{6}$$

This allows us to take $T \to \infty$, $\eta = 1/(4L)$, and $\epsilon \leq 1/\sqrt{d}$ to achieve a $d$-independent error bound $O(\gamma^2 + \sigma_2^2/b')$. When $\gamma \to 0$, the remaining term $\sigma_2^2/b'$ is due to stochastic public data sampling.

We give complete statements and proofs for Theorem 4.1, 4.2, and 4.3 in Appendix B.2, B.3 and B.4. Additionally, if we further assume the bounded loss that for some constant $S$, $|f(x_t)| \leq S$ for all $t$, we can use smaller noise in proofs under the same privacy budget. This yields improved error bounds that $O(\frac{1-\alpha}{\alpha}d^{\frac{1}{4}})$ for PAZO-M and $O(\sqrt{k}\log k)$ for PAZO-P, whose complete statements and proofs are given in Appendix B.5 and B.6. We summarize the baselines and our error bounds in Table 1.

## 5 Empirical Evaluation

In this section, we present the empirical performance of PAZO-{M,P,S} across both vision and language domains, and pre-training, fine-tuning, and prompt tuning tasks. In Section 5.1, we introduce experiment setups including datasets and models. In Section 5.2, we present the privacy/utility tradeoffs of PAZO, showing that PAZO performs comparably to public data augmented first-order methods over a number of tasks in moderate privacy regimes and outperforms them in highly private regimes. In Section 5.3, we highlight the time efficiency of PAZO. In Section 5.4, we present the sensitivity study of the hyperparameters, showing that PAZO is non-sensitive to introduced hyperparameters. Our code is publicly available at `github.com/xuchengong/pazo`.

### 5.1 Experimental Setups

The settings of our experiments cover and follow the experiments in the existing DP literature, including (1) Training NFResNet18 on CIFAR-10 [35] from scratch, (2) fine-tuning Places365

pre-trained ViT-S on Tiny-ImageNet [36], (3) training LSTM on IMDB [37] from scratch, and (4) fine-tuning RoBERTa-base with prompts on MNLI [38]. We introduce distribution shifts between private and public data, such as class imbalance and semantic context shifts of various extents. The details of public data generation and the impact of different public data distribution shifts on algorithm performance and $\gamma$-similarity values are presented in Appendix C.1.

## 5.2 Improved Privacy/Utility Tradeoffs

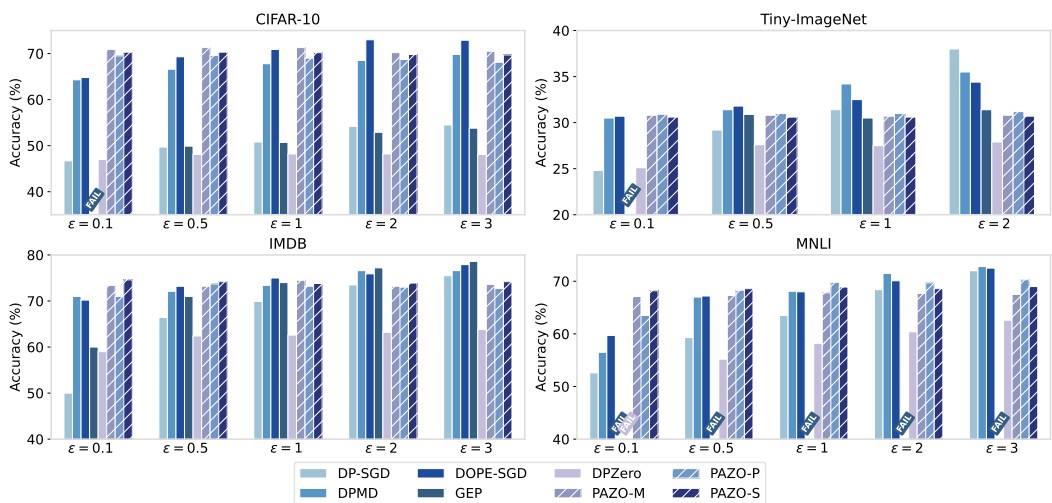

Figure 2: Performance of PAZO and the baselines in four settings. It shows that (1) all three PAZO variants outperform DPZero across all datasets, (2) all of the first-order methods (DP-SGD, DPMD, DOPE-SGD, and GEP), with or without public data, are more sensitive to smaller $\varepsilon$'s than zeroth-order ones, and (3) when $\varepsilon$'s are small, PAZO is superior to first-order baselines. "Fail" indicates failure to converge; the detailed accuracy numbers are in Tables $2-5$.

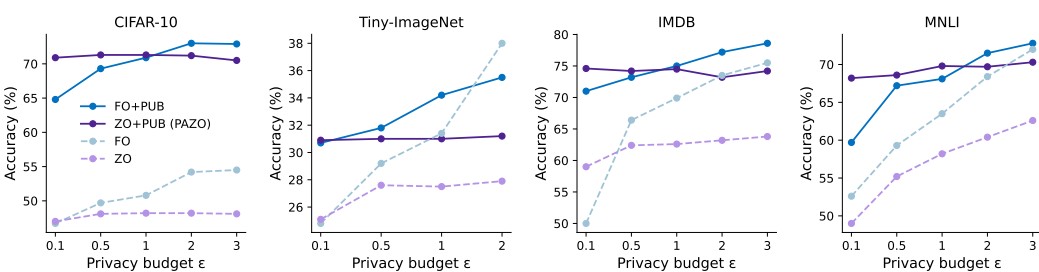

Figure 3: We compare the best private zeroth-order (ZO) methods with the best private first-order (FO) methods, with public data (+PUB) or without. Note that ZO+PUB is PAZO. It shows that (1) with or without public data, the performance gap between ZO and FO decreases as $\varepsilon$ decreases, (2) using public data expands the range of $\varepsilon$'s where ZO methods outperform FO ones, and (3) ZO+PUB (PAZO) achieves better privacy/utility tradeoff than FO+PUB when $\varepsilon$'s are small.

First, we compare PAZO with vanilla zeroth-order methods and various strong first-order baselines with public data under various privacy budgets $\varepsilon = \{0.1, 0.5, 1, 2, 3\}$. In Figure 2, we compare with (1) DP-SGD [1], the plain first-order method without public data, (2) DPZero [8], the plain zeroth-order method without public data, and (3) the state-of-the-art first-order algorithms with public data, including DPMD [39], GEP [40], and DOPE-SGD [41].

We observe that all three PAZO variants outperform DPZero across the four datasets, though there is not a single PAZO algorithm that dominates other PAZO instances in all settings. In addition, all of the first-order methods (DP-SGD, DPMD, DOPE-SGD, and GEP), with or without public data, are much more sensitive to more strict privacy requirements (smaller $\varepsilon$'s) than zeroth-order ones.

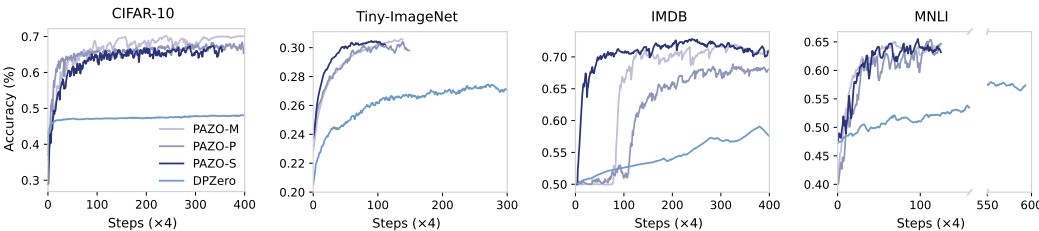

Figure 4: Convergence speed of private zeroth-order methods with (PAZO) or without (DPZero) public data. We observe that PAZO variants have slightly different convergence speed, but they are all consistently faster than the baseline. The reported are smoothed test accuracies under privacy $\varepsilon = 1$.

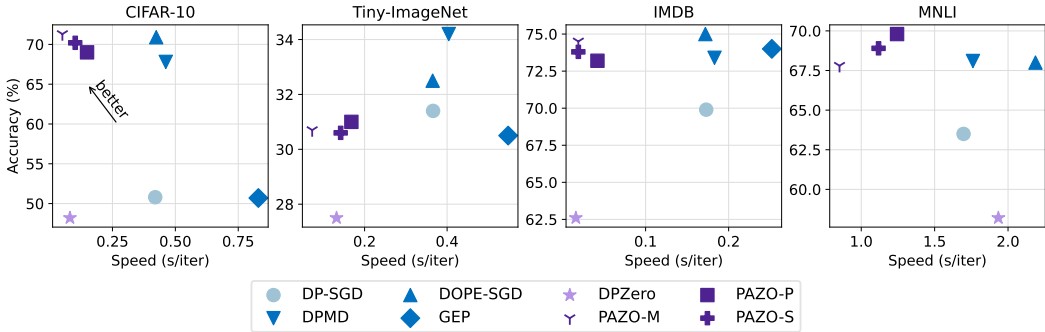

Figure 5: The utility/speed tradeoffs of different methods. It shows that PAZO is up to $16\times$ faster in each training iteration than FO and FO+PUB while being comparably performant. The reported results are under privacy budget $\varepsilon = 1$, and the detailed numbers are in Table 8.

This suggests that PAZO (and zeroth-order methods in general) possess more robust privacy/utility tradeoffs than the first-order methods across model types, training types, and task domains. Under small $\varepsilon$', PAZO is superior to first-order baselines by a large margin. We provide concrete accuracy numbers in Tables $2-5$ in the appendix.

Furthermore, we report performance of the best PAZO variant among three (denoted as 'ZO+PUB') and performance of the best public-data-augmented first-order method (denoted as 'FO+PUB') under different $\varepsilon$'s in Figure 3. It shows that although vanilla zeroth-order (ZO) may underperform first-order (FO) methods, if we augment both with public data, PAZO performs comparably or even superior to the best first-order approach with public data (FO+PUB), while being more memory-efficient.

## 5.3 Time Efficiency

In this section, we present the time efficiency of PAZO. It is faster than private first-order methods (with or without public data) as it does not require per-sample gradient clipping, and it also converges faster than private zeroth-order baselines.

#**Iterations to converge.** MeZO and DPZero present results with zeroth-order methods running $100\times$ and $10\times$ more steps than first-order ones [6, 8], but PAZO converges much faster due to assistance from public data. Figure 4 plots the convergence speed of DPZero and PAZO-{M, P, S}, illustrating that public information significantly accelerates the convergence of (private) zeroth-order methods. This property is particularly favorable to differentially private training as smaller accumulative noise would be added due to fewer iterations needed to converge.

**Runtime per iteration.** Theoretically, we compare the number of different operations in each method in Table 9. Since the number of forward and backward passes in first-order methods depends on the private batch size, first-order methods can be dramatically slow since large-batch training is favorable in DP [14, 42]. Empirically, we compare the speed of each method in terms of training time per iteration. Each experiment is conducted on one 48GB L40S GPU. For a fair comparison,

we adopt optimized implementations to speed up first-order DP algorithms, including vectorization, just-in-time compilation, and static graph optimization [2]. In practice, due to the memory burden of parallelization and compilation overhead, a hybrid of `vmap` and sequential processing is often faster. We choose the fastest implementation for each first- and zeroth-order method under memory constraints. By comparing the utility/speed tradeoff (Figure 5), we observe that PAZO is comparable to or more performant than the baselines, while being $2 \sim 16\times$ faster in each training iteration.

## 5.4 Robustness to Hyperparameters

We have each method's hyperparameters tuned via grid search, and the detailed grid values are in Appendix C.3. Zeroth-order methods sample $q$ random directions to reduce variance in each iteration, so we perform preliminary studies on $q \in \{1, 5\}$ for each setting and choose $q = 1$ if the performance gap is negligible. As shown in Table 11, DPZero benefits from increased $q$ for improved accuracy, while PAZO has reduced dependence on $q$'s due to the guidance from public information.

Furthermore, compared to vanilla zeroth-order methods, PAZO has additional hyperparameters due to public data sampling, including the public batch size $b'$, the mixing coefficient $\alpha$, number of public candidates $k$, and the perturbation scale $\epsilon$. However, as presented in Figure 6 and Figure 8, the performance of all PAZO variants is robust to the values of these hyperparameters. In fact, a wide range of combinations of these hyperparameter values can yield performance close to the best performance we report.

**CIFAR-10**

**PAZO-M**

| $b'$ | $\alpha$ 0.25 | 0.5 | 0.75 |
|---|---|---|---|
| 8 | 70.3 | 70.5 | 70.2 |
| 32 | 66.8 | 68.4 | 67.2 |

**PAZO-P**

| $b'$ | $k$ 3 | 6 | 10 |
|---|---|---|---|
| 8 | 67.9 | 68.2 | 67.7 |
| 16 | 67.8 | 68.1 | 68.1 |
| 32 | 68.6 | 68.0 | 67.7 |

**PAZO-S**

| $\epsilon$ | $b'$ 8 | 16 | 32 |
|---|---|---|---|
| 1e-2 | 68.2 | 67.2 | 66.6 |
| 1e-3 | 69.7 | 68.5 | 68.9 |
| 1e-4 | **69.8** | 67.8 | 68.1 |
| 0 | **69.5** | 68.2 | 69.0 |

**MNLI**

**PAZO-M**

| $b'$ | $\alpha$ 0.25 | 0.5 | 0.75 |
|---|---|---|---|
| 8 | 67.4 | 67.4 | 66.9 |
| 32 | 67.5 | 67.1 | 67.5 |

**PAZO-P**

| $b'$ | $k$ 3 | 6 | 10 |
|---|---|---|---|
| 8 | 69.8 | 70.3 | 70.9 |
| 16 | 69.2 | 68.8 | 69.7 |
| 32 | 69.7 | 70.3 | 68.9 |

**PAZO-S**

| $\epsilon$ | $b'$ 8 | 32 | 128 |
|---|---|---|---|
| 1e-2 | **69.0** | 67.5 | 68.8 |
| 1e-3 | 68.1 | 68.1 | 68.5 |
| 1e-4 | 66.6 | 67.3 | 68.1 |
| 0 | 66.9 | **67.3** | 66.5 |

Figure 6: PAZO is non-sensitive to their introduced hyperparameters. Each number represents the best accuracy after the standard hyperparameters for zeroth-order private optimization ($C$ and $\eta$) are tuned. Blue cells indicate PAZO-S performance w/o a noisy candidate.

## 6 Conclusion and Future Work

We propose PAZO, a suite of public-data-assisted zeroth-order optimization methods for differentially private training. By leveraging modest amounts of public data and their gradients to guide zeroth-order updates, PAZO significantly improves the privacy/utility tradeoff over prior zeroth-order approaches while preserving their computational efficiencies. Through theoretical analysis and experiments across vision and language tasks, we demonstrate that PAZO closes the gap between zeroth- and first-order methods in moderate privacy regimes and even surpasses the best first-order baselines with public data under high privacy constraints. Our results position public-data-assisted zeroth-order optimization as a practical and scalable alternative for private training, especially in settings where private first-order methods are costly or infeasible. Future work could include sharpening the current convergence bounds by considering other similarity metrics and exploring a broader set of public and private dataset pairs in practical DP training applications.

## Acknowledgement

We thank Kamalika Chaudhuri, Chuan Guo, Saeed Mahloujifar, and Manzil Zaheer for helpful discussions at early stages of this project. We acknowledge the NAIRR Pilot program and AWS for contributing cloud credits to support this research project.

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

# A  Algorithm Details

## A.1  PAZO-M Norm Alignment

To jusify sampling the perturbation $u$ from the sphere with radius $d^{\frac{1}{4}}$, we present the following analysis. For a random direction sampled uniformly from a sphere of radius $r$, the two-point estimator $g_\lambda(x)$ has the squared norm

$$\|g_\lambda(x)\|^2 = \left( \frac{f(x + \lambda u) - f(x - \lambda u)}{2\lambda} \right)^2 r^2.$$

The Taylor expansion of $f$ with $O(\lambda^2)$ terms ignored gives $f(x \pm \lambda u) \approx f(x) \pm \lambda \nabla f(x)^\top u$, hence

$$\|g_\lambda(x)\|^2 \approx (\nabla f(x)^\top u)^2 r^2.$$

Since $\mathbb{E}_u[uu^\top] = \frac{r^2}{d} I_d$,

$$\mathbb{E}_u[\|g_\lambda(x)\|^2] \approx r^2 \mathbb{E}_u[(\nabla f(x)^\top u)^2] = r^2 \nabla f(x)^\top \mathbb{E}_u[uu^\top] \nabla f(x) = \frac{r^4}{d} \|\nabla f(x)\|^2.$$

We thus have $\mathbb{E}_u[\|g_\lambda(x)\|^2] \approx \|\nabla f(x)\|^2$ if $r = d^{\frac{1}{4}}$.

## A.2  PAZO-P Perturbation Sampling

We visualize the sampled perturbation set of the vanilla zeroth-order methods and PAZO-P as follows. We set $d = 3, k = 2$ and generate $G \in \mathbb{R}^{3 \times 2}$ with normalized columns to represent the public gradients. The vanilla zeroth-order method samples the perturbations $u$ in the full-dimensional sphere ($\mathbb{R}^3$), while PAZO-P samples in the column space of $G$. When $G$ is orthonormal, we sample fairly in every direction in the public gradient subspace; when $G$ is not orthonormal, we have larger effective learning rates in the directions in which the public gradients agree.

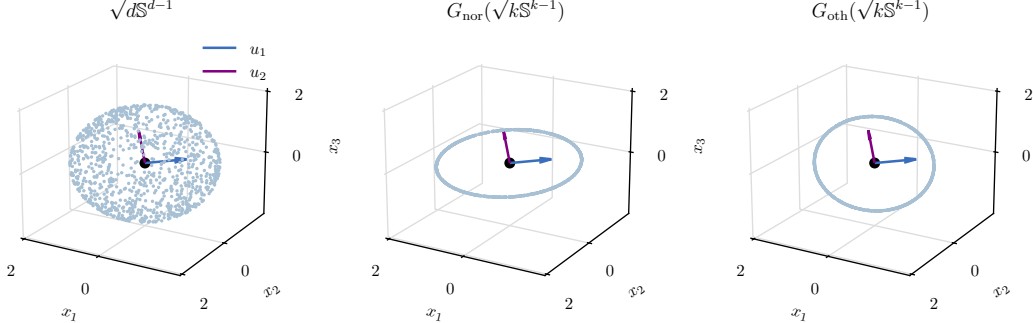

Figure 7: Comparison of the sampled perturbations in full-dimensional space and the public gradient subspace. $u_1$ and $u_2$ denote the top-2 left singular vectors of normalized $G$. *Left*: Vanilla zeroth-order perturbation sampling from $\sqrt{d}\mathbb{S}^{d-1}$. *Middle*: Sampling from $G(\sqrt{k}\mathbb{S}^{k-1})$ where $G$ has normalized columns, which is functionally the border of a sphere elongated in the directions of top public gradient singular vectors. *Right*: Sampling from $G(\sqrt{k}\mathbb{S}^{k-1})$ where $G$ is orthonormal.

# B  Detailed Convergence Analysis

In this section, we provide detailed proof of convergence for PAZO-{M,P,S} with and without the bounded loss assumption that $|f(x_t)| \leq S, \forall t$. We summarize the convergence of DP-SGD, DPZero, and our method in Table 1. Without the bounded loss assumption, we choose the clipping threshold $C$ larger than needed, which results in more pessimistic bounds. With the bounded loss assumption, we follow Zhang et al. [8] to upper-bound the probability of the event that clipping ever happens throughout the training, which results in better choices of clipping threshold and thus a tighter bound.

Table 1: Convergence error rate of DP-SGD, DPZero, and PAZO-{M,P,S} in terms of model dimension $d$, with and without the assumption that $|f(x_t)| \leq S, \forall t$. We denote $c$ as some constant independent of the model dimension $d$ and number of public gradients $k$.

| Method | Without $|f(x)| \leq S$ | With $|f(x)| \leq S$ |
|--------|------------------------|---------------------|
| DP-SGD | $O(\sqrt{d})$ | / |
| DPZero | / | $O(\sqrt{d}\log d)$ |
| PAZO-M | $O(\frac{1-\alpha}{\alpha}\sqrt{d})$ [§B.2] | $O(\frac{1-\alpha}{\alpha}d^{\frac{1}{4}})$ [§B.5] |
| PAZO-P | $O(k)$ [§B.3] | $O(\sqrt{k}\log k)$ [§B.6] |
| PAZO-S | $O(c)$ [§B.4] | |

## B.1 Lemmas

**Lemma B.1.** *Let the private and public data be $\gamma$-similar and Assumption 3 and 4 hold. Denote $b := |B|$ and $b' := |B'|$ as the private and public batch sizes, respectively. Denote $g_t := \nabla f(x_t)$ and $g'_t := \nabla f'(x_t)$ as the gradient under full private and public data, respectively. Due to the stochasticity of sampling, the private and public batch gradients are*

$$\nabla f(x_t; B_t) = \frac{1}{b}\sum_{i \in B_t}(g_t + \zeta_{t,i}) \quad and \quad \nabla f'(x_t; B'_t) = \frac{1}{b'}\sum_{i \in B'_t}(g'_t + \zeta'_{t,i})$$

*where $\zeta_{t,i}$ is independently sampled from some noise distribution $\mathcal{D}$ with zero mean and bounded variance $\sigma_1^2$; $\zeta'_{t,i}$ is independently sampled from some noise distribution $\mathcal{D}'$ with zero mean and bounded variance $\sigma_2^2$; $B_t$ and $B'_t$ are private and public batch at step $t$, respectively. So we have*

$$\mathbb{E}[\|\nabla f(x_t; B_t) - \nabla f'(x_t; B'_t)\|]^2 \leq \mathbb{E}[\|\nabla f(x_t; B_t) - \nabla f'(x_t; B'_t)\|^2]$$

$$= \mathbb{E}[\|g_t - g'_t\|^2] + \mathbb{E}\left[\left\|\frac{1}{b}\sum_{i \in B_t}\zeta_{t,i}\right\|^2\right] + \mathbb{E}\left[\left\|\frac{1}{b'}\sum_{i \in B'_t}\zeta'_{t,i}\right\|^2\right]$$

$$\leq \gamma^2 + \frac{\sigma_1^2}{b} + \frac{\sigma_2^2}{b'}$$

*where the first inequality is due to Jensen's inequality.*

**Lemma B.2** (Zhang et al. [8], Lemma C.1 and C.2). *Let $u$ be uniformly sampled from the Euclidean sphere $\sqrt{d}\mathbb{S}^{d-1}$ and $v$ be uniformly sampled from the Euclidean ball $\sqrt{d}\mathbb{B}^d = \{x \in \mathbb{R}^d \mid \|x\| \leq \sqrt{d}\}$. Let $a \in \mathbb{R}^d$ be some fixed vector independent of $u$. We have*

1. *$\mathbb{E}_u[u] = 0$ and $\mathbb{E}_u[uu^\top] = I_d$.*

2. *$\mathbb{E}_u[u^\top a] = 0$, $\mathbb{E}_u[(u^\top a)^2] = \|a\|^2$, and $\mathbb{E}_u[(u^\top a)u] = a$.*

3. *For any function $f(x) : \mathbb{R}^d \to \mathbb{R}$ and $\lambda > 0$, we define its zeroth-order gradient estimator as $g_\lambda(x) = \frac{f(x+\lambda u) - f(x-\lambda u)}{2\lambda}u$ and the smoothed function as $f_\lambda(x) = \mathbb{E}_u[f(x + \lambda u)]$. Then the following properties hold*

   (a) *$f_\lambda(x)$ is differentiable and $\mathbb{E}_u[g_\lambda(x)] = \nabla f_\lambda(x)$.*

   (b) *If $f(x)$ is $L$-smooth, then we have*

$$\|\nabla f(x) - \nabla f_\lambda(x)\| \leq \frac{L}{2}\lambda d^{3/2},$$

$$\mathbb{E}_u[\|g_\lambda(x)\|^2] \leq 2d \cdot \|\nabla f(x)\|^2 + \frac{L^2}{2}\lambda^2 d^3.$$

## B.2 Convergence of PAZO-M

**Theorem B.3** (Full statement of Theorem 4.1). *Let the private and public data be $\gamma$-similar and Assumption 1, 2, 3, and 4 hold. For possibly non-convex $f(\cdot)$, running Algorithm 1 for $T$ rounds gives*

$$\frac{1}{T}\sum_{t=0}^{T-1}[\|\nabla f(x_t)\|^2]\leq\frac{16\sqrt{d}L[f(x_0)-f(x_*)]}{T}\frac{(1-\alpha)^2\sqrt{d}+\alpha(1-\alpha)}{(2(1-\alpha)+\alpha\sqrt{d})^2}+2L\lambda d^{\frac{5}{4}}M\frac{1-\alpha}{2(1-\alpha)+\alpha\sqrt{d}}$$

$$+2\gamma^2\frac{\alpha\sqrt{d}}{2(1-\alpha)+\alpha\sqrt{d}}+\left[\frac{L^2\lambda^2d^2}{4}+\frac{\sigma_1^2\sqrt{d}}{b}+\frac{d\sigma^2C^2}{2b^2}\right]\frac{1-\alpha}{(1-\alpha)\sqrt{d}+\alpha}$$

$$+\frac{\sigma_2^2}{2b'}\frac{\alpha^2\sqrt{d}}{(1-\alpha)^2\sqrt{d}+\alpha(1-\alpha)}+\left[\frac{L\lambda d^{\frac{5}{4}}\gamma}{2}+\left(\gamma+\frac{L\lambda d^{\frac{5}{4}}}{2}\right)M\right]\frac{\alpha}{(1-\alpha)\sqrt{d}+\alpha}.$$

*Additionally, let $c_1$ and $c_2$ be the constants that make PAZO-M satisfy $(\varepsilon,\delta)$-differential privacy for any $\varepsilon < c_1 b^2 T/n^2, \delta > 0$. Then PAZO-M obtains the error rate*

$$O\left(\frac{1-\alpha}{\alpha}\sqrt{d}\right)+O\left(\gamma^2\frac{\alpha\sqrt{d}}{2(1-\alpha)+\alpha\sqrt{d}}+\frac{\sigma_1^2}{b}\frac{(1-\alpha)\sqrt{d}}{(1-\alpha)\sqrt{d}+\alpha}+\frac{\sigma_2^2}{b'}\frac{\alpha^2\sqrt{d}}{(1-\alpha)^2\sqrt{d}+\alpha(1-\alpha)}\right)$$

*by choosing the parameters*

$$\eta=\frac{2(1-\alpha)+\alpha\sqrt{d}}{4L((1-\alpha)^2\sqrt{d}+\alpha(1-\alpha))},\quad \lambda\leq\frac{1}{Ld^{\frac{5}{4}}},\quad C=1+\sqrt{2}d^{\frac{1}{4}}M,\quad and$$

$$T=\frac{4n\varepsilon[(1-\alpha)\sqrt{d}+\alpha]}{c_2C[2(1-\alpha)+\alpha\sqrt{d}]}\sqrt{\frac{2L[f(x_0)-f(x_*)]}{\sqrt{d}\log(1/\delta)}}.$$

*Proof.* We choose the clipping threshold $C$ large enough such that clipping does not happen, then the update rule is $x_{t+1}-x_t=-\eta_t((1-\alpha)(\Delta(x_t;u_t,B_t)+z_t)u_t+\alpha g'(x_t;B_t'))$ where

$$\Delta(x_t;u_t,B_t)=\frac{1}{b}\sum_{\xi_i\in B_t}\frac{f(x_t+\lambda u_t;\xi_i)-f(x_t-\lambda u_t;\xi_i)}{2\lambda}.$$

At a step $t$, let $x_t$ be a fixed parameter. We apply the update to the property of $L$-smooth objectives and take expectation over all the randomness at this iteration, i.e., $\mathbb{E}_t:=\mathbb{E}_{u_t,z_t,B_t,B_t'}$. We have

$$\mathbb{E}_t[f(x_{t+1})]$$

$$\leq f(x_t)+\langle\nabla f(x_t),\mathbb{E}_t[x_{t+1}-x_t]\rangle+\frac{L}{2}\mathbb{E}_t[\|x_{t+1}-x_t\|^2]$$

$$= f(x_t)-(1-\alpha)\eta_t\nabla\underbrace{f(x_t)^\top\mathbb{E}_t[\Delta(x_t;u_t,B_t)u_t]}_{T_1}+\frac{(1-\alpha)^2L\eta_t^2\sqrt{d}}{2}\underbrace{\mathbb{E}_t[\Delta(x_t;u_t,B_t)^2]}_{T_2}$$

$$+\underbrace{\frac{\alpha^2L\eta_t^2}{2}\mathbb{E}_t[\|g'(x_t;B_t')\|^2]-\alpha\eta_t\nabla f(x_t)^\top g_t'+\alpha(1-\alpha)L\eta_t^2\mathbb{E}_t\left[\Delta(x_t;u_t,B_t)u_t^\top g'(x_t;B_t')\right]}_{T_3}$$

$$+\frac{(1-\alpha)^2L\eta_t^2\sqrt{d}\sigma^2C^2}{2b^2}.$$

For $T_1$, note that $\mathbb{E}_u[\Delta(x_t; u)u] = \mathbb{E}_t[\Delta(x_t; u_t, B_t)u_t]$ and when $\lambda \to 0$, it holds that $f_\lambda(x_t) := \mathbb{E}_u[\Delta(x_t; u)u] = \mathbb{E}_{u_t}[u_t u_t^\top \nabla f(x_t)] = \frac{1}{\sqrt{d}} \nabla f(x_t)$ for $u_t \sim \text{Unif}(d^{\frac{1}{4}} \mathbb{S}^{d-1})$. We thus obtain

$$
\begin{aligned}
&-\nabla f(x_t)^\top \mathbb{E}_t[\Delta(x_t; u_t, B_t)u_t] \\
&= -\nabla f(x_t)^\top \mathbb{E}_{u_t}[\Delta(x_t; u_t)u_t] \\
&= -\langle \nabla f(x_t)^\top, \nabla f(x_t) + \mathbb{E}_{u_t}[\Delta(x_t; u_t)u_t] - \nabla f(x_t) \rangle \\
&\leq -\|\nabla f(x_t)\|^2 + \|\nabla f(x_t)\| \|\mathbb{E}_{u_t}[\Delta(x_t; u_t)u_t] - \nabla f(x_t)\|
\end{aligned}
$$

$$
\leq -\|\nabla f(x_t)\|^2 + \|\nabla f(x_t)\| \left[ \underbrace{\left\|\mathbb{E}_{u_t}[\Delta(x_t; u_t)u_t] - \frac{1}{\sqrt{d}} \nabla f(x_t)\right\|}_{T_5} + \left(1 - \frac{1}{\sqrt{d}}\right)\|\nabla f(x_t)\| \right] \quad (7)
$$

where $T_5$ satisfies

$$
\begin{aligned}
\left\|\frac{1}{\sqrt{d}} \nabla f(x_t) - \mathbb{E}_{u_t}[\Delta(x_t; u_t)u_t]\right\| &\leq \mathbb{E}_t\left[\left\|\left(\nabla f(x_t)^\top u_t - \frac{f(x_t + \lambda u_t) - f(x_t - \lambda u_t)}{2\lambda}\right)u_t\right\|\right] \\
&= \frac{d^{\frac{1}{4}}}{2\lambda} \mathbb{E}_t\left[|\left(f(x_t + \lambda u_t) - f(x_t - \lambda u_t) - 2\lambda \nabla f(x_t)^\top u_t\right)|\right] \\
&\leq \frac{d^{\frac{1}{4}}}{2\lambda} \mathbb{E}_t\left[|\left(f(x_t + \lambda u_t) - f(x_t) - \lambda \nabla f(x_t)^\top u_t\right)|\right] \\
&\quad + \frac{d^{\frac{1}{4}}}{2\lambda} \mathbb{E}_t\left[|\left(f(x_t) - f(x_t - \lambda u_t) - \lambda \nabla f(x_t)^\top u_t\right)|\right] \\
&\leq \frac{L\lambda d^{\frac{3}{4}}}{2}
\end{aligned}
$$

due to L-smoothness applied to the last inequality. Therefore, $-\nabla f(x_t)^\top \mathbb{E}_t[\Delta(x_t; u_t, B_t)u_t] \leq -\frac{1}{\sqrt{d}}\|\nabla f(x_t)\| + \frac{L\lambda d^{\frac{3}{4}}}{2} M$.

For $T_2$, note that per-sample $L$-smoothness implies batch $L$-smoothness. Therefore, we follow Zhang et al. [8] by noting that

$$
\begin{aligned}
\Delta(x_t; u_t, B_t)^2 &= \frac{(f(x_t + \lambda u_t; B_t) - f(x_t - \lambda u_t; B_t) - 2\lambda u_t^\top \nabla f(x_t; B_t) + 2\lambda u_t^\top \nabla f(x_t; B_t))^2}{4\lambda^2} \\
&\overset{(a)}{\leq} \frac{(f(x_t + \lambda u_t; B_t) - f(x_t - \lambda u_t; B_t) - 2\lambda u_t^\top \nabla f(x_t; B_t))^2 + (2\lambda u_t^\top \nabla f(x_t; B_t))^2}{2\lambda^2} \\
&\overset{(b)}{\leq} \frac{(f(x_t + \lambda u_t; B_t) - f(x_t; B_t) - \lambda u_t^\top \nabla f(x_t; B_t))^2}{\lambda^2} \\
&\quad + \frac{(f(x_t; B_t) - f(x_t - \lambda u_t; B_t) - \lambda u_t^\top \nabla f(x_t; B_t))^2}{\lambda^2} + 2(u_t^\top \nabla f(x_t; B_t))^2 \\
&\overset{(c)}{\leq} \frac{L^2 \lambda^2 d}{2} + 2(u_t^\top \nabla f(x_t; B_t))^2
\end{aligned}
$$

where $(a)$ and $(b)$ follow $(a + b)^2 \leq 2(a^2 + b^2)$ and $(c)$ follows $|f(x + \lambda u) - f(x) - \lambda u^\top \nabla f(x)| \leq L\lambda^2 d/2$ and $|f(x) - f(x - \lambda u) - \lambda u^\top \nabla f(x)| \leq L\lambda^2 d/2$ due to $L$-smoothness. Therefore,

$$
\begin{aligned}
\mathbb{E}_{u_t}[\Delta(x_t; u_t, B_t)^2] &\overset{(a)}{=} \frac{L^2 \lambda^2 d}{2} + \frac{2}{\sqrt{d}}\|\nabla f(x_t; B_t)\|^2 \\
&\leq \frac{L^2 \lambda^2 d}{2} + \frac{2}{\sqrt{d}}\|\nabla f(x_t)\|^2 + \frac{2\sigma_1^2}{b\sqrt{d}} \quad (8)
\end{aligned}
$$

where $(a)$ follows Lemma B.2 (2).

For $T_3$, applying the equalities

$$
\mathbb{E}_{B_t'}[\|g'(x_t; B_t')\|^2] = \|g'\|^2 + \frac{\sigma_2^2}{b'},
$$

$$\nabla f(x_t)^\top g_t' = \frac{1}{2}(\|g_t'\|^2 + \|\nabla f(x_t)\|^2 - \|g_t' - \nabla f(x_t)\|^2),$$

$$\mathbb{E}_{u_t, B_t, B_t'}[\Delta(x_t; u_t, B_t)u_t^\top g'(x_t; B_t')] = \nabla f_\lambda(x_t)^\top g_t'$$
$$= \frac{1}{2}(\|g_t'\|^2 + \|\nabla f_\lambda(x_t)\|^2 - \|g_t' - \nabla f_\lambda(x_t)\|^2)$$

gives us

$$T_3 = \frac{\alpha L \eta_t^2}{2}\left[\left(1 - \frac{1}{L\eta_t}\right)\|g_t'\|^2 + (1 - \alpha)\|\nabla f_\lambda(x_t)\|^2 - (1 - \alpha)\|g_t' - \nabla f_\lambda(x_t)\|^2\right] + T_4, \quad (9)$$

where

$$T_4 = \frac{\alpha \eta_t}{2}\|g_t' - \nabla f(x_t)\|^2 + \frac{\alpha^2 L \eta_t^2 \sigma_2^2}{2b'} - \frac{\alpha \eta_t}{2}\|\nabla f(x_t)\|^2$$
$$\le \frac{\alpha \eta_t}{2}\gamma^2 + \frac{\alpha^2 L \eta_t^2 \sigma_2^2}{2b'} - \frac{\alpha \eta_t}{2}\|\nabla f(x_t)\|^2. \quad (10)$$

We take $\alpha$ and $\eta_t$ so that $\alpha L \eta_t < 1$, which implies $1 - \frac{1}{L\eta_t} < 1 - \alpha$. We thus have

$$T_3 \le \frac{\alpha(1-\alpha)L\eta_t^2}{2}\left[\|g_t'\|^2 + \|\nabla f_\lambda(x_t)\|^2 - \|g_t' - \nabla f_\lambda(x_t)\|^2\right] + T_4$$
$$= \alpha(1-\alpha)\langle g_t', \nabla f_\lambda(x_t)\rangle + T_4$$
$$\le \alpha(1-\alpha)\|g_t'\|\|\nabla f_\lambda(x_t)\| + T_4$$
$$\le \alpha(1-\alpha)(\|g_t' - \nabla f(x_t)\| + \|\nabla f(x_t)\|)(\|\nabla f_\lambda(x_t) - \nabla f(x_t)\| + \|\nabla f(x_t)\|) + T_4$$
$$\le \alpha(1-\alpha)(\gamma L\lambda d^{\frac{3}{4}}/2 + (\gamma/\sqrt{d} + L\lambda d^{\frac{3}{4}}/2)M + \|\nabla f(x_t)\|^2/\sqrt{d}) + T_4. \quad (11)$$

Combining $T_1$ (7), $T_2$ (8), $T_3$ (11), and $T_4$ (10) yields

$$\left[\frac{\eta_t(1-\alpha)}{\sqrt{d}} + \frac{\eta_t\alpha}{2} - L\eta_t^2(1-\alpha)^2 - \frac{L\eta_t^2\alpha(1-\alpha)}{\sqrt{d}}\right]\|\nabla f(x_t)\|^2$$
$$\le f(x_t) - \mathbb{E}_t[f(x_{t+1})] + \frac{(1-\alpha)L\eta_t\lambda d^{\frac{3}{4}}M}{2} + \frac{(1-\alpha)^2 L\eta_t^2\sigma_1^2}{b}$$
$$+ \frac{(1-\alpha)^2 L^3\eta_t^2\lambda^2 d^{\frac{3}{2}}}{4} + \frac{(1-\alpha)^2 L\eta_t^2\sigma^2 C^2\sqrt{d}}{2b^2} + \frac{\alpha\eta_t\gamma^2}{2}$$
$$+ \frac{\alpha^2 L\eta_t^2\sigma_2^2}{2b'} + \frac{\alpha(1-\alpha)L^2\eta_t^2\gamma\lambda d^{\frac{3}{4}}}{2} + \alpha(1-\alpha)L\eta_t^2 M\left(\frac{\gamma}{\sqrt{d}} + \frac{L\lambda d^{\frac{3}{4}}}{2}\right).$$

Choosing $\eta_t = \frac{2(1-\alpha)+\alpha\sqrt{d}}{4L((1-\alpha)^2\sqrt{d}+\alpha(1-\alpha))}$, we have $\alpha L\eta_t < 1$ if $\alpha < 1 - \frac{3\sqrt{d}-3}{3\sqrt{d}-2}$. Denote $\mathbb{E}_{<t} := \mathbb{E}_{u_{<t}, z_{<t}, B_{<t}, B_{<t}'}$ where $u_{<t}$ is the set $\{u_0, \ldots, u_{t-1}\}$ and similarly for $z_{<t}$, $B_{<t}$, and $B_{<t}'$. We sum up from $t = 0$ to $T - 1$, telescope terms, and divide both sides by $T$ to obtain

$$\frac{1}{T}\sum_{t=0}^{T-1}[\|\nabla f(x_t)\|^2]$$
$$\le \frac{16\sqrt{d}L[f(x_0) - f(x_*)]}{T}\frac{(1-\alpha)^2\sqrt{d}+\alpha(1-\alpha)}{(2(1-\alpha)+\alpha\sqrt{d})^2} + 2L\lambda d^{\frac{5}{4}}M\frac{1-\alpha}{2(1-\alpha)+\alpha\sqrt{d}}$$
$$+ 2\sqrt{d}\gamma^2\frac{\alpha}{2(1-\alpha)+\alpha\sqrt{d}} + \left[\frac{L^2\lambda^2 d^2}{4} + \frac{\sigma_1^2\sqrt{d}}{b} + \frac{d\sigma^2 C^2}{2b^2}\right]\frac{1-\alpha}{(1-\alpha)\sqrt{d}+\alpha}$$
$$+ \frac{\sqrt{d}\sigma_2^2}{2b'}\frac{\alpha^2}{(1-\alpha)^2\sqrt{d}+\alpha(1-\alpha)} + \left[\frac{L\lambda d^{\frac{5}{4}}\gamma}{2} + \left(\gamma + \frac{L\lambda d^{\frac{5}{4}}}{2}\right)M\right]\frac{\alpha}{(1-\alpha)\sqrt{d}+\alpha}. \quad (12)$$

By privacy analysis in Section 3, we take $\sigma = c_2 b\sqrt{T\log(1/\delta)}/(n\varepsilon)$ and then there exist constants $c_1$ and $c_2$ such that PAZO-M is $(\varepsilon,\delta)$-differentially private for any $\varepsilon < c_1 b^2 T/n^2, \delta > 0$. We apply $\eta_t$ and $\sigma$ to Eq. (12) and obtain

$$\frac{1}{T}\sum_{t=0}^{T-1}[\|\nabla f(x_t)\|^2]$$

$$\leq \frac{16\sqrt{d}L[f(x_0)-f(x_*)]}{T}\frac{(1-\alpha)^2\sqrt{d}+\alpha(1-\alpha)}{(2(1-\alpha)+\alpha\sqrt{d})^2} + 2L\lambda d^{\frac{5}{4}}M\frac{1-\alpha}{2(1-\alpha)+\alpha\sqrt{d}}$$

$$+ 2\sqrt{d}\gamma^2\frac{\alpha}{2(1-\alpha)+\alpha\sqrt{d}} + \left[\frac{L^2\lambda^2 d^2}{4} + \frac{\sigma_1^2\sqrt{d}}{b} + \frac{c_2^2 C^2 dT\log(1/\delta)}{2n^2\varepsilon^2}\right]\frac{1-\alpha}{(1-\alpha)\sqrt{d}+\alpha}$$

$$+ \frac{\sqrt{d}\sigma_2^2}{2b'}\frac{\alpha^2}{(1-\alpha)^2\sqrt{d}+\alpha(1-\alpha)} + \left[\frac{L\lambda d^{\frac{5}{4}}\gamma}{2} + \left(\gamma+\frac{L\lambda d^{\frac{5}{4}}}{2}\right)M\right]\frac{\alpha}{(1-\alpha)\sqrt{d}+\alpha}. \quad (13)$$

To choose the optimal $T$, we organize the terms involving $T$, which are of the form $\frac{p}{T}+qT$. We solve $\min_{T>0}\frac{p}{T}+qT = 2\sqrt{pq}$ by taking $T^* = \sqrt{p/q}$, which yields

$$T^* = \frac{4n\varepsilon[(1-\alpha)\sqrt{d}+\alpha]}{c_2 C[2(1-\alpha)+\alpha\sqrt{d}]}\sqrt{\frac{2L[f(x_0)-f(x_*)]}{\sqrt{d}\log(1/\delta)}}.$$

By $\Delta(x_t;u_t,\xi_i)^2 \leq \frac{L^2\lambda^2 d}{2} + 2(u_t^\top\nabla f(x_t;\xi_i))^2$ and per-sample $M$-Lipschitz, we have

$$\Delta(x_t;u_t,\xi_i) \leq \sqrt{d^{-\frac{3}{2}}/2 + 2\sqrt{d}M^2} \leq 1 + \sqrt{2}d^{\frac{1}{4}}M$$

due to $\sqrt{p+q} \leq \sqrt{p} + \sqrt{q}$ for $p,q \geq 0$ and choosing $\lambda \leq \frac{1}{Ld^{\frac{5}{4}}}$. We choose $C = 1 + \sqrt{2}d^{\frac{1}{4}}M$ and thus have

$$\frac{1}{T}\sum_{t=0}^{T-1}[\|\nabla f(x_t)\|^2]$$

$$\leq \frac{4c_2(1+\sqrt{2}d^{\frac{1}{4}}M)(1-\alpha)d^{\frac{3}{4}}}{n\varepsilon[2(1-\alpha)+\alpha\sqrt{d}]}\sqrt{2L[f(x_0)-f(x_*)]\log(1/\delta)} + 2M\frac{1-\alpha}{2(1-\alpha)+\alpha\sqrt{d}}$$

$$+ 2\gamma^2\frac{\alpha\sqrt{d}}{2(1-\alpha)+\alpha\sqrt{d}} + \left[\frac{1}{4\sqrt{d}} + \frac{\sigma_1^2\sqrt{d}}{b}\right]\frac{1-\alpha}{(1-\alpha)\sqrt{d}+\alpha}$$

$$+ \frac{\sigma_2^2}{2b'}\frac{\alpha^2\sqrt{d}}{(1-\alpha)^2\sqrt{d}+\alpha(1-\alpha)} + \left[\frac{\gamma}{2} + \left(\gamma+\frac{1}{2}\right)M\right]\frac{\alpha}{(1-\alpha)\sqrt{d}+\alpha},$$

which indicates that the error depends on $d, \sigma_1, \sigma_2$, and $\gamma$ by

$$O\left(\frac{1-\alpha}{\alpha}\sqrt{d}\right) + O\left(\gamma^2\frac{\alpha\sqrt{d}}{2(1-\alpha)+\alpha\sqrt{d}} + \frac{\sigma_1^2}{b}\frac{(1-\alpha)\sqrt{d}}{(1-\alpha)\sqrt{d}+\alpha} + \frac{\sigma_2^2}{b'}\frac{\alpha^2\sqrt{d}}{(1-\alpha)^2\sqrt{d}+\alpha(1-\alpha)}\right).$$

Therefore, we have error dependence $O(\frac{1-\alpha}{\alpha}\sqrt{d})$, which saves a factor of $\log d$ compared to DPZero's $O(\sqrt{d}\log d)$, together with constant improvement if $\alpha > \frac{1}{2}$. We additionally have the error term $O(\gamma^2 + \sigma_2^2/b')$ that reduces as $\alpha$ decreases due to using biased public gradients. $\qquad\square$

### B.3 Convergence of PAZO-P

**Theorem B.4** (Full statement of Theorem 4.2). *Let the private and public data be $\gamma$-similar and Assumption 1, 2, 3, and 4 hold. For possibly non-convex $f(\cdot)$, running Algorithm 2 for $T$ rounds gives*

$$\frac{1}{T}\sum_{t=0}^{T-1}\mathbb{E}_{<t}[\|\nabla f(x_t)\|^2] \leq \frac{4Lk}{T}[f(x_0)-f(x_*)] + 2M\sqrt{2\left(\frac{\sigma_2^2}{b'}+\gamma^2\right)}$$

$$+ L\lambda k^{\frac{3}{2}}M + \frac{L^2\lambda^2 k^2}{4} + \frac{\sigma_1^2}{b} + \frac{\sigma^2 C^2}{2b^2}.$$

*Additionally, let $c_1$ and $c_2$ be the constants that make PAZO-M satisfy $(\varepsilon, \delta)$-differential privacy for any $\varepsilon < c_1 b^2 T/n^2, \delta > 0$. Then PAZO-P obtains the error rate*

$$O(k) + O\left(\sqrt{\gamma^2 + \frac{\sigma_2^2}{b'} + \frac{\sigma_1^2}{b}}\right)$$

*by choosing the parameters*

$$\eta = \frac{1}{2Lk}, \quad \lambda \leq \frac{1}{Lk^{\frac{3}{2}}}, \quad C = 1 + \sqrt{2k}M, \quad and \ T = \frac{n\varepsilon}{c_2 C}\sqrt{\frac{8Lk[f(x_0) - f(x_*)]}{\log(1/\delta)}}.$$

*Proof.* We choose the clipping threshold $C$ large enough such that clipping does not happen, then the update rule is $x_{t+1} - x_t = -\eta_t(\Delta(x_t; u_t, B_t) + z_t)G_t u_t$ where

$$\Delta(x_t; u_t, B_t) = \frac{1}{b}\sum_{\xi_i \in B_t} \frac{f(x_t + \lambda G_t u_t; \xi_i) - f(x_t - \lambda G_t u_t; \xi_i)}{2\lambda}.$$

At a step $t$, let $x_t$ be a fixed parameter. We apply the update to the property of $L$-smooth objectives and take expectation over all the randomness at this iteration, i.e., $\mathbb{E}_t := \mathbb{E}_{u_t, z_t, B_t, B_t'}$. We have

$$\mathbb{E}_t[f(x_{t+1})]$$

$$\leq f(x_t) + \langle \nabla f(x_t), \mathbb{E}_t[x_{t+1} - x_t]\rangle + \frac{L}{2}\mathbb{E}_t[\|x_{t+1} - x_t\|^2]$$

$$= f(x_t) - \eta_t\langle \nabla f(x_t), \mathbb{E}_t[\Delta(x_t; u_t, B_t)G_t u_t]\rangle + \frac{L\eta_t^2}{2}\mathbb{E}_t[\|\Delta(x_t; u_t, B_t)G_t u_t\|^2] + \frac{L\eta_t^2}{2}\mathbb{E}_t\left[\left\|\frac{z_t}{b}G_t u_t\right\|^2\right]$$

$$\overset{(a)}{=} f(x_t) - \eta_t\|\nabla f(x_t)\|^2 + \eta_t\underbrace{\langle \nabla f(x_t), \nabla f(x_t) - \mathbb{E}_t[\Delta(x_t; u_t, B_t)G_t u_t]\rangle}_{T_1}$$

$$+ \frac{L\eta_t^2 k}{2}\underbrace{\mathbb{E}_t[\|\Delta(x_t; u_t, B_t)\|^2]}_{T_2} + \frac{L\eta_t^2\sigma^2 C^2 k}{2b^2}, \tag{14}$$

where $(a)$ is due to the orthonormality of $G_t$ and thus $\|G_t u_t\| = \|u_t\| = \sqrt{k}$.

For $T_1$, we proceed by

$$\langle \nabla f(x_t), \nabla f(x_t) - \mathbb{E}_t[\Delta(x_t; u_t, B_t)G_t u_t]\rangle$$

$$\leq \|\nabla f(x_t)\| \|\nabla f(x_t) - \mathbb{E}_t[\Delta(x_t; u_t, B_t)G_t u_t]\|$$

$$\leq \|\nabla f(x_t)\| [\underbrace{\|\nabla f(x_t) - \mathbb{E}_t[G_t G_t^\top \nabla f(x_t)]\|}_{T_3} + \underbrace{\|\mathbb{E}_t[G_t G_t^\top \nabla f(x_t)] - \mathbb{E}_t[\Delta(x_t; u_t, B_t)G_t u_t]\|}_{T_4}].$$

For a $G_t$, we denote its un-orthonormalized columns as $\{g'(x_t; B_{t,1}'), \ldots, g'(x_t; B_{t,k}')\}$. Note that for any public candidate index $i \in [k]$, we have

(i) $g'(x_t; B_{t,i}') \in \text{Col}(G_t)$

(ii) $\mathbb{E}_t[\|g(x_t; B_{t,i}') - \nabla f(x_t)\|^2] = \mathbb{E}_t[\|g(x_t; B_{t,i}') - g_t' + g_t' - \nabla f(x_t)\|^2]$

$$\overset{(a)}{\leq} 2\mathbb{E}_t[\|g(x_t; B_{t,i}') - g_t'\|^2] + \|g_t' - \nabla f(x_t)\|^2$$

$$\overset{(b)}{\leq} 2(\sigma_2^2/b' + \gamma^2)$$

where $(a)$ holds due to $(a + b)^2 \leq 2(a^2 + b^2)$ and $(b)$ follows the $\gamma$-similar assumption. Therefore,

$$\left(\mathbb{E}_t[\|\nabla f(x_t) - G_t G_t^\top \nabla f(x_t)\|]\right)^2 \overset{(a)}{\leq} \mathbb{E}_t[\|\nabla f(x_t) - G_t G_t^\top \nabla f(x_t)\|^2]$$

$$\overset{(b)}{\leq} \mathbb{E}_t[\|\nabla f(x_t) - g(x_t; B_{t,i}')\|^2]$$

$$\leq 2(\sigma_2^2/b' + \gamma^2),$$

where $(a)$ follows Jensen's inequality and $(b)$ is due to the fact that $\left\|\nabla f(x_t) - G_t G_t^\top \nabla f(x_t)\right\| \leq \|\nabla f(x_t) - x\|$ for any $x \in \mathrm{Col}(G_t)$.

For $T_3$, we thus have

$$\left\|\nabla f(x_t) - \mathbb{E}_t[G_t G_t^\top \nabla f(x_t)]\right\| \leq \mathbb{E}_t[\|\nabla f(x_t) - G_t G_t^\top \nabla f(x_t)\|]$$
$$\leq \sqrt{2(\sigma_2^2/b' + \gamma^2)}. \tag{15}$$

For $T_4$, we have

$$\left\|\mathbb{E}_t[G_t G_t^\top \nabla f(x_t)] - \mathbb{E}_t[\Delta(x_t; u_t, B_t) G_t u_t]\right\|$$
$$= \left\|\mathbb{E}_t[\nabla f(x_t)^\top G_t u_t G_t u_t - \Delta(x_t; u_t) G_t u_t]\right\|$$
$$\leq \mathbb{E}_t\left[\left\|\left(\nabla f(x_t)^\top G_t u_t - \frac{f(x_t + \lambda G_t u_t) - f(x_t - \lambda G_t u_t)}{2\lambda}\right) G_t u_t\right\|\right]$$
$$= \frac{\sqrt{k}}{2\lambda}\mathbb{E}_t\left[|\left(f(x_t + \lambda G_t u_t) - f(x_t - \lambda G_t u_t) - 2\lambda \nabla f(x_t)^\top G_t u_t\right)|\right]$$
$$\leq \frac{\sqrt{k}}{2\lambda}\mathbb{E}_t\left[|\left(f(x_t + \lambda G_t u_t) - f(x_t) - \lambda \nabla f(x_t)^\top G_t u_t\right)|\right]$$
$$+ \frac{\sqrt{k}}{2\lambda}\mathbb{E}_t\left[|\left(f(x_t) - f(x_t - \lambda G_t u_t) - \lambda \nabla f(x_t)^\top G_t u_t\right)|\right]$$
$$\leq \frac{L\lambda k^{\frac{3}{2}}}{2}$$

where the last inequality is due to L-smoothness. Therefore,

$$T_1 \leq M\left(\sqrt{2(\sigma_2^2/b' + \gamma^2)} + \frac{L\lambda k^{\frac{3}{2}}}{2}\right). \tag{16}$$

For $T_2$, note that

$$\Delta(x_t; u_t, B_t)^2$$
$$= \frac{(f(x_t + \lambda G_t u_t; B_t) - f(x_t - \lambda G_t u_t; B_t) - 2\lambda u_t^\top G_t^\top \nabla f(x_t; B_t) + 2\lambda u_t^\top G_t^\top \nabla f(x_t; B_t))^2}{4\lambda^2}$$
$$\overset{(a)}{\leq} \frac{(f(x_t + \lambda G_t u_t; B_t) - f(x_t - \lambda G_t u_t; B_t) - 2\lambda u_t^\top G_t^\top \nabla f(x_t; B_t))^2 + (2\lambda u_t^\top G_t^\top \nabla f(x_t; B_t))^2}{2\lambda^2}$$
$$\overset{(b)}{\leq} \frac{(f(x_t + \lambda G_t u_t; B_t) - f(x_t; B_t) - \lambda u_t^\top G_t^\top \nabla f(x_t; B_t))^2}{\lambda^2}$$
$$+ \frac{(f(x_t; B_t) - f(x_t - \lambda G_t u_t; B_t) - \lambda u_t^\top \nabla f(x_t; B_t))^2}{\lambda^2} + 2(u_t^\top G_t^\top \nabla f(x_t; B_t))^2$$
$$\overset{(c)}{\leq} \frac{L^2 \lambda^2 k^2}{2} + 2(u_t^\top G_t^\top \nabla f(x_t; B_t))^2,$$

where $(a)$ and $(b)$ are implied by $(a+b)^2 \leq 2(a^2 + b^2)$ and $(c)$ uses the facts $|f(x + \lambda u) - f(x) - \lambda u^\top \nabla f(x)| \leq L\lambda^2 d/2$ and $|f(x) - f(x - \lambda u) - \lambda u^\top \nabla f(x)| \leq L\lambda^2 d/2$ due to $L$-smoothness. Therefore, applying Lemma B.2 (2) gives us

$$\mathbb{E}_t[\|\Delta(x_t; u_t, B_t)\|^2] = \frac{L^2 \lambda^2 k^2}{2} + 2\mathbb{E}_{B_t, B_t'}\mathbb{E}_{u_t}[(u_t^\top G_t^\top \nabla f(x_t; B_t))^2]$$
$$= \frac{L^2 \lambda^2 k^2}{2} + 2\mathbb{E}_{B_t, B_t'}[\|G_t^\top \nabla f(x_t; B_t)\|^2]$$
$$= \frac{L^2 \lambda^2 k^2}{2} + 2\mathbb{E}_{B_t, B_t'}[\nabla f(x_t; B_t)^\top G_t G_t^\top \nabla f(x_t; B_t)]$$
$$= \frac{L^2 \lambda^2 k^2}{2} + 2\mathbb{E}_{B_t, B_t'}[\nabla f(x_t; B_t)^\top \mathrm{Proj}_G(\nabla f(x_t; B_t))]$$
$$\leq \frac{L^2 \lambda^2 k^2}{2} + 2\mathbb{E}_{B_t}[\|\nabla f(x_t; B_t)\|^2]$$

$$\leq \frac{L^2\lambda^2k^2}{2} + 2\left(\|\nabla f(x_t)\|^2 + \frac{\sigma_1^2}{b}\right). \tag{17}$$

Applying $T_1$ (16) and $T_2$ (17) to Eq. (14) yields

$$(\eta_t - L\eta_t^2 k)\|\nabla f(x_t)\|^2 \leq f(x_t) - \mathbb{E}_t[f(x_{t+1})] + \eta_t M\left(\sqrt{2(\frac{\sigma_2^2}{b'} + \gamma^2)} + \frac{L\lambda k^{\frac{3}{2}}}{2}\right)$$
$$+ \frac{L^3\eta_t^2\lambda^2k^3}{4} + \frac{L\eta_t^2 k\sigma_1^2}{b} + \frac{L\eta_t^2\sigma^2C^2k}{2b^2}.$$

We choose $\eta_t = \frac{1}{2Lk}$ so that $\eta_t - L\eta_t^2 k = \frac{\eta_t}{2}$. Denote $\mathbb{E}_{<t} := \mathbb{E}_{u_{<t}, z_{<t}, B_{<t}, B'_{<t}}$ where $u_{<t}$ is the set $\{u_0, \ldots, u_{t-1}\}$ and similarly for $z_{<t}$, $B_{<t}$, and $B'_{<t}$. Then we have

$$\mathbb{E}_{<t}\|\nabla f(x_t)\|^2 \leq 4Lk\mathbb{E}_{<t+1}[f(x_t) - f(x_{t+1})] + 2M\sqrt{2\left(\frac{\sigma_2^2}{b'} + \gamma^2\right)} + L\lambda k^{\frac{3}{2}}M$$
$$+ \frac{L^2\lambda^2k^2}{4} + \frac{\sigma_1^2}{b} + \frac{\sigma^2C^2}{2b^2}.$$

Summing up from $t = 0$ to $T - 1$ and dividing both sides by $T$ yields

$$\frac{1}{T}\sum_{t=0}^{T-1}\mathbb{E}_{<t}[\|\nabla f(x_t)\|^2] \leq \frac{4Lk}{T}[f(x_0) - f(x_*)] + 2M\sqrt{2\left(\frac{\sigma_2^2}{b'} + \gamma^2\right)}$$
$$+ L\lambda k^{\frac{3}{2}}M + \frac{L^2\lambda^2k^2}{4} + \frac{\sigma_1^2}{b} + \frac{\sigma^2C^2}{2b^2}. \tag{18}$$

By privacy analysis in Section 3, we take $\sigma = c_2 b\sqrt{T\log(1/\delta)}/(n\varepsilon)$ and then there exist constants $c_1$ and $c_2$ such that PAZO-P is $(\varepsilon, \delta)$-differentially private for any $\varepsilon < c_1 b^2 T/n^2, \delta > 0$. We apply $\eta_t$ and $\sigma$ to Eq. (18) and obtain the RHS of Eq. (18) as

$$\frac{4Lk[f(x_0) - f(x_*)]}{T} + 2M\sqrt{2\left(\frac{\sigma_2^2}{b'} + \gamma^2\right)} + L\lambda k^{\frac{3}{2}}M + \frac{L^2\lambda^2k^2}{4} + \frac{\sigma_1^2}{b} + \frac{c_2^2C^2\log(1/\delta)T}{2n^2\varepsilon^2}.$$

Choosing the optimal $T$ again requires solving $\arg\min_{T>0}\frac{p}{T} + qT = \sqrt{p/q}$, which yields

$$T^* = \frac{n\varepsilon}{c_2 C}\sqrt{\frac{8Lk[f(x_0) - f(x_*)]}{\log(1/\delta)}}.$$

By $\Delta(x_t; u_t, \xi_i)^2 \leq \frac{L^2\lambda^2k^2}{2} + 2(u_t^\top G_t^\top \nabla f(x_t; \xi_i))^2$ and per-sample $M$-Lipschitz, we have

$$\Delta(x_t; u_t, \xi_i) \leq \sqrt{k^{-1}/2 + 2kM^2} \leq 1 + \sqrt{2k}M$$

due to choosing $\lambda \leq \frac{1}{Lk^{\frac{3}{2}}}$. We take $C = 1 + \sqrt{2k}M$ and thus the RHS of Eq. (18) becomes

$$\frac{2(1 + \sqrt{2k}M)c_2}{n\varepsilon}\sqrt{2Lk[f(x_0) - f(x_*)]\log(1/\delta)} + 2M]\sqrt{2\left(\frac{\sigma_2^2}{b'} + \gamma^2\right)} + M + \frac{1}{4k} + \frac{\sigma_1^2}{b},$$

which indicates that the error depends on $k, \sigma_1, \sigma_2$, and $\gamma$ by

$$O(k) + O\left(\sqrt{\frac{\sigma_2^2}{b'} + \gamma^2} + \frac{\sigma_1^2}{b}\right).$$

Therefore, we have $d$-independent error rate $O(k)$, which is an improvement due to $k$ being a small constant $\ll \log d$ in practice. We additionally have the error term $O(\gamma^2 + \sigma_2^2/b')$ from the biased public gradients and $O(\sigma_1^2/b)$ from the stochastic private gradients. $\qquad\square$

## B.4 Convergence of PAZO-S

**Theorem B.5** (Full statement of Theorem 4.3). *Let the private and public data be $\gamma$-similar and Assumption 1, 2, 3, and 4 hold. For possibly non-convex $f(\cdot)$, running Algorithm 3 for $T$ rounds using a fixed step size $\eta = \frac{1}{4L}$ and $\epsilon \le 1/\sqrt{d}$ gives*

$$\frac{1}{T} \sum_{t=0}^{T-1} \mathbb{E}_{<t}[\|\nabla f(x_t)\|^2] \le \frac{8L\mathbb{E}_{<t+1}[f(x_0) - f(x_*)]}{T} + 2M\left(\gamma + \frac{\sigma_2}{\sqrt{b'}}\right) + 2\gamma^2 + \frac{2\sigma_2^2}{b'} + \frac{1}{2}.$$

*Additionally, let $c_1$ and $c_2$ be the constants that make PAZO-S satisfy $(\varepsilon, \delta)$-differential privacy for any $\varepsilon < c_1 b^2 T/n^2, \delta > 0$. Then by taking $T \to \infty$, PAZO-S obtains the error rate $O\left(\gamma^2 + \sigma_2^2/b'\right)$.*

*Proof.* Our public data sampling process is equivalent to first sampling $B'_t$ and then dividing it into $k$ non-overlapping partitions. We choose the clipping threshold $C$ large enough such that clipping does not happen, then the update rule is $x_{t+1} - x_t = -\eta_t(g'(x_t; B'_{t,I}) + \mathbb{1}(z')z')$ where $I := \arg\min_{i \in [k]}\{f(x_t - \eta_t g(x_t; B'_{t,i}); B_t) + z_{t,i}\}$ is the index of public batch that yields the best public gradients and $\mathbb{1}(z')$ is an indicator variable denoting whether the proposal of adding $z' \sim \mathcal{N}(0, \epsilon^2 I_d)$ is adopted.

At a step $t$, let $x_t$ be a fixed parameter. We apply the update to the property of $L$-smooth objectives and take expectation over all the randomness at this iteration, i.e., $\mathbb{E}_t := \mathbb{E}_{z_t, B_t, B'_t}$. We have

$$\begin{aligned}
&\mathbb{E}_t[f(x_{t+1})] \\
&= \mathbb{E}_t[f(x_t - \eta_t(g'(x_t; B'_{t,I}) + \mathbb{1}(z')z'))] \\
&\le f(x_t) - \eta_t\left\langle \nabla f(x_t), \mathbb{E}_t[g'(x_t; B'_{t,I}) + \mathbb{1}(z')z']\right\rangle + \frac{L\eta_t^2}{2}\underbrace{\mathbb{E}_t[\|g'(x_t; B'_{t,I}) + \mathbb{1}(z')z'\|^2]}_{T_1} \\
&= f(x_t) - \eta_t\left\langle \nabla f(x_t), \mathbb{E}_t[g'(x_t; B'_{t,I})]\right\rangle + \frac{L\eta_t^2}{2} T_1 \\
&= f(x_t) - \eta_t\|\nabla f(x_t)\|^2 + \eta_t\left\langle \nabla f(x_t), \nabla f(x_t) - \mathbb{E}_t[g'(x_t; B'_{t,I})]\right\rangle + \frac{L\eta_t^2}{2} T_1 \\
&\le f(x_t) - \eta_t\|\nabla f(x_t)\|^2 + \eta_t\|\nabla f(x_t)\|\underbrace{\mathbb{E}_t[\|\nabla f(x_t) - g'(x_t; B'_{t,I})\|]}_{T_2} + \frac{L\eta_t^2}{2} T_1.
\end{aligned}$$

For $T_1$, we have

$$\begin{aligned}
\mathbb{E}_t[\|g'(x_t; B'_{t,I}) + \mathbb{1}(z')z'\|^2] &\le 2\mathbb{E}_t[\|g'(x_t; B'_{t,I})\|^2] + 2\mathbb{E}_t[\|\mathbb{1}(z')z'\|^2] \\
&\le 2\mathbb{E}_t[\|g'(x_t; B'_{t,I}) - \nabla f(x_t) + \nabla f(x_t)\|^2] + 2d\epsilon^2 \\
&\le 4\mathbb{E}_t[\|g'(x_t; B'_{t,I}) - \nabla f(x_t)\|^2] + 4\|\nabla f(x_t)\|^2 + 2d\epsilon^2 \\
&= 4\mathbb{E}_t[\|g'_t - g_t + \frac{1}{b'}\sum_{j \in B'_t} \zeta_{t,j}^{(I)'}\|^2] + 4\|\nabla f(x_t)\|^2 + 2d\epsilon^2 \\
&\le 8\gamma^2 + \frac{8\sigma_2^2}{b'} + 4\|\nabla f(x_t)\|^2 + 2d\epsilon^2.
\end{aligned}$$

For $T_2$, we note that for a sampled public batch $i \in [k]$, its gradient is $g'(x_t; B'_{t,i}) = g'_t + \frac{1}{b'}\sum_{j=1}^{b'} \zeta_{t,j}^{(i)'}$ where $\zeta_{t,j}^{(i)'}$ is the stochastic gradient noise for the public sample $j$ in the $i$-th batch. We denote the selected best batch as $I$ and thus

$$\mathbb{E}_{B'_t}[\|g'_t - g'(x_t; B'_{t,I})\|^2] = \mathbb{E}_{B'_t}\left[\left\|\frac{1}{b'}\sum_{j=1}^{b'} \zeta_{t,j}^{(I)'}\right\|^2\right] = \frac{1}{b'}\mathbb{E}_{B'_t}\left[\left\|\zeta_t^{(I)'}\right\|^2\right].$$

By assumption, $\mathbb{E}_{B'_t}\left[\left\|\zeta_t^{(i)'}\right\|^2\right] \leq \sigma_2^2$ for any batch $i$. Therefore,

$$\mathbb{E}_{B'_t}\left[\left\|\zeta_t^{(I)'}\right\|^2\right] = \mathbb{E}_i\left[\mathbb{E}_{B'_t}\left[\left\|\zeta_t^{(I)'}\right\|^2\right]|I=i\right] \leq \sigma_2^2.$$

Therefore, $(\mathbb{E}_t[\|g_t - g'(x_t; B'_{t,I})\|])^2 \leq \mathbb{E}_t[\|g_t - g'(x_t; B'_{t,I})\|^2] \leq \sigma_2^2/b'$ and

$$\mathbb{E}_t[\|\nabla f(x_t) - g'(x_t; B'_{t,I})\|] \leq \mathbb{E}_t[\|\nabla f(x_t) - g'_t\|] + \mathbb{E}_t[\|g'_t - g'(x_t; B'_{t,I})\|]$$
$$\leq \gamma + \sigma_2/\sqrt{b'}.$$

Denote $\mathbb{E}_{<t} := \mathbb{E}_{z_{<t}, B_{<t}, B'_{<t}}$ where $z_{<t}$ is the set $\{z_0, \ldots, z_{t-1}\}$ and similarly for $B_{<t}$ and $B'_{<t}$. We have

$$(\eta_t - 2L\eta_t^2)\mathbb{E}_{<t}\|\nabla f(x_t)\|^2 \leq \mathbb{E}_{<t+1}[f(x_t) - f(x_{t+1})] + \eta_t M\left(\gamma + \frac{\sigma_2}{\sqrt{b'}}\right) + 4L\eta_t^2\left(\gamma^2 + \frac{\sigma_2^2}{b'} + \frac{d\epsilon^2}{4}\right).$$

We set $\epsilon \leq 1/\sqrt{d}$ and choose $\eta_t = \frac{1}{4L}$ so that $2L\eta_t^2 = \eta_t/2$. We sum up from $t = 0$ to $T - 1$, and dividing both sides by $T$ yields

$$\frac{1}{T}\sum_{t=0}^{T-1}\mathbb{E}_{<t}[\|\nabla f(x_t)\|^2] \leq \frac{8L\mathbb{E}_{<t+1}[f(x_0) - f(x_*)]}{T} + 2M\left(\gamma + \frac{\sigma_2}{\sqrt{b'}}\right) + 2\gamma^2 + \frac{2\sigma_2^2}{b'} + \frac{1}{2}.$$

We take $T \to \infty$ and achieve a $d$-independent error bound $O(\gamma^2 + \sigma_2^2/b')$. When $\gamma$ approaches zero, the remaining term $\sigma_2^2/b'$ is due to stochastic public data sampling. $\qquad\square$

## B.5 Convergence of PAZO-M under Bounded Loss

**Theorem B.6** (PAZO-M under Bounded Loss). *Assume public and private data are $\gamma$-similar. Let Assumption 1, 2, 3, and 4 hold. Assume that $|f(x_t)| \leq S$ for all t, then for possibly non-convex $f(\cdot)$ and any $\alpha \in (0, \frac{2}{3})$, Algorithm 1 has the error rate*

$$\frac{1}{T}\sum_{t=0}^{T-1}\mathbb{E}[\|\nabla f(x_t)\|^2] \leq O\left(\frac{1-\alpha}{\alpha}d^{\frac{1}{4}}\right)$$
$$+ O\left(\gamma^2\frac{4\alpha\sqrt{d}}{4(1-\alpha)+\alpha\sqrt{d}} + \frac{\sigma_1^2}{b}\frac{(1-\alpha)\sqrt{d}}{(1-\alpha)\sqrt{d}+\alpha} + \frac{\sigma_2^2}{b'}\frac{\alpha^2\sqrt{d}}{(1-\alpha)^2\sqrt{d}+\alpha(1-\alpha)}\right)$$

*by choosing the parameters*

$$\eta = \frac{4(1-\alpha)+\alpha\sqrt{d}}{8L(1-\alpha)((1-\alpha)\sqrt{d}+\alpha)}, \quad T = \frac{8n\varepsilon[(1-\alpha)\sqrt{d}+\alpha]}{c_2[4(1-\alpha)+\alpha\sqrt{d}]}\sqrt{\frac{2L}{\sqrt{d}\log(1/\delta)}},$$

$$\lambda \leq \frac{2(\sqrt{2}-1)C_0}{Ld^{\frac{3}{4}}}, \quad C^2 = 2C_0^2 = 16M^2\log\left(\frac{32\sqrt{L\pi}bn\varepsilon\tilde{S}}{c_2[4(1-\alpha)+\alpha\sqrt{d}]^3\sqrt{d}\log(1/\delta)}\right) \quad \text{where}$$

$$\tilde{S} = 128\sqrt{d}SL(1-\alpha)[(1-\alpha)\sqrt{d}+\alpha]^2 + 8d(1-\alpha)M^2[4(1-\alpha)+\alpha\sqrt{d}][(1-\alpha)\sqrt{d}+\alpha]$$
$$+ \alpha dM(\gamma+M)[4(1-\alpha)+\alpha\sqrt{d}]^2.$$

*Proof.* The update rule is $x_{t+1} - x_t = -\eta_t((1-\alpha)(\hat{\Delta}(x_t; u_t, B_t) + z_t)u_t + \alpha g'(x_t; B'_t))$ where

$$\hat{\Delta}(x_t; u_t, B_t) = \frac{1}{b}\sum_{\xi_i \in B_t}\text{clip}_C\left(\frac{f(x_t + \lambda u_t; \xi_i) - f(x_t - \lambda u_t; \xi_i)}{2\lambda}\right).$$

When clipping is ineffective due to large $C$, we denote the non-clipped version as $\Delta(x_t; u_t, B_t)$. Since $f(x; \xi)$ is $L$-smooth, we have

$$\frac{|f(x_t+\lambda u_t;\xi_i)-f(x_t-\lambda u_t;\xi_i)|}{2\lambda} \leq |u_t^\top \nabla f(x_t;\xi_i)| + \frac{|f(x_t+\lambda u_t;\xi_i)-f(x_t;\xi_i)-\lambda u_t^\top \nabla f(x_t;\xi_i)|}{2\lambda}$$

$$+ \frac{|f(x_t-\lambda u_t;\xi_i)-f(x_t;\xi_i)+\lambda u_t^\top \nabla f(x_t;\xi_i)|}{2\lambda}$$

$$\leq |u_t^\top \nabla f(x_t;\xi_i)| + \frac{L\lambda\sqrt{d}}{2}.$$

Therefore, by the per-sample Lipschitz assumption, we have

$$\mathbb{P}\left(\frac{|f(x_t+\lambda u_t;\xi_i)-f(x_t-\lambda u_t;\xi_i)|}{2\lambda} \geq C_0 + \frac{L\lambda\sqrt{d}}{2}\right) \leq \mathbb{P}(u_t^\top \nabla f(x_t;\xi_i) \geq C_0)$$

$$\leq 2\sqrt{2\pi}\exp\left(-\frac{C_0^2}{8\|\nabla f(x_t;\xi_i)\|^2}\right)$$

$$\leq 2\sqrt{2\pi}\exp\left(-\frac{C_0^2}{8M^2}\right).$$

Denote $\bar{Q}_{t,i}$ as the event that clipping happens for sample $\xi_i$ at iteration $t$, and $\bar{Q}$ as the event that clipping happens for some $\xi_i$ at some iteration $t$. Then, following Zhang et al. [8], if we choose the clipping threshold $C \geq C_0 + L\lambda\sqrt{d}/2$, the probability of event $\bar{Q}$ is upper-bounded. By the union bound, we have

$$\mathbb{P}(\bar{Q}) = \mathbb{P}\left(\bigcup_{t=0}^{T-1}\bigcup_{i=1}^{b}\bar{Q}_{t,i}\right) \leq 2\sqrt{2\pi}bT\exp\left(-\frac{C_0^2}{8M^2}\right). \tag{19}$$

At a step $t$, let $x_t$ be a fixed parameter. We apply the update to the property of $L$-smooth objectives and take expectation over all the randomness at this iteration, i.e., $\mathbb{E}_t := \mathbb{E}_{u_t, z_t, B_t, B_t'}$. Note that the scalar noise $z_t$ is independent of $u_t$, $\hat{\Delta}_t$, and the event $Q_t$, i.e., clipping does not happen for any $\xi_i$ at iteration $t$. Conditioned on $Q_t$, $\hat{\Delta}(x_t; u_t, B_t)$ is equal to $\Delta(x_t; u_t, B_t)$ and we have

$$\mathbb{E}_t[f(x_{t+1})|Q_t]$$

$$\leq f(x_t) + \langle \nabla f(x_t), \mathbb{E}_t[x_{t+1}-x_t|Q_t]\rangle + \frac{L}{2}\mathbb{E}_t[\|x_{t+1}-x_t\|^2|Q_t]$$

$$= f(x_t) - (1-\alpha)\eta_t \underbrace{\nabla f(x_t)^\top \mathbb{E}_t[\Delta(x_t;u_t,B_t)u_t|Q_t]}_{T_1} + \frac{(1-\alpha)^2 L\eta_t^2\sqrt{d}}{2}\underbrace{\mathbb{E}_t[\Delta(x_t;u_t,B_t)^2|Q_t]}_{T_2}$$

$$+ \underbrace{\frac{\alpha^2 L\eta_t^2}{2}\mathbb{E}_t[\|g'(x_t;B_t')\|^2|Q_t] - \alpha\eta_t\nabla f(x_t)^\top g_t' + \alpha(1-\alpha)L\eta_t^2\mathbb{E}_t[\Delta(x_t;u_t,B_t)u_t^\top g'(x_t;B_t')|Q_t]}_{T_3}$$

$$+ \frac{(1-\alpha)^2 L\eta_t^2\sqrt{d}\sigma^2 C^2}{2b^2}. \tag{20}$$

For $T_1$, note that $\mathbb{E}_u[\Delta(x_t;u)u] = \mathbb{E}_t[\Delta(x_t;u_t,B_t)u_t]$ and when $\lambda \to 0$, it holds that $f_\lambda(x_t) := \mathbb{E}_u[\Delta(x_t;u)u] = \mathbb{E}_{u_t}[u_t u_t^\top \nabla f(x_t)] = \frac{1}{\sqrt{d}}\nabla f(x_t)$ for $u_t \sim \text{Unif}(d^{\frac{1}{4}}\mathbb{S}^{d-1})$. We thus have

$$-\nabla f(x_t)^\top \mathbb{E}_t[\Delta(x_t;u_t,B_t)u_t]$$

$$= -\nabla f(x_t)^\top \mathbb{E}_{u_t}[\Delta(x_t;u_t)u_t]$$

$$= -\langle \nabla f(x_t), \nabla f(x_t) + \mathbb{E}_{u_t}[\Delta(x_t;u_t)u_t] - \nabla f(x_t)\rangle$$

$$\leq -\|\nabla f(x_t)\|^2 + \|\nabla f(x_t)\|\|\mathbb{E}_{u_t}[\Delta(x_t;u_t)u_t] - \nabla f(x_t)\|$$

$$\leq -\|\nabla f(x_t)\|^2 + \|\nabla f(x_t)\|\left[\underbrace{\left\|\mathbb{E}_{u_t}[\Delta(x_t;u_t)u_t] - \frac{1}{\sqrt{d}}\nabla f(x_t)\right\|}_{T_4} + \left(1-\frac{1}{\sqrt{d}}\right)\|\nabla f(x_t)\|\right] \tag{21}$$

where $T_4$ satisfies

$$\left\|\frac{1}{\sqrt{d}}\nabla f(x_t) - \mathbb{E}_{u_t}[\Delta(x_t; u_t)u_t]\right\| \leq \mathbb{E}_t\left[\left\|\left(\nabla f(x_t)^\top u_t - \frac{f(x_t + \lambda u_t) - f(x_t - \lambda u_t)}{2\lambda}\right)u_t\right\|\right]$$

$$= \frac{d^{\frac{1}{4}}}{2\lambda}\mathbb{E}_t\left[|\left(f(x_t + \lambda u_t) - f(x_t - \lambda u_t) - 2\lambda\nabla f(x_t)^\top u_t\right)|\right]$$

$$\leq \frac{d^{\frac{1}{4}}}{2\lambda}\mathbb{E}_t\left[|\left(f(x_t + \lambda u_t) - f(x_t) - \lambda\nabla f(x_t)^\top u_t\right)|\right]$$

$$+ \frac{d^{\frac{1}{4}}}{2\lambda}\mathbb{E}_t\left[|\left(f(x_t) - f(x_t - \lambda u_t) - \lambda\nabla f(x_t)^\top u_t\right)|\right]$$

$$\leq \frac{L\lambda d^{\frac{3}{4}}}{2}$$

due to L-smoothness applied to the last inequality. Therefore, $-\nabla f(x_t)^\top\mathbb{E}_t[\Delta(x_t; u_t, B_t)u_t] \leq -\frac{1}{\sqrt{d}}\|\nabla f(x_t)\|^2 + \frac{L\lambda M}{2}d^{\frac{3}{4}}$.

By the law of total expectation, we have

$$\mathbb{E}_t[\Delta(x_t; u_t, B_t)u_t|Q_t]\mathbb{P}(Q_t) + \mathbb{E}_t[\Delta(x_t; u_t, B_t)u_t|\bar{Q}_t]\mathbb{P}(\bar{Q}_t) = \mathbb{E}_t[\Delta(x_t; u_t, B_t)u_t]$$

and thus

$$-\nabla f(x_t)^\top\mathbb{E}_t[\Delta(x_t; u_t, B_t)u_t|Q_t]$$

$$= -\frac{\nabla f(x_t)^\top\mathbb{E}_t[\Delta(x_t; u_t, B_t)u_t]}{\mathbb{P}(Q_t)} + \frac{\nabla f(x_t)^\top\mathbb{E}_t[\Delta(x_t; u_t, B_t)u_t|\bar{Q}_t]\mathbb{P}(\bar{Q}_t)}{\mathbb{P}(Q_t)}$$

$$\leq -\frac{1}{\sqrt{d}\mathbb{P}(Q_t)}\|\nabla f(x_t)\|^2 + \frac{L\lambda M}{2\mathbb{P}(Q_t)}d^{\frac{3}{4}} + \frac{M^2\sqrt{d}\mathbb{P}(\bar{Q}_t)}{\mathbb{P}(Q_t)}.$$

For $T_2$, note that per-sample $L$-smoothness implies batch $L$-smoothness. Therefore, we have

$$\Delta(x_t; u_t, B_t)^2 = \frac{(f(x_t + \lambda u_t; B_t) - f(x_t - \lambda u_t; B_t) - 2\lambda u_t^\top\nabla f(x_t; B_t) + 2\lambda u_t^\top\nabla f(x_t; B_t))^2}{4\lambda^2}$$

$$\overset{(a)}{\leq} \frac{(f(x_t + \lambda u_t; B_t) - f(x_t - \lambda u_t; B_t) - 2\lambda u_t^\top\nabla f(x_t; B_t))^2 + (2\lambda u_t^\top\nabla f(x_t; B_t))^2}{2\lambda^2}$$

$$\overset{(b)}{\leq} \frac{(f(x_t + \lambda u_t; B_t) - f(x_t; B_t) - \lambda u_t^\top\nabla f(x_t; B_t))^2}{\lambda^2}$$

$$+ \frac{(f(x_t; B_t) - f(x_t - \lambda u_t; B_t) - \lambda u_t^\top\nabla f(x_t; B_t))^2}{\lambda^2} + 2(u_t^\top\nabla f(x_t; B_t))^2$$

$$\overset{(c)}{\leq} \frac{L^2\lambda^2 d}{2} + 2(u_t^\top\nabla f(x_t; B_t))^2$$

where $(a)$ and $(b)$ follow $(a + b)^2 \leq 2(a^2 + b^2)$ and $(c)$ follows $|f(x + \lambda u) - f(x) - \lambda u^\top\nabla f(x)| \leq L\lambda^2 d/2$ and $|f(x) - f(x - \lambda u) - \lambda u^\top\nabla f(x)| \leq L\lambda^2 d/2$ due to $L$-smoothness. Therefore,

$$\mathbb{E}_t[\Delta(x_t; u_t, B_t)^2] \overset{(a)}{=} \frac{L^2\lambda^2 d}{2} + \frac{2}{\sqrt{d}}\mathbb{E}_{B_t}[\|\nabla f(x_t; B_t)\|^2]$$

$$\leq \frac{L^2\lambda^2 d}{2} + \frac{2}{\sqrt{d}}\|\nabla f(x_t)\|^2 + \frac{2\sigma_1^2}{b\sqrt{d}} \qquad (22)$$

where $(a)$ follows Lemma B.2 (2). By the law of total expectation,

$$\mathbb{E}_t[\Delta(x_t; u_t, B_t)^2] = \mathbb{E}_t[\Delta(x_t; u_t, B_t)^2|Q_t]\mathbb{P}(Q_t) + \mathbb{E}_t[\Delta(x_t; u_t, B_t)^2|\bar{Q}_t]\mathbb{P}(\bar{Q}_t)$$

$$\geq \mathbb{E}_t[\Delta(x_t; u_t, B_t)^2|Q_t]\mathbb{P}(Q_t),$$

so we have

$$\mathbb{E}_t[\Delta(x_t; u_t, B_t)^2 | Q_t] \leq \frac{\mathbb{E}_t[\Delta(x_t; u_t, B_t)^2]}{\mathbb{P}(Q_t)}$$
$$\leq \frac{L^2 \lambda^2 d}{2\mathbb{P}(Q_t)} + \frac{2}{\sqrt{d}\mathbb{P}(Q_t)} \|\nabla f(x_t)\|^2 + \frac{2\sigma_1^2}{b\sqrt{d}\mathbb{P}(Q_t)}.$$

For $T_3$, note that

$$\mathbb{E}_{u_t, B_t, B_t'}[\Delta(x_t; u_t, B_t) u_t^\top g'(x_t; B_t')] = \nabla f_\lambda(x_t)^\top g_t'$$
$$= \frac{1}{2}(\|g_t'\|^2 + \|\nabla f_\lambda(x_t)\|^2 - \|g_t' - \nabla f_\lambda(x_t)\|^2),$$

so by the law of total expectation,

$$\mathbb{E}_t\left[\Delta(x_t; u_t, B_t) u_t^\top g'(x_t; B_t') | Q_t\right]$$
$$= \mathbb{E}_t\left[\Delta(x_t; u_t, B_t) u_t | Q_t\right]^\top g_t'$$
$$= \frac{\mathbb{E}_t\left[\Delta(x_t; u_t, B_t) u_t\right]^\top g_t'}{\mathbb{P}(Q_t)} - \frac{\mathbb{P}(\bar{Q}_t)}{\mathbb{P}(Q_t)} \mathbb{E}_t\left[\Delta(x_t; u_t, B_t) u_t | \bar{Q}_t\right]^\top g_t'$$
$$\leq \frac{\mathbb{E}_t\left[\Delta(x_t; u_t, B_t) u_t\right]^\top g_t'}{\mathbb{P}(Q_t)} + \frac{\mathbb{P}(\bar{Q}_t)}{\mathbb{P}(Q_t)} M\sqrt{d}(\gamma + M)$$
$$\leq \frac{1}{2\mathbb{P}(Q_t)}(\|g_t'\|^2 + \|\nabla f_\lambda(x_t)\|^2 - \|g_t' - \nabla f_\lambda(x_t)\|^2) + \frac{\mathbb{P}(\bar{Q}_t)}{\mathbb{P}(Q_t)} M\sqrt{d}(\gamma + M).$$

Additionally,

$$\mathbb{E}_{B_t'}[\|g'(x_t; B_t')\|^2] \leq \|g'\|^2 + \frac{\sigma_2^2}{b'},$$

$$\nabla f(x_t)^\top g_t' = \frac{1}{2}(\|g_t'\|^2 + \|\nabla f(x_t)\|^2 - \|g_t' - \nabla f(x_t)\|^2),$$

so we apply the above to Eq. (20) and have

$$\mathbb{E}_t[f(x_{t+1}) - f(x_t) | Q_t]$$
$$\leq (1-\alpha)\eta_t \left[-\frac{1}{\sqrt{d}\mathbb{P}(Q_t)} \|\nabla f(x_t)\|^2 + \frac{L\lambda M}{2\mathbb{P}(Q_t)} d^{\frac{3}{4}} + \frac{M^2 \sqrt{d}\mathbb{P}(\bar{Q}_t)}{\mathbb{P}(Q_t)}\right]$$
$$+ \frac{(1-\alpha)^2 L\eta_t^2 \sqrt{d}}{2\mathbb{P}(Q_t)} \left[\frac{L^2 \lambda^2 d}{2} + \frac{2}{\sqrt{d}} \|\nabla f(x_t)\|^2 + \frac{2\sigma_1^2}{b\sqrt{d}}\right]$$
$$+ \frac{\alpha^2 L\eta_t^2}{2\mathbb{P}(Q_t)} \left(\|g_t'\|^2 + \frac{\sigma_2^2}{b'}\right) - \frac{\alpha\eta_t}{2}(\|g_t'\|^2 + \|\nabla f(x_t)\|^2 - \|g_t' - \nabla f(x_t)\|^2)$$
$$+ \frac{\alpha(1-\alpha)L\eta_t^2}{\mathbb{P}(Q_t)} \left[\frac{1}{2}(\|g_t'\|^2 + \|\nabla f_\lambda(x_t)\|^2 - \|g_t' - \nabla f_\lambda(x_t)\|^2) + \mathbb{P}(\bar{Q}_t) M\sqrt{d}(\gamma + M)\right]$$
$$+ \frac{(1-\alpha)^2 L\eta_t^2 \sqrt{d}\sigma^2 C^2}{2b^2 \mathbb{P}(Q_t)},$$

which is

$$\mathbb{E}_t[f(x_{t+1}) - f(x_t) | Q_t]\mathbb{P}(Q_t)$$
$$\leq (1-\alpha)\eta_t \left[-\frac{1}{\sqrt{d}} \|\nabla f(x_t)\|^2 + \frac{L\lambda M}{2} d^{\frac{3}{4}} + M^2 \sqrt{d}\mathbb{P}(\bar{Q}_t)\right] + \alpha(1-\alpha)L\eta_t^2 \mathbb{P}(\bar{Q}_t) M\sqrt{d}(\gamma + M)$$
$$+ \frac{(1-\alpha)^2 L\eta_t^2 \sqrt{d}}{2} \left[\frac{L^2 \lambda^2 d}{2} + \frac{2}{\sqrt{d}} \|\nabla f(x_t)\|^2 + \frac{2\sigma_1^2}{b\sqrt{d}}\right] + \frac{\alpha^2 L\eta_t^2 \sigma_2^2}{2b'} + \frac{(1-\alpha)^2 L\eta_t^2 \sqrt{d}\sigma^2 C^2}{2b^2} + T_5$$

where

$$T_5 = \frac{\alpha^2 L\eta_t^2 \|g_t'\|^2}{2} - \frac{\alpha\eta_t \mathbb{P}(Q_t)}{2}(\|\nabla f(x_t)\|^2 + \|g_t'\|^2 - \|g_t' - \nabla f(x_t)\|^2)$$

$$+ \frac{\alpha(1-\alpha)L\eta_t^2}{2}\left[\|g_t'\|^2 + \|\nabla f_\lambda(x_t)\|^2 - \|g_t' - \nabla f_\lambda(x_t)\|^2\right]$$

$$\leq \frac{\alpha L\eta_t^2}{2}\left(1 - \frac{\mathbb{P}(Q_t)}{L\eta_t}\right)\|g_t'\|^2 + \frac{\alpha\eta_t}{2}\|g_t' - \nabla f(x_t)\|^2 - \frac{\alpha\eta_t \mathbb{P}(Q_t)}{2}\|\nabla f(x_t)\|^2$$

$$+ \frac{\alpha(1-\alpha)L\eta_t^2}{2}\left[\|\nabla f_\lambda(x_t)\|^2 - \|g_t' - \nabla f_\lambda(x_t)\|^2\right].$$

If we have $C_0^2 \geq 8M^2 \log(4\sqrt{2\pi}b)$, then $\mathbb{P}(\bar{Q}_t) = \mathbb{P}\left(\bigcup_{i=1}^b \bar{Q}_{t,i}\right) \leq 2\sqrt{2\pi}b \exp\left(-\frac{C_0^2}{8M^2}\right) \leq \frac{1}{2}$ and thus $\mathbb{P}(Q_t) = 1 - \mathbb{P}(\bar{Q}_t) \geq \frac{1}{2}$. We choose $\eta_t$ and $\alpha$ such that $L\eta_t\alpha \leq \frac{1}{2}$, which implies that $1 - \frac{\mathbb{P}(Q_t)}{L\eta_t} \leq 1 - \alpha$. Then we have

$$T_5 \leq \frac{\alpha(1-\alpha)L\eta_t^2}{2}\left[\|g_t'\|^2 + \|\nabla f_\lambda(x_t)\|^2 - \|g_t' - \nabla f_\lambda(x_t)\|^2\right] + \frac{\alpha\eta_t\gamma^2}{2} - \frac{\alpha\eta_t}{4}\|\nabla f(x_t)\|^2$$

$$= \alpha(1-\alpha)L\eta_t^2\langle g_t', \nabla f_\lambda(x_t)\rangle + \frac{\alpha\eta_t\gamma^2}{2} - \frac{\alpha\eta_t}{4}\|\nabla f(x_t)\|^2$$

$$\leq \alpha(1-\alpha)L\eta_t^2 \|g_t'\| \|\nabla f_\lambda(x_t)\| + \frac{\alpha\eta_t\gamma^2}{2} - \frac{\alpha\eta_t}{4}\|\nabla f(x_t)\|^2$$

$$\leq \alpha(1-\alpha)L\eta_t^2 (\|g_t' - \nabla f(x_t)\| + \|\nabla f(x_t)\|)\left(\left\|\nabla f_\lambda(x_t) - \frac{1}{\sqrt{d}}\nabla f(x_t)\right\| + \frac{1}{\sqrt{d}}\|\nabla f(x_t)\|\right)$$

$$+ \frac{\alpha\eta_t\gamma^2}{2} - \frac{\alpha\eta_t}{4}\|\nabla f(x_t)\|^2$$

$$\leq \alpha(1-\alpha)L\eta_t^2 (\gamma + \|\nabla f(x_t)\|)\left(\frac{L\lambda d^{\frac{3}{4}}}{2} + \frac{1}{\sqrt{d}}\|\nabla f(x_t)\|\right) + \frac{\alpha\eta_t\gamma^2}{2} - \frac{\alpha\eta_t}{4}\|\nabla f(x_t)\|^2$$

$$\leq \alpha(1-\alpha)L\eta_t^2\left[\frac{\gamma L\lambda d^{\frac{3}{4}}}{2} + \left(\frac{\gamma}{\sqrt{d}} + \frac{L\lambda d^{\frac{3}{4}}}{2}\right)M + \frac{\|\nabla f(x_t)\|^2}{\sqrt{d}}\right] + \frac{\alpha\eta_t\gamma^2}{2} - \frac{\alpha\eta_t}{4}\|\nabla f(x_t)\|^2.$$

Additionally, by the assumption that $|f(x_t)| \leq S$ for all $t$, we have

$$\mathbb{E}_t[f(x_t) - f(x_{t+1})|Q_t]\mathbb{P}(Q_t)$$

$$= \mathbb{E}_t[f(x_t) - f(x_{t+1})|Q_t \cap Q]\mathbb{P}(Q_t \cap Q) + \mathbb{E}_t[f(x_t) - f(x_{t+1})|Q_t \cap \bar{Q}]\mathbb{P}(Q_t \cap \bar{Q})$$

$$= \mathbb{E}_t[f(x_t) - f(x_{t+1})|Q]\mathbb{P}(Q) + \mathbb{E}_t[f(x_t) - f(x_{t+1})|Q_t \cap \bar{Q}]\mathbb{P}(Q_t \cap \bar{Q})$$

$$\leq \mathbb{E}_t[f(x_t) - f(x_{t+1})|Q]\mathbb{P}(Q) + 2S\mathbb{P}(\bar{Q}).$$

We additionally apply $\mathbb{P}(\bar{Q}_t) \leq \mathbb{P}(\bar{Q})$ and obtain

$$\left[\frac{\eta_t(1-\alpha)}{\sqrt{d}} + \frac{\eta_t\alpha}{4} - L\eta_t^2(1-\alpha)^2 - \frac{L\eta_t^2\alpha(1-\alpha)}{\sqrt{d}}\right]\|\nabla f(x_t)\|^2$$

$$\leq \mathbb{E}_t[f(x_t) - f(x_{t+1})|Q]\mathbb{P}(Q) + \frac{(1-\alpha)L\eta_t\lambda d^{\frac{3}{4}}M}{2} + \frac{(1-\alpha)^2 L\eta_t^2\sigma_1^2}{b}$$

$$+ \frac{(1-\alpha)^2 L^3\eta_t^2\lambda^2 d^{\frac{3}{2}}}{4} + \frac{(1-\alpha)^2 L\eta_t^2\sigma^2 C^2\sqrt{d}}{2b^2} + \frac{\alpha\eta_t\gamma^2}{2}$$

$$+ \frac{\alpha^2 L\eta_t^2\sigma_2^2}{2b'} + \frac{\alpha(1-\alpha)L^2\eta_t^2\gamma\lambda d^{\frac{3}{4}}}{2} + \alpha(1-\alpha)L\eta_t^2 M\left(\frac{\gamma}{\sqrt{d}} + \frac{L\lambda d^{\frac{3}{4}}}{2}\right)$$

$$+ (1-\alpha)\eta_t M^2\sqrt{d}\mathbb{P}(\bar{Q}) + \alpha(1-\alpha)L\eta_t^2 M\sqrt{d}(\gamma + M)\mathbb{P}(\bar{Q}) + 2S\mathbb{P}(\bar{Q}).$$

Choosing $\eta_t = \frac{4(1-\alpha)+\alpha\sqrt{d}}{8L(1-\alpha)((1-\alpha)\sqrt{d}+\alpha)}$, we have $\alpha L\eta_t < \frac{1}{2}$ for any $\alpha \in (0, \frac{2}{3})$. Denote $\mathbb{E}_{<t} := \mathbb{E}_{u_{<t}, z_{<t}, B_{<t}, B'_{<t}}$ where $u_{<t}$ is the set $\{u_0, \ldots, u_{t-1}\}$ and similarly for $z_{<t}, B_{<t}$, and $B'_{<t}$. By

privacy analysis in Section 3, we take $\sigma = c_2 b \sqrt{T \log(1/\delta)}/(n\varepsilon)$ and then there exist constants $c_1$ and $c_2$ such that PAZO-M is $(\varepsilon, \delta)$-differentially private for any $\varepsilon < c_1 b^2 T/n^2, \delta > 0$. We apply $\eta_t$ and $\sigma$, sum up from $t = 0$ to $T - 1$, and divide both sides by $T$ to obtain

$$\frac{1}{T}\sum_{t=0}^{T-1}\mathbb{E}_{<t}[\|\nabla f(x_t)\|^2]$$

$$\leq \frac{64\sqrt{d}L[f(x_0)-f(x_*)]}{T}\frac{(1-\alpha)^2\sqrt{d}+\alpha(1-\alpha)}{(4(1-\alpha)+\alpha\sqrt{d})^2}+L\lambda d^{\frac{5}{4}}M\frac{4(1-\alpha)}{4(1-\alpha)+\alpha\sqrt{d}}$$

$$+\gamma^2\frac{4\alpha\sqrt{d}}{4(1-\alpha)+\alpha\sqrt{d}}+\left[\frac{L^2\lambda^2 d^2}{4}+\frac{\sigma_1^2\sqrt{d}}{b}+\frac{dc_2^2\log(1/\delta)C^2T}{2n^2\varepsilon^2}\right]\frac{1-\alpha}{(1-\alpha)\sqrt{d}+\alpha}$$

$$+\frac{\sqrt{d}\sigma_2^2}{2b'}\frac{\alpha^2}{(1-\alpha)^2\sqrt{d}+\alpha(1-\alpha)}+\left[\frac{L\lambda d^{\frac{5}{4}}\gamma}{2}+\left(\gamma+\frac{L\lambda d^{\frac{5}{4}}}{2}\right)M\right]\frac{\alpha}{(1-\alpha)\sqrt{d}+\alpha}$$

$$+\left[\frac{128\sqrt{d}SL(1-\alpha)((1-\alpha)\sqrt{d}+\alpha)}{(4(1-\alpha)+\alpha\sqrt{d})^2}+\frac{8d(1-\alpha)M^2}{4(1-\alpha)+\alpha\sqrt{d}}+\frac{\alpha dM(\gamma+M)}{(1-\alpha)\sqrt{d}+\alpha}\right]2\sqrt{2\pi}bT\exp\left(-\frac{C_0^2}{8M^2}\right).$$

If $\lambda \leq 2(\sqrt{2}-1)C_0/(L\sqrt{d})$ and $C = \sqrt{2}C_0$, then $C \geq C_0 + L\lambda\sqrt{d}/2$ holds. So we choose

$$\lambda \leq \frac{2(\sqrt{2}-1)C_0}{Ld^{\frac{3}{4}}} \leq \frac{C_0}{Ld^{\frac{3}{4}}} \quad \text{and} \quad C^2 = 2C_0^2.$$

Additionally, let

$$T=\frac{8n\varepsilon[(1-\alpha)\sqrt{d}+\alpha]}{c_2[4(1-\alpha)+\alpha\sqrt{d}]}\sqrt{\frac{2L}{\sqrt{d}\log(1/\delta)}}, \quad C_0^2=8M^2\log\left(\frac{32\sqrt{L}\pi bn\varepsilon\tilde{S}}{c_2[4(1-\alpha)+\alpha\sqrt{d}]^3\sqrt{d}\log(1/\delta)}\right)$$

where

$$\tilde{S} = 128\sqrt{d}SL(1-\alpha)[(1-\alpha)\sqrt{d}+\alpha]^2+8d(1-\alpha)M^2[4(1-\alpha)+\alpha\sqrt{d}][(1-\alpha)\sqrt{d}+\alpha]$$
$$+\alpha dM(\gamma+M)[4(1-\alpha)+\alpha\sqrt{d}]^2.$$

This yields

$$\frac{1}{T}\sum_{t=0}^{T-1}[\|\nabla f(x_t)\|^2]$$

$$\leq \frac{4c_2(1-\alpha)d^{\frac{3}{4}}\sqrt{2L\log(1/\delta)}}{n\varepsilon[4(1-\alpha)+\alpha\sqrt{d}]}[(f(x_0)-f(x_*))+C^2]+MC_0\frac{4(1-\alpha)\sqrt{d}}{4(1-\alpha)+\alpha\sqrt{d}}$$

$$+\gamma^2\frac{4\alpha\sqrt{d}}{4(1-\alpha)+\alpha\sqrt{d}}+\left(\frac{C_0^2}{4}+\frac{\sigma_1^2}{b}\right)\frac{(1-\alpha)\sqrt{d}}{(1-\alpha)\sqrt{d}+\alpha}+\frac{\sigma_2^2}{2b'}\frac{\alpha^2\sqrt{d}}{(1-\alpha)^2\sqrt{d}+\alpha(1-\alpha)}$$

$$+\left[\frac{\gamma C_0}{2}+\left(\frac{\gamma}{\sqrt{d}}+\frac{C_0}{2}\right)M\right]\frac{\alpha\sqrt{d}}{(1-\alpha)\sqrt{d}+\alpha}+\sqrt{\log(1/\delta)}d^{\frac{1}{4}}$$

with $C^2$ and $C_0^2$ defined as above. Since $C^2 = 2C_0^2 = O\left(\log\frac{(1-\alpha)^2 d+\alpha^2 d}{\alpha(1-\alpha)\sqrt{d}+\alpha^2 d}\right) = O\left(\log\frac{(1-\alpha)^2}{\alpha^2}\right)$, the error depends on $d, \sigma_1, \sigma_2$, and $\gamma$ by

$$O\left(\frac{1-\alpha}{\alpha}d^{\frac{1}{4}}\right)+O\left(\gamma^2\frac{4\alpha\sqrt{d}}{4(1-\alpha)+\alpha\sqrt{d}}+\frac{\sigma_1^2}{b}\frac{(1-\alpha)\sqrt{d}}{(1-\alpha)\sqrt{d}+\alpha}+\frac{\sigma_2^2}{b'}\frac{\alpha^2\sqrt{d}}{(1-\alpha)^2\sqrt{d}+\alpha(1-\alpha)}\right).$$

Therefore, we have error dependence $O(\frac{1-\alpha}{\alpha}d^{\frac{1}{4}})$, which saves a factor of $d^{\frac{1}{4}}\log d$ compared to DPZero's $O(\sqrt{d}\log d)$, together with constant improvement if $\alpha > \frac{1}{2}$. We additionally have the error term $O(\gamma^2 + \sigma_2^2/b')$ that reduces as $\alpha$ decreases due to using biased public gradients. $\qquad \square$

## B.6 Convergence of PAZO-P under Bounded Loss

**Theorem B.7** (PAZO-P under Bounded Loss). *Let the private and public data be $\gamma$-similar and Assumption 1, 2, 3, and 4 hold. Assume that $|f(x_t)| \leq S$ for all $t$, then for possibly non-convex $f$, Algorithm 2 has the error rate*

$$O(\sqrt{k}\log k) + O\left(\sqrt{\frac{\sigma_2^2}{b'} + \gamma^2} + \frac{\sigma_1^2}{b}\right)$$

*by choosing the parameters*

$$\eta_t = \frac{1}{4Lk}, \quad T = \frac{4n\varepsilon}{c_2}\sqrt{\frac{2Lk}{\log(1/\delta)}}, \quad \lambda \leq \frac{2(\sqrt{2}-1)C_0}{Lk^{3/2}}, \quad and$$

$$C^2 = 2C_0^2 = 16M^2 \log\left(\frac{128\sqrt{2\pi}SLkbn^2\varepsilon^2}{c_2^2 \log(1/\delta)}\right).$$

*Proof.* The update rule is $x_{t+1} - x_t = -\eta_t(\hat{\Delta}(x_t; u_t, B_t) + z_t)G_t u_t$ where $u_t \sim \mathrm{Unif}(\sqrt{k}\mathbb{S}^{k-1})$ and

$$\hat{\Delta}(x_t; u_t, B_t) = \frac{1}{b}\sum_{\xi_i \in B_t} \mathrm{clip}_C\left(\frac{f(x_t + \lambda G_t u_t; \xi_i) - f(x_t - \lambda G_t u_t; \xi_i)}{2\lambda}\right).$$

When clipping is ineffective due to large $C$, we denote the non-clipped version as $\Delta(x_t; u_t, B_t)$. We abbreviate $\Delta(x_t; u_t, B_t)$ as $\Delta_t$ and $\hat{\Delta}(x_t; u_t, B_t)$ as $\hat{\Delta}_t$. Since $f(x; \xi)$ is $L$-smooth, we have

$$\frac{|f(x_t + \lambda G_t u_t; \xi_i) - f(x_t - \lambda G_t u_t; \xi_i)|}{2\lambda}$$

$$\leq |u_t^\top G_t^\top \nabla f(x_t; \xi_i)| + \frac{|f(x_t + \lambda G_t u_t; \xi_i) - f(x_t; \xi_i) - \lambda u_t^\top G_t^\top \nabla f(x_t; \xi_i)|}{2\lambda}$$

$$+ \frac{|f(x_t - \lambda G_t u_t; \xi_i) - f(x_t; \xi_i) + \lambda u_t^\top G_t^\top \nabla f(x_t; \xi_i)|}{2\lambda}$$

$$\leq |u_t^\top G_t^\top \nabla f(x_t; \xi_i)| + \frac{L\lambda k}{2}.$$

Therefore, by the per-sample Lipschitz assumption, we have

$$\mathbb{P}\left(\frac{|f(x_t + \lambda G_t u_t; \xi_i) - f(x_t - \lambda G_t u_t; \xi_i)|}{2\lambda} \geq C_0 + \frac{L\lambda k}{2}\right) \leq \mathbb{P}(u_t^\top G_t^\top \nabla f(x_t; \xi_i) \geq C_0)$$

$$\leq 2\sqrt{2\pi}\exp\left(-\frac{C_0^2}{8\|G_t^\top \nabla f(x_t; \xi_i)\|^2}\right)$$

$$\leq 2\sqrt{2\pi}\exp\left(-\frac{C_0^2}{8M^2}\right)$$

due to $\|G_t^\top \nabla f(x_t; \xi_i)\|^2 \leq \|\nabla f(x_t; \xi_i)\|^2 \leq M^2$. Denote $\bar{Q}_{t,i}$ as the event that clipping happens for sample $\xi_i$ at iteration $t$, and $\bar{Q}$ as the event that clipping happens for some $\xi_i$ at some iteration $t$. Then if we choose the clipping threshold $C \geq C_0 + L\lambda k/2$, the probability of event $\bar{Q}$ is bounded. By the union bound, we have

$$\mathbb{P}(\bar{Q}) = \mathbb{P}\left(\bigcup_{t=0}^{T-1}\bigcup_{i=1}^{b}\bar{Q}_{t,i}\right) \leq 2\sqrt{2\pi}bT\exp\left(-\frac{C_0^2}{8M^2}\right). \tag{23}$$

At a step $t$, let $x_t$ be a fixed parameter. We apply the update to the property of $L$-smooth objectives and take expectation over all the randomness at this iteration, i.e., $\mathbb{E}_t := \mathbb{E}_{u_t, z_t, B_t, B_t'}$. Note that the scalar noise $z_t$ is independent of $G_t$, $u_t$, $\hat{\Delta}_t$, and the event $Q_t$, i.e., clipping does not happen for any

$\xi_i$ at iteration $t$. Conditioned on $Q_t$, $\hat{\Delta}_t$ is equal to $\Delta_t$ and we have

$$\mathbb{E}_t[f(x_{t+1})|Q_t]$$
$$\leq f(x_t) + \langle \nabla f(x_t), \mathbb{E}_t[x_{t+1} - x_t|Q_t]\rangle + \frac{L}{2}\mathbb{E}_t[\|x_{t+1} - x_t\|^2 |Q_t]$$
$$= f(x_t) - \eta_t\langle \nabla f(x_t), \mathbb{E}_t[\Delta_t G_t u_t|Q_t]\rangle + \frac{L\eta_t^2}{2}\mathbb{E}_t\left[\|\Delta_t G_t u_t\|^2 \middle| Q_t\right] + \frac{L\eta_t^2}{2}\mathbb{E}_t\left[\|z_t G_t u_t\|^2 |Q_t\right]$$
$$\overset{(a)}{=} f(x_t) - \eta_t \|\nabla f(x_t)\|^2 + \eta_t \underbrace{\langle \nabla f(x_t), \nabla f(x_t) - \mathbb{E}_t[\Delta_t G_t u_t|Q_t]\rangle}_{T_1}$$
$$+ \frac{L\eta_t^2 k}{2}\underbrace{\mathbb{E}_t\left[\|\Delta_t\|^2 \middle| Q_t\right]}_{T_2} + \frac{L\eta_t^2 \sigma^2 C^2 k}{2b^2}, \tag{24}$$

where $(a)$ is due to the orthonormality of $G_t$ and thus $\|G_t u_t\| = \|u_t\| = \sqrt{k}$.

For $T_1$, we proceed by

$$\langle \nabla f(x_t), \nabla f(x_t) - \mathbb{E}_t[\Delta_t G_t u_t|Q_t]\rangle$$
$$\leq \|\nabla f(x_t)\| \|\nabla f(x_t) - \mathbb{E}_t[\Delta_t G_t u_t|Q_t]\|$$
$$\leq \|\nabla f(x_t)\| \left[\|\nabla f(x_t) - \mathbb{E}_t[G_t G_t^\top \nabla f(x_t)|Q_t]\| + \|\mathbb{E}_t[G_t G_t^\top \nabla f(x_t)|Q_t] - \mathbb{E}_t[\Delta_t G_t u_t|Q_t]\|\right]$$
$$\leq \|\nabla f(x_t)\| [\underbrace{\|\nabla f(x_t) - \mathbb{E}_t[G_t G_t^\top \nabla f(x_t)]\|}_{T_3} + \underbrace{\|\mathbb{E}_t[G_t G_t^\top \nabla f(x_t) - \Delta_t G_t u_t|Q_t]\|}_{T_4}].$$

For a $G_t$, we denote its un-orthonormalized columns as $\{g'(x_t; B'_{t,1}), \ldots, g'(x_t; B'_{t,k})\}$. Note that for any public candidate index $i \in [k]$, we have

    (i) $g'(x_t; B'_{t,i}) \in \mathrm{Col}(G_t)$

    (ii) $\mathbb{E}_t[\|g(x_t; B'_{t,i}) - \nabla f(x_t)\|^2] = \mathbb{E}_t[\|g(x_t; B'_{t,i}) - g'_t + g'_t - \nabla f(x_t)\|^2]$

$$\overset{(a)}{\leq} 2\mathbb{E}_t[\|g(x_t; B'_{t,i}) - g'_t\|^2] + 2\|g'_t - \nabla f(x_t)\|^2$$
$$\overset{(b)}{\leq} 2(\sigma_2^2/b' + \gamma^2)$$

where $(a)$ holds due to $(a + b)^2 \leq 2(a^2 + b^2)$ and $(b)$ follows the $\gamma$-similar assumption. Therefore,

$$\left(\mathbb{E}_t[\|\nabla f(x_t) - G_t G_t^\top \nabla f(x_t)\|]\right)^2 \overset{(a)}{\leq} \mathbb{E}_t[\|\nabla f(x_t) - G_t G_t^\top \nabla f(x_t)\|^2]$$
$$\overset{(b)}{\leq} \mathbb{E}_t[\|\nabla f(x_t) - g(x_t; B'_{t,i})\|^2]$$
$$\leq 2(\sigma_2^2/b' + \gamma^2),$$

where $(a)$ follows Jensen's inequality and $(b)$ is due to the fact that $\|\nabla f(x_t) - G_t G_t^\top \nabla f(x_t)\| \leq \|\nabla f(x_t) - x\|$ for any $x \in \mathrm{Col}(G_t)$. We thus have

$$T_3 \leq \mathbb{E}_t[\|\nabla f(x_t) - G_t G_t^\top \nabla f(x_t)\|] \leq \sqrt{2(\sigma_2^2/b' + \gamma^2)}. \tag{25}$$

For $T_4$, by the law of total expectation, it holds for any random variable $A \geq 0$ that

$$\mathbb{E}_t[A] = \mathbb{E}_t[A|Q_t]\mathbb{P}(Q_t) + \mathbb{E}_t[A|\bar{Q}_t]\mathbb{P}(\bar{Q}_t) \geq \mathbb{E}_t[A|Q_t]\mathbb{P}(Q_t).$$

Therefore,

$$\left\| \mathbb{E}_t[G_t G_t^\top \nabla f(x_t) - \Delta(x_t; u_t, B_t) G_t u_t | Q_t] \right\|$$

$$= \left\| \mathbb{E}_t[\nabla f(x_t)^\top G_t u_t G_t u_t - \Delta(x_t; u_t) G_t u_t | Q_t] \right\|$$

$$\leq \mathbb{E}_t \left[ \left\| \left( \nabla f(x_t)^\top G_t u_t - \frac{f(x_t + \lambda G_t u_t) - f(x_t - \lambda G_t u_t)}{2\lambda} \right) G_t u_t \right\| \Big| Q_t \right]$$

$$\leq \frac{1}{\mathbb{P}(Q_t)} \mathbb{E}_t \left[ \left\| \left( \nabla f(x_t)^\top G_t u_t - \frac{f(x_t + \lambda G_t u_t) - f(x_t - \lambda G_t u_t)}{2\lambda} \right) G_t u_t \right\| \right]$$

$$= \frac{\sqrt{k}}{2\lambda \mathbb{P}(Q_t)} \mathbb{E}_t \left[ \left| \left( f(x_t + \lambda G_t u_t) - f(x_t - \lambda G_t u_t) - 2\lambda \nabla f(x_t)^\top G_t u_t \right) \right| \right]$$

$$\leq \frac{\sqrt{k}}{2\lambda \mathbb{P}(Q_t)} \mathbb{E}_t \left[ \left| \left( f(x_t + \lambda G_t u_t) - f(x_t) - \lambda \nabla f(x_t)^\top G_t u_t \right) \right| \right]$$

$$+ \frac{\sqrt{k}}{2\lambda \mathbb{P}(Q_t)} \mathbb{E}_t \left[ \left| \left( f(x_t) - f(x_t - \lambda G_t u_t) - \lambda \nabla f(x_t)^\top G_t u_t \right) \right| \right]$$

$$\leq \frac{L \lambda k^{\frac{3}{2}}}{2\mathbb{P}(Q_t)}$$

where the last inequality is due to L-smoothness. Therefore,

$$T_1 \leq M \left( \sqrt{2(\sigma_2^2/b' + \gamma^2)} + \frac{L\lambda k^{\frac{3}{2}}}{2\mathbb{P}(Q_t)} \right). \tag{26}$$

For $T_2$, note that

$$\Delta_t^2 = \frac{(f(x_t + \lambda G_t u_t; B_t) - f(x_t - \lambda G_t u_t; B_t) - 2\lambda u_t^\top G_t^\top \nabla f(x_t; B_t) + 2\lambda u_t^\top G_t^\top \nabla f(x_t; B_t))^2}{4\lambda^2}$$

$$\overset{(a)}{\leq} \frac{(f(x_t + \lambda G_t u_t; B_t) - f(x_t - \lambda G_t u_t; B_t) - 2\lambda u_t^\top G_t^\top \nabla f(x_t; B_t))^2 + (2\lambda u_t^\top G_t^\top \nabla f(x_t; B_t))^2}{2\lambda^2}$$

$$\overset{(b)}{\leq} \frac{(f(x_t + \lambda G_t u_t; B_t) - f(x_t; B_t) - \lambda u_t^\top G_t^\top \nabla f(x_t; B_t))^2}{\lambda^2}$$

$$+ \frac{(f(x_t; B_t) - f(x_t - \lambda G_t u_t; B_t) - \lambda u_t^\top \nabla f(x_t; B_t))^2}{\lambda^2} + 2(u_t^\top G_t^\top \nabla f(x_t; B_t))^2$$

$$\overset{(c)}{\leq} \frac{L^2 \lambda^2 k^2}{2} + 2(u_t^\top G_t^\top \nabla f(x_t; B_t))^2,$$

where $(a)$ and $(b)$ are implied by $(a + b)^2 \leq 2(a^2 + b^2)$ and $(c)$ uses the facts $|f(x + \lambda Gu) - f(x) - \lambda u^\top G^\top \nabla f(x)| \leq L\lambda^2 k/2$ and $|f(x) - f(x - \lambda Gu) - \lambda u^\top G^\top \nabla f(x)| \leq L\lambda^2 k/2$ due to $L$-smoothness. Therefore, applying Lemma B.2 (2) gives us

$$\mathbb{E}_t[\Delta_t^2] \leq \frac{L^2 \lambda^2 k^2}{2} + 2\mathbb{E}_{B_t, B_t'} \mathbb{E}_{u_t}[(u_t^\top G_t^\top \nabla f(x_t; B_t))^2]$$

$$= \frac{L^2 \lambda^2 k^2}{2} + 2\mathbb{E}_{B_t, B_t'}[\left\| G_t^\top \nabla f(x_t; B_t) \right\|^2]$$

$$= \frac{L^2 \lambda^2 k^2}{2} + 2\mathbb{E}_{B_t, B_t'}[\nabla f(x_t; B_t)^\top G_t G_t^\top \nabla f(x_t; B_t)]$$

$$= \frac{L^2 \lambda^2 k^2}{2} + 2\mathbb{E}_{B_t, B_t'}[\nabla f(x_t; B_t)^\top \text{Proj}_G(\nabla f(x_t; B_t))]$$

$$\leq \frac{L^2 \lambda^2 k^2}{2} + 2\mathbb{E}_{B_t}[\|\nabla f(x_t; B_t)\|^2]$$

$$\leq \frac{L^2 \lambda^2 k^2}{2} + 2 \left( \|\nabla f(x_t)\|^2 + \frac{\sigma_1^2}{b} \right).$$

By the law of total expectation, we have

$$\mathbb{E}_t \left[ \Delta_t^2 | Q_t \right] \leq \frac{\mathbb{E}_t[\Delta_t^2]}{\mathbb{P}(Q_t)} \leq \frac{L^2 \lambda^2 k^2}{2\mathbb{P}(Q_t)} + \frac{2}{\mathbb{P}(Q_t)} \left( \|\nabla f(x_t)\|^2 + \frac{\sigma_1^2}{b} \right). \tag{27}$$

If we have $C_0^2 \geq 8M^2 \log(4\sqrt{2\pi}b)$, then $\mathbb{P}(\bar{Q}_t) \leq 2\sqrt{2\pi}b \exp\left(-\frac{C_0^2}{8M^2}\right) \leq \frac{1}{2}$ and thus $\mathbb{P}(Q_t) = 1 - \mathbb{P}(\bar{Q}_t) \geq \frac{1}{2}$. Therefore, applying $T_1$ (26) and $T_2$ (27) to Eq. (24) yields

$$\mathbb{E}_t[f(x_{t+1})|Q_t] \leq f(x_t) - \frac{\eta_t}{2\mathbb{P}(Q_t)}\|\nabla f(x_t)\|^2 + \eta_t T_1 + \frac{L\eta_t^2 k}{2}T_2 + \frac{L\eta_t^2\sigma^2 C^2 k}{2b^2\mathbb{P}(Q_t)}$$

$$\leq f(x_t) - \frac{\eta_t}{2\mathbb{P}(Q_t)}\|\nabla f(x_t)\|^2 + \frac{\eta_t M}{\mathbb{P}(Q_t)}\left(\sqrt{2(\sigma_2^2/b' + \gamma^2)} + \frac{L\lambda k^{\frac{3}{2}}}{2}\right)$$

$$+ \frac{L\eta_t^2 k}{\mathbb{P}(Q_t)}\left[\frac{L^2\lambda^2 k^2}{4} + \|\nabla f(x_t)\|^2 + \frac{\sigma_1^2}{b}\right] + \frac{L\eta_t^2\sigma^2 C^2 k}{2b^2\mathbb{P}(Q_t)}. \tag{28}$$

Additionally, by the assumption that $|f(x_t)| \leq S$ for all $t$, we have

$$\mathbb{E}_t[f(x_t) - f(x_{t+1})|Q_t]\mathbb{P}(Q_t)$$
$$= \mathbb{E}_t[f(x_t) - f(x_{t+1})|Q_t \cap Q]\mathbb{P}(Q_t \cap Q) + \mathbb{E}_t[f(x_t) - f(x_{t+1})|Q_t \cap \bar{Q}]\mathbb{P}(Q_t \cap \bar{Q})$$
$$= \mathbb{E}_t[f(x_t) - f(x_{t+1})|Q]\mathbb{P}(Q) + \mathbb{E}_t[f(x_t) - f(x_{t+1})|Q_t \cap \bar{Q}]\mathbb{P}(Q_t \cap \bar{Q})$$
$$\leq \mathbb{E}_t[f(x_t) - f(x_{t+1})|Q]\mathbb{P}(Q) + 2S\mathbb{P}(\bar{Q})$$

and thus

$$\left(\frac{\eta_t}{2} - L\eta_t^2 k\right)\|\nabla f(x_t)\|^2 \leq \mathbb{E}_t[f(x_t) - f(x_{t+1})|Q]\mathbb{P}(Q) + \eta_t M\left(\sqrt{2\left(\frac{\sigma_2^2}{b'} + \gamma^2\right)} + \frac{L\lambda k^{\frac{3}{2}}}{2}\right)$$

$$+ \frac{L^3\eta_t^2\lambda^2 k^3}{4} + \frac{L\eta_t^2 k\sigma_1^2}{b} + \frac{L\eta_t^2\sigma^2 C^2 k}{2b^2} + 2S\mathbb{P}(\bar{Q}).$$

We choose $\eta_t = \frac{1}{4Lk}$ so that $\eta_t - L\eta_t^2 k = \frac{\eta_t}{4}$. Denote $\mathbb{E}_{<t} := \mathbb{E}_{u_{<t}, z_{<t}, B_{<t}, B'_{<t}}$ where $u_{<t}$ is the set $\{u_0, \ldots, u_{t-1}\}$ and similarly for $z_{<t}$, $B_{<t}$, and $B'_{<t}$. By privacy analysis in Section 3, we take $\sigma = c_2 b\sqrt{T\log(1/\delta)}/(n\varepsilon)$ and then there exist constants $c_1$ and $c_2$ such that PAZO-M is $(\varepsilon, \delta)$-differentially private for any $\varepsilon < c_1 b^2 T/n^2, \delta > 0$. We apply $\eta_t, \sigma$, and Eq. (23), sum up from $t = 0$ to $T - 1$, telescope terms, and divide both sides by $T$ to obtain

$$\frac{1}{T}\sum_{t=0}^{T-1}\mathbb{E}[\|\nabla f(x_t)\|^2] \leq \frac{2c_2\sqrt{2Lk\log(1/\delta)}}{n\varepsilon}[f(x_0) - f(x_*) + C^2] + 4M\sqrt{2\left(\frac{\sigma_2^2}{b'} + \gamma^2\right)} + \frac{\sigma_1^2}{b}$$

$$+ 2L\lambda k^{\frac{3}{2}}M + \frac{L^2\lambda^2 k^2}{4} + \frac{512SLkbn\varepsilon}{c_2}\sqrt{\frac{\pi Lk}{\log(1/\delta)}}\exp\left(-\frac{C_0^2}{8M^2}\right)$$

by choosing $T = \frac{4n\varepsilon}{c_2}\sqrt{\frac{2Lk}{\log(1/\delta)}}$. If we choose $\lambda \leq \frac{2(\sqrt{2}-1)C_0}{Lk}$ and $C^2 = 2C_0^2$, then $C \geq C_0 + \frac{L\lambda k}{2}$ is satisfied. Then we choose $C^2 = 2C_0^2 = 16M^2\log\left(\frac{128\sqrt{2\pi}SLkbn^2\varepsilon^2}{c_2^2\log(1/\delta)}\right)$ and have

$$\frac{1}{T}\sum_{t=0}^{T-1}\mathbb{E}[\|\nabla f(x_t)\|^2] \leq \frac{2c_2\sqrt{2Lk\log(1/\delta)}}{n\varepsilon}\left[f(x_0) - f(x_*) + 16M^2\log\left(\frac{128\sqrt{2\pi}SLkbn^2\varepsilon^2}{c_2^2\log(1/\delta)}\right) + 1\right]$$

$$+ 4M\sqrt{2\left(\frac{\sigma_2^2}{b'} + \gamma^2\right)} + \frac{\sigma_1^2}{b} + 2L\lambda k^{\frac{3}{2}}M + \frac{L^2\lambda^2 k^2}{4}$$

$$\leq \frac{2c_2\sqrt{2Lk\log(1/\delta)}}{n\varepsilon}\left[f(x_0) - f(x_*) + 16M^2\log\left(\frac{128\sqrt{2\pi}SLkbn^2\varepsilon^2}{c_2^2\log(1/\delta)}\right) + 1\right]$$

$$+ 4M\sqrt{2\left(\frac{\sigma_2^2}{b'} + \gamma^2\right)} + \frac{\sigma_1^2}{b} + 2MC_0 + \frac{C_0^2}{4k}$$

by choosing $\lambda \leq \frac{2(\sqrt{2}-1)C_0}{Lk^{\frac{3}{2}}} \leq \frac{C_0}{Lk^{\frac{3}{2}}}$. This indicates that the error depends on $k, \sigma_1, \sigma_2$, and $\gamma$ by

$$O(\sqrt{k}\log k) + O\left(\sqrt{\frac{\sigma_2^2}{b'} + \gamma^2} + \frac{\sigma_1^2}{b}\right)$$

with parameters

$$\eta_t = \frac{1}{4Lk}, \quad T = \frac{4n\varepsilon}{c_2}\sqrt{\frac{2Lk}{\log(1/\delta)}}, \quad \lambda \le \frac{2(\sqrt{2}-1)C_0}{Lk^{\frac{3}{2}}}, \quad \text{and}$$

$$C^2 = 2C_0^2 = 16M^2 \log\left(\frac{128\sqrt{2\pi}SLkbn^2\varepsilon^2}{c_2^2\log(1/\delta)}\right).$$

Therefore, we have $d$-independent error rate $O(\sqrt{k}\log k)$, which improves DPZero's $O(\sqrt{d}\log d)$ due to $k$ being a small constant $\ll d$ in practice. We additionally have the error term $O(\gamma^2 + \sigma_2^2/b')$ from the biased public gradients and $O(\sigma_1^2/b)$ from the stochastic private gradients. □

## C   Experiment Details

### C.1   Datasets

The four pairs of datasets and models closely follow the experiments in the existing DP literature. We provide the details of public data generation as follows.

**CIFAR-10.** We follow previous work [41] that uses 4% of the training samples as public data and warm-start on the public data by training on it for a small number of epochs. Additionally, we create class imbalances among the 10 classes for public data. We treat this imbalance as a mild distribution shift from the private data. To avoid information leakage from the batchnorm layer, we start from a randomly initialized NFResNet18 [43].

**Tiny-ImageNet.** We follow Kurakin et al. [44], which first pre-trains a ResNet18 on Places365 [45] and then fine-tunes the model on Tiny-ImageNet with differential privacy. We randomly sample 4% of the Tiny-ImageNet training samples as public data, which thus comprises 20 samples per class. We use a small ViT model (10M) [46] with random initialization.

**IMDB.** We follow Li et al. [16], which uses Amazon Polarity [47] samples as out-of-distribution (OOD) public data to guide the private learning on IMDB. We build the vocabulary based on the top 10K tokens in the IMDB training set and construct the Amazon Polarity public dataset with a size 4% of the IMDB training size, which gives us 2,000 public samples.

**MNLI.** We follow the few-shot setting in the past work [6, 8] and sample 512 MNLI training examples per class. We adopt the same prompt template and start from a pre-trained RoBERTa-base model. We randomly sample 100 training examples per class from SNLI [48] as the OOD public data.

### C.2   Experiment Results

We present detailed results on four datasets in Table 2−5. We report the performance under multiple privacy budgets ($\varepsilon, \delta = 1/\#$train samples) and the non-private performance, which corresponds to accuracies of SGD and MeZO. All results are obtained under the same random seed 0. Entries with '−' indicate failure to converge. The best accuracies are in bold and the second places are underlined.

**Implementation details.**    For each first-order methods with public data, we vectorize the per-sample gradient computation and privatization using `vmap`. For the method with open-sourced code (GEP [40]), we adopt their provided implementation and privacy accounting.

The experiment on MNLI utilizes the codebase from Malladi et al. [6], including their dataset processing and prompt tuning workflow. Following MeZO and DPZero, we sample the zeroth-order direction $u_t$ from Gaussian distribution $\mathcal{N}(0, I_d)$ in the experiments, since prior work verifies that it produces very similar performance [8] to sampling from $\sqrt{d}\mathbb{S}^{d-1}$. Similar to the first-order methods, we apply `vmap` for speedup by vectoring the $q$ forward calls. However, given that PAZO needs smaller $q$'s than the vanilla zeroth-order methods, we do not need to employ this memory-inefficient implementation in most settings.

**PAZO-P vs. PAZO-P′.**    Table 2−5 shows the performance of PAZO-P with orthonormalized public gradients (row 'PAZO-P') and with normalized public gradients (row 'PAZO-P′'). PAZO-P and PAZO-P′ have similar performance, with the deviation being $0.1\% \sim 2.5\%$.

Table 2: Training NFResNet18 on CIFAR-10 from scratch.

| Type | Method | $\varepsilon = 0.1$ | $\varepsilon = 0.5$ | $\varepsilon = 1$ | $\varepsilon = 2$ | $\varepsilon = 3$ | Non-private |
|------|--------|------|------|------|------|------|------|
| FO | DP-SGD | 46.7 | 49.7 | 50.8 | 54.2 | 54.5 | 86.3 |
| | DPMD | 64.3 | 66.6 | 67.8 | 68.5 | 69.8 | |
| FO+PUB | DOPE-SGD | 64.8 | 69.3 | 70.9 | **73.0** | **72.9** | |
| | GEP | – | 49.9 | 50.7 | 52.9 | 53.8 | |
| ZO | DPZero | 47.0 | 48.1 | 48.2 | 48.2 | 48.1 | 49.0 |
| | PAZO-M | **70.9** | **71.3** | **71.3** | 71.2 | 70.5 | |
| ZO+PUB | PAZO-P | 69.5 | 69.6 | 69.0 | 68.7 | 68.1 | |
| (ours) | PAZO-P′ | 69.6 | 69.2 | 69.2 | 68.9 | 68.0 | |
| | PAZO-S | 70.3 | 70.3 | 70.2 | 69.8 | 69.7 | |

Table 3: Fine-tuning Places365 pre-trained ViT-small on Tiny-ImageNet.

| Type | Method | $\varepsilon = 0.1$ | $\varepsilon = 0.5$ | $\varepsilon = 1$ | $\varepsilon = 2$ | Non-private |
|------|--------|------|------|------|------|------|
| FO | DP-SGD | 24.8 | 29.2 | 31.4 | 38.0 | 52.9 |
| | DPMD | 30.5 | 31.4 | **34.2** | **35.5** | |
| FO+PUB | DOPE-SGD | 30.7 | **31.8** | 32.5 | 34.4 | |
| | GEP | – | 30.9 | 30.5 | 31.4 | |
| ZO | DPZero | 25.1 | 27.6 | 27.5 | 27.9 | 28.6 |
| | PAZO-M | 30.8 | 30.8 | 30.7 | 30.8 | |
| ZO+PUB | PAZO-P | **30.9** | 31.0 | 31.0 | 31.2 | |
| (ours) | PAZO-P′ | 30.7 | 30.8 | 30.8 | 30.9 | |
| | PAZO-S | 30.6 | 30.6 | 30.6 | 30.7 | |

Table 4: Training LSTM on IMDB from scratch.

| Type | Method | $\varepsilon = 0.1$ | $\varepsilon = 0.5$ | $\varepsilon = 1$ | $\varepsilon = 2$ | $\varepsilon = 3$ | Non-private |
|------|--------|------|------|------|------|------|------|
| FO | DP-SGD | 50.0 | 66.4 | 69.9 | 73.5 | 75.5 | 89.5 |
| | DPMD | 71.0 | 72.1 | 73.4 | 76.6 | 76.6 | |
| FO+PUB | DOPE-SGD | 70.2 | 73.2 | **75.0** | 75.9 | 77.9 | |
| | GEP | 60.0 | 71.0 | 74.0 | **77.2** | **78.6** | |
| ZO | DPZero | 59.0 | 62.4 | 62.6 | 63.2 | 63.8 | 63.8 |
| | PAZO-M | 73.4 | 73.2 | 74.5 | 73.2 | 73.6 | |
| ZO+PUB | PAZO-P | 71.0 | 73.7 | 73.2 | 73.0 | 72.7 | |
| (ours) | PAZO-P′ | 69.4 | 69.8 | 70.7 | 70.0 | 70.5 | |
| | PAZO-S | **74.6** | **74.2** | 73.8 | 73.9 | 74.2 | |

Table 5: Prompt-tuning RoBERTa-base on MNLI.

| Type | Method | $\varepsilon = 0.1$ | $\varepsilon = 0.5$ | $\varepsilon = 1$ | $\varepsilon = 2$ | $\varepsilon = 3$ | Non-private |
|------|--------|------|------|------|------|------|------|
| FO | DP-SGD | 52.6 | 59.3 | 63.5 | 68.4 | 72.0 | 78.9 |
| | DPMD | 56.5 | 67.0 | 68.1 | **71.5** | **72.8** | |
| FO+PUB | DOPE-SGD | 59.7 | 67.2 | 68.0 | 70.1 | 72.5 | |
| | GEP | – | – | – | – | – | |
| ZO | DPZero | – | 55.2 | 58.2 | 60.4 | 62.6 | 68.4 |
| | PAZO-M | 67.1 | 67.3 | 67.8 | 67.7 | 67.5 | |
| ZO+PUB | PAZO-P | 63.5 | 68.3 | **69.8** | 69.7 | 70.3 | |
| (ours) | PAZO-P′ | 61.0 | 68.1 | 68.8 | 69.0 | 69.4 | |
| | PAZO-S | **68.2** | **68.6** | 68.9 | 68.6 | 69.0 | |

**Performance of public only.** We demonstrate that the improvements of using public data are not due to overfitting to public data. We train on public data alone using SGD with batch size, learning rate, and weight-decay tuned, and the optimal hyperparameter for each setting gives us accuracies equal to 66.1% for CIFAR-10, 27.1% for Tiny-ImageNet, 68.4% for IMDB, and 60.8% for MNLI. We denote these results as 'public only' and pick the best first-order with public data (FO+PUB) and zeroth-order with public data (PAZO) algorithm for each dataset. In Table 6, we present the performance gain when private data is included (i.e., 'FO+PUB/PAZO' minus 'public only' across $\varepsilon = 0.1, 0.5, 1, 2, 3$). Note that 'public only' accuracies come with severe overfitting due to the small number of public samples, while the DP accuracies are not overfit.

Table 6: Performance of training with public data only and the improvements from using private data via first-order (FO+PUB) and zeroth-order (PAZO) methods. We observe that (1) FO enjoys up to 12.0% performance gain and ZO enjoys up to 8.2% when private data is included; (2) ZO consistently enjoys performance gain when private data is included, while FO does not, since private first-order gradients can be too noisy under tight privacy.

|                   | CIFAR-10        | Tiny-ImageNet  | IMDB           | MNLI            |
| ----------------- | --------------- | -------------- | -------------- | --------------- |
| Public only       | 66.1            | 27.1           | 68.4           | 60.8            |
| FO+PUB improvement | $-1.3 \sim 6.8$ | $3.6 \sim 8.4$ | $2.6 \sim 10.2$ | $-1.1 \sim 12.0$ |
| PAZO improvement  | $4.4 \sim 5.2$  | $3.8 \sim 4.1$ | $5.3 \sim 6.2$ | $7.4 \sim 9.5$  |

**Performance under various $\gamma$.** We demonstrate that PAZO performs better when public data is closer to the private data in two settings: pre-training on CIFAR-10 and fine-tuning with prompts on MNLI. To create public data of different extents of distribution shifts (different $\gamma$'s), we mix ID public data and OOD public data with different proportions. For CIFAR-10, we use non-overlapped training samples with small class imbalance as ID public data and those with big class imbalance as OOD public data. The slight class imbalance has class-size ratios $[1 : \ldots : 0.85]$ and big class imbalance has class-size ratios $[1.0 : 0.9 : 0.8 : \ldots : 0.2 : 0.1]$. For MNLI, we use non-overlapped MNLI training samples as ID public data and SNLI training samples as OOD public data. We present the performance and $\gamma$ of PAZO under these scenarios in Table 7. We observe that (1) the range of $\gamma$ is method-dependent and (2) for any fixed PAZO variant, the accuracy increases as the data become more similar (smaller $\gamma$'s).

Table 7: Performance under different public data with $\gamma$ under privacy $\varepsilon = 1.0$. We observe that though the range of $\gamma$-similarity depends on specific methods, the values are consistently bounded and small. For a fixed PAZO variant, as the public data becomes more in-distribution, performance improves and $\gamma$ decreases (i.e., gradients for public and private data become more similar).

| Private Data | Public Data      | PAZO-M               | PAZO-P               | PAZO-S               |
| ------------ | ---------------- | -------------------- | -------------------- | -------------------- |
| CIFAR-10     | Slight imbalance | 71.3 ($\gamma = 4.3$) | 69.0 ($\gamma = 3.4$) | 70.2 ($\gamma = 1.6$) |
|              | Half-half        | 69.9 ($\gamma = 4.8$) | 67.3 ($\gamma = 4.1$) | 67.3 ($\gamma = 1.8$) |
|              | Big imbalance    | 66.7 ($\gamma = 6.2$) | 63.8 ($\gamma = 4.8$) | 63.0 ($\gamma = 2.1$) |
| MNLI         | MNLI only        | 75.7 ($\gamma = 39$)  | 74.1 ($\gamma = 41$)  | 74.0 ($\gamma = 49$)  |
|              | Half-half        | 73.2 ($\gamma = 50$)  | 73.8 ($\gamma = 43$)  | 73.1 ($\gamma = 67$)  |
|              | SNLI only        | 67.8 ($\gamma = 71$)  | 69.8 ($\gamma = 65$)  | 68.9 ($\gamma = 81$)  |

**Runtime efficiency.** Theoretically, we list the number of different types of operations involved in each algorithm in Table 9. Since the first-order methods require per-sample gradient computation and clipping, the number of "gradient backward", the slowest operation, is dependent on the private batch size. This is a discouraging feature since large batch sizes offer better utility/privacy tradeoffs [14, 42], creating an additional tradeoff between utility and efficiency. In contrast, the number of gradient backward steps is either 1 or $k (k \ll b)$ in zeroth-order methods. Together with the fact that the forward calls are more memory-efficient than the backward ones when vectorized, zeroth-order methods are principally more scalable.

Empirically, we evaluate the runtime in each training iteration in all settings (Table 8). We vectorize the three settings other than the IMDB-LSTM experiment due to incompatibility between the model

architecture and `vmap`. Although the MNLI experiments enjoys only $2\times$ of speedup by using PAZO, Malladi et al. [6] show that zeroth-order methods will be significantly faster as the model scales up.

Table 8: Speed of each method on different datasets (in s/iter). It shows that PAZO offers up to $16\times$ runtime speedup per training iteration compared to the baselines. All numbers are averaged over 20 iterations. Note that we report the speed of each method under optimal $(k, b', q)$. DPZero is occasionally slower than PAZO because we try $q = \{1, 5\}$ for each method and observe that DPZero needs $q = 5$ while PAZO can take $q = 1$ to achieve competitive accuracies.

|  | CIFAR-10 | Tiny-ImageNet | IMDB | MNLI |
|---|---|---|---|---|
| DP-SGD | 0.420 | 0.366 | 0.173 | 1.697 |
| DPMD | 0.462 | 0.404 | 0.183 | 1.761 |
| DOPE-SGD | 0.424 | 0.365 | 0.172 | 2.187 |
| GEP | 0.830 | 0.548 | 0.252 | – |
| DPZero | 0.081 | 0.132 | 0.016 | 1.934 |
| PAZO-M | 0.051 | 0.073 | 0.019 | 0.852 |
| PAZO-P | 0.149 | 0.168 | 0.042 | 1.244 |
| PAZO-S | 0.102 | 0.142 | 0.019 | 1.118 |
| Speedup | $16\times$ | $7\times$ | $15\times$ | $2\times$ |

**Memory efficiency.** Table 9 presents the number of different operations needed per iteration of each method, showing that PAZO-{M,P,S} has memory overhead to store public gradients compared to DPZero. PAZO-M requires one batch of public gradient, so the memory overhead is $O(d)$, where $d$ is the number of model parameters. PAZO-S is also $O(d)$ since we can compute the $k$ public batch gradients sequentially. Though PAZO-P has an $O(kd)$ memory overhead than DPZero, it is still more memory- and computation-efficient than the first-order DP methods since the latter generally requires $O(bd)$ memory to maintain per-sample gradients. Our experimental results are obtained using $k = \{3, 6\}$ while $b = 64$. Such entangled dependence on $b$ and $d$ is also restrictive since larger batch sizes improve performance [14, 49].

Table 9: Number of different operations per iteration of each method.

|  | # Private forward | # Public for+backward | # Private backward |
|---|---|---|---|
| DP-SGD | $b$ | – | $b$ |
| DPMD | $b$ | 1 | $b$ |
| DOPE-SGD | $b$ | 1 | $b$ |
| GEP | $b$ | $b'$ | $b$ |
| DPZero | $2q$ | – | – |
| PAZO-M | $2q$ | 1 | – |
| PAZO-P | $2q$ | $k$ | – |
| PAZO-S | $k+1$ | $k$ | – |

### C.3 Hyperparameter Tuning

This section presents our hyperparameter search grid and the results of our methods under different hyperparameter values.

**Hyperparameter selection.** For all the first-order methods and PAZO, we set the number of epochs to 100. Since the vanilla zeroth-order methods benefit from training for more iterations [8, 6], we try training for 100, 200, and 300 epochs with their corresponding correct noise multiplier $\sigma$ applied. Due to increased noise added when more epochs are allowed, we observe that the epoch number of 200 produces the best performance across settings. We thus train for 200 epochs in all DPZero experiments. The values of the smoothing parameter $\lambda$ are presented in Table 10. We also report the hyperparameter search grid for each method in Table $12-13$, where the batch size $b$ is only tuned for

non-private methods (SGD and MeZO); We fix the private batch size to 64 for all private methods, including zeroth-order and first-order, with and without public data.

Table 10: Values of the smoothing parameter $\lambda$ in each experiment.

|  | CIFAR-10 | Tiny-ImageNet | IMDB | MNLI |
|---|---|---|---|---|
| MeZO | $10^{-2}$ | $10^{-2}$ | $10^{-2}$ | $10^{-3}$ |
| DPZero | $10^{-2}$ | $10^{-2}$ | $10^{-2}$ | $10^{-3}$ |
| PAZO-M | $10^{-2}$ | $10^{-2}$ | $10^{-2}$ | $10^{-3}$ |
| PAZO-P | $10^{-2}$ | $10^{-2}$ | $10^{-1}$ | $10^{-2}$ |

**Sensitivity to $q$.**    Table 11 shows that the performance of the vanilla private zeroth-order method relies on setting $q > 1$, which slows down the training and harms utility due to increased noise added for privatization. In contrast, PAZO is less dependent on increased $q$ due to the assistance from public data. This implies that PAZO has approximately the same workload of hyperparameter tuning as DPZero: Under a reasonable or intuitive choice of the hyperparameters for public data sampling, one only needs to find a good combination of clipping norm $C$ and learning rate $\eta$.

Table 11: Performance vs. $q$ in different settings. In each cell, the first row represents the accuracy under $q = 1$ and the second represents that under $q = 5$. We observe that DPZero benefits from increased $q$ in accuracies by 1.0%, 2.4%, 4.8%, and 7.2% on four datasets. In contrast, PAZO has stable performance under different $q$.

| $\frac{q=1}{q=5}$ | CIFAR-10 | Tiny-ImageNet | IMDB | MNLI |
|---|---|---|---|---|
| DPZero | 47.1 | 25.5 | 59.0 | 55.4 |
|  | 48.1 | 27.9 | 63.8 | 62.6 |
| PAZO-M | 70.1 | 30.8 | 72.9 | 67.5 |
|  | 70.3 | 30.8 | 73.6 | 68.3 |
| PAZO-P | 68.1 | 31.2 | 72.7 | 68.6 |
|  | 68.6 | 31.0 | 72.7 | 70.9 |

**Sensitivity to introduced hyperparameters.**    Apart from Figure 6, we also present the hyperparameter sensitivity study on the other two datasets Tiny-ImageNet and IMDB in Figure 8. The conclusion is the same as in the main text: PAZO is not sensitive to the values of the introduced hyperparameters.

**Influence of $\epsilon$ in PAZO-S.**    Figure 6 and Figure 8 show that the performance of PAZO-S is robust to different $\epsilon$ values. Since having no noisy candidate is equivalent to setting $\epsilon = 0$, we compare the best performance of having a noisy candidate (purple cells) with none (blue cells). The conclusion is consistent: Having $\epsilon \neq 0$ offers the opportunity to improve performance in general, but it does not harm significantly to leave it less tuned.

**Figure 8:** All PAZO methods are robust to different values of their introduced hyperparameters. Each number represents the best accuracy with standard hyperparameters for zeroth-order private methods ($C$ and $\eta$) tuned. Blue cells indicate PAZO-S performance without having a noisy candidate.

**PAZO-M**

| Tiny-ImageNet | $\alpha$=0.25 | $\alpha$=0.5 | $\alpha$=0.75 | | IMDB | $\alpha$=0.25 | $\alpha$=0.5 | $\alpha$=0.75 |
|---|---|---|---|---|---|---|---|---|
| $b'$=8 | 30.7 | 30.6 | 30.8 | | $b'$=8 | 73.0 | 73.6 | 73.2 |
| $b'$=32 | 30.7 | 30.7 | 30.7 | | $b'$=32 | 73.5 | 73.4 | 73.3 |

**PAZO-P**

| Tiny-ImageNet | $k$=3 | $k$=6 | $k$=10 | | IMDB | $k$=3 | $k$=6 | $k$=10 |
|---|---|---|---|---|---|---|---|---|
| $b'$=8 | 30.6 | 30.8 | 30.5 | | $b'$=32 | 69.8 | 70.7 | 70.3 |
| $b'$=16 | 31.1 | 30.8 | 30.9 | | $b'$=64 | 71.1 | 72.5 | 72.5 |
| $b'$=32 | 31.2 | 31.0 | 30.9 | | $b'$=128 | 71.4 | 71.5 | 72.7 |

**PAZO-S**

| Tiny-ImageNet | $b'$=8 | $b'$=16 | $b'$=32 | | IMDB | $b'$=4 | $b'$=8 | $b'$=32 |
|---|---|---|---|---|---|---|---|---|
| $\epsilon$=1e-3 | **30.7** | 30.6 | 30.7 | | $\epsilon$=1e-2 | 72.0 | 72.1 | 71.4 |
| $\epsilon$=1e-4 | 30.6 | 30.6 | 30.7 | | $\epsilon$=1e-3 | **74.2** | 73.2 | 72.8 |
| $\epsilon$=1e-5 | 30.7 | 30.6 | 30.6 | | $\epsilon$=1e-4 | 73.2 | 72.8 | 72.9 |
| $\epsilon$=0 | **30.7** | 30.6 | 30.7 | | $\epsilon$=0 | 73.6 | **74.5** | 71.8 |

**Table 12:** The hyperparameter search grid for CIFAR-10 and Tiny-ImageNet.

| Algorithm | | CIFAR-10 | Tiny-ImageNet |
|---|---|---|---|
| SGD | $\eta$ | {0.01, 0.02, 0.05, 0.1, 0.2, 0.5} | {0.001, 0.005, 0.01, 0.05, 0.1} |
| | $b$ | {8, 32, 64} | {64} |
| DP-SGD | $\eta$ | {0.01, 0.02, 0.05, 0.1, 0.2} | {0.01, 0.02, 0.05, 0.1, 0.2, 0.5, 1.0, 2.0} |
| | $C$ | {0.1, 0.5, 1.0, 2.0} | {0.01, 0.1, 0.5, 1.0, 2.0} |
| DOPE-SGD | $\eta$ | {0.01, 0.02, 0.05, 0.1, 0.2} | {0.001, 0.005, 0.01, 0.02, 0.05, 0.1, 0.2} |
| | $b'$ | {8, 32, 128} | {8, 32, 128} |
| | $C$ | {0.1, 0.5, 1.0, 2.0} | {0.1, 0.5, 1.0, 2.0, 4.0} |
| DPMD | $\eta$ | {0.02, 0.05, 0.1, 0.2, 0.5} | {0.005, 0.01, 0.02, 0.05, 0.1, 0.2} |
| | $b'$ | {8, 32, 128} | {8, 32, 128} |
| | $C$ | {0.1, 0.5, 1.0, 2.0} | {0.01, 0.1, 0.5, 1.0, 2.0} |
| GEP | $\eta$ | {0.005, 0.01, 0.02, 0.05, 0.1, 0.2, 0.5} | {0.01, 0.02, 0.05, 0.1, 0.2, 0.5} |
| | $b'$ | {8, 32, 128} | {8, 32, 128} |
| | $C_1$ | {0.1, 0.5, 1.0, 2.0} | {0.1, 0.5, 1.0, 1.5, 2.0} |
| MeZO | $\eta$ | {0.001, 0.002, 0.005, 0.01, 0.02, 0.05, 0.1} | {1e-4, 2e-4, 5e-4, 1e-3, 2e-3} |
| | $b$ | {64} | {64} |
| DPZero | $\eta$ | {0.01, 0.02, 0.05, 0.1, 0.2, 0.5, 1.0} | {1e-4, 2e-4, 5e-4, 1e-3, 2e-3} |
| | $C$ | {1.0} | {1.0} |
| PAZO-M | $\eta$ | {0.1, 0.2, 0.5} | {1e-5, 2e-5, 5e-5, 1e-4, 2e-4, 5e-4} |
| | $b'$ | {8, 32} | {8, 32} |
| | $\alpha$ | {0.25, 0.5, 0.75} | {0.25, 0.5, 0.75} |
| | $C$ | {1.0} | {1.0} |
| PAZO-P | $\eta$ | {0.2, 0.5, 1.0, 1.5, 2.0} | {0.2, 0.5, 1.0, 1.5, 2.0} |
| | $b'$ | {8, 16, 32} | {8, 16, 32} |
| | $k$ | {3, 6, 10} | {3, 6, 10} |
| | $C$ | {0.5, 1.0, 2.0} | {0.5, 1.0, 2.0} |
| PAZO-S | $\eta$ | {0.01, 0.02, 0.05, 0.1, 0.2} | {0.001, 0.005, 0.01, 0.02, 0.05, 0.1, 0.2} |
| | $b'$ | {8, 16, 32} | {8, 32, 128} |
| | $k$ | {3} | {3} |
| | $\epsilon$ | {0.01, 0.001} | {0.001, 0.0001} |
| | $C$ | {0.5, 1.0, 2.0, 4.0} | {0.5, 1.0, 2.0, 4.0} |

Table 13: The hyperparameter search grid for IMDB and MNLI.

| Algorithm | | IMDB | MNLI |
|---|---|---|---|
| SGD | $\eta$ | {0.1, 0.2, 0.5, 1.0, 1.5} | {1e-6, 1e-5, 1e-4, 1e-3, 5e-3, 1e-2} |
| | $b$ | {64} | {8, 32, 64} |
| DP-SGD | $\eta$ | {0.01, 0.02, 0.05, 0.1, 0.2, 0.1} | {2e-6, 5e-6, 1e-5, 2e-5, 5e-5, 1e-4} |
| | $C$ | {0.1, 0.5, 1.0, 2.0, 4.0} | {10, 20, 50, 100, 150, 200, 250} |
| DOPE-SGD | $\eta$ | {0.005, 0.01, 0.02, 0.05, 0.1} | {5e-6, 1e-5, 2e-5, 5e-5, 1e-4} |
| | $b'$ | {8, 32, 128} | {8. 32} |
| | $C$ | {0.1, 0.5, 1.0, 2.0, 4.0} | {10, 20, 50, 100, 150, 200, 250} |
| DPMD | $\eta$ | {0.005, 0.01, 0.02, 0.05, 0.1} | {2e-6, 5e-6, 1e-5, 2e-5, 5e-5, 1e-4, 2e-4} |
| | $b'$ | {8, 32, 128} | {8, 32} |
| | $C$ | {0.1, 0.5, 1.0, 2.0, 4.0} | {10, 20, 50, 100, 150, 200, 250} |
| GEP | $\eta$ | {0.01, 0.02, 0.05, 0.1} | {2e-6, 5e-6, 1e-5, 2e-5, 5e-5, 1e-4, 2e-4} |
| | $b'$ | {8, 32} | {8, 32} |
| | $C_1$ | {0.1, 0.5, 1.0, 2.0} | {10, 20, 50, 100, 150, 200, 250} |
| MeZO | $\eta$ | {0.002, 0.005, 0.01, 0.02, 0.05, 0.1} | {1e-7, 1e-6, 2e-6, 5e-6, 1e-5, 1e-4} |
| | $b$ | {64} | {64} |
| DPZero | $\eta$ | {0.002, 0.005, 0.01, 0.02, 0.05, 0.1} | {1e-6, 2e-6, 5e-6, 1e-5, 2e-5, 5e-5} |
| | $C$ | {0.1, 0.5, 1.0, 2.0} | {10, 20, 50, 100, 150, 200, 250} |
| PAZO-M | $\eta$ | {1.0, 1.5, 2.0, 2.5, 3.0, 3.5, 4.0} | {1e-4, 2e-4, 5e-4, 1e-3, 2e-3} |
| | $b'$ | {8, 32} | {8, 32} |
| | $\alpha$ | {0.25, 0.5, 0.75} | {0.25, 0.5, 0.75} |
| | $C$ | {0.1, 0.5, 1.0, 2.0, 4.0} | {10, 20, 50, 100, 150, 200, 250} |
| PAZO-P | $\eta$ | {0.1, 0.2, 0.5, 1.0, 1.4, 2.0} | {5e-5, 1e-4, 2e-4, 5e-4, 1e-3, 2e-3} |
| | $b'$ | {32, 64, 128} | {8, 16, 32} |
| | $k$ | {3, 6, 10} | {3, 6, 10} |
| | $C$ | {0.5, 1.0, 2.0, 4.0} | {10, 20, 50, 100, 150, 200, 250} |
| PAZO-S | $\eta$ | {0.1, 0.2, 0.5, 1.0, 1.5, 2.0, 2.5, 3.0, 3.5, 4.0} | {1e-4, 2e-4, 5e-4, 1e-3, 2e-3, 5e-3} |
| | $b'$ | {8, 32, 128} | {8, 32} |
| | $k$ | {3} | {3} |
| | $\epsilon$ | {0.01, 0.001} | {0.01, 0.001} |
| | $C$ | {0.1, 0.5, 1.0} | {0.1, 0.5, 1.0} |

