# OpenReview forum: "Private Zeroth-Order Optimization with Public Data"
_NeurIPS.cc/2025/Conference — NeurIPS 2025 poster_

### Official Review · Reviewer_jMhd · 2025-06-26

**Clarity:** 2
**Significance:** 3
**Originality:** 3
**Rating:** 4
**Confidence:** 3

**Summary:**

This paper aims to accelerate zeroth-order differentially private optimization while improving gradient approximation by leveraging public data. The authors propose three PAZO variants and validate them both theoretically and empirically.

**Questions:**

- Why is the noise added to the scalar directional derivative $\{f(x+\lambda u)- f(x -\lambda u)\} / 2\lambda$, instead of $\text{clip}(\{f(x+\lambda u)- f(x -\lambda u)\} / 2\lambda) \cdot u$ as DPZero[7]? DPZero added noise to the vector after clipping, resulting in the noise added in all $d$ dimensions. However, in this paper, scalar noise is added and then multiplied by the random vector. This makes the direction of $u$ preserved. What are the privacy implications of preserving the direction of $u$? Even though the vector $u$ is random, it is used as an input in the directional derivative $f(x+\lambda u)- f(x -\lambda u) / 2\lambda$, and thus may potentially leak private information. How can this be justified from a privacy perspective?
- Figure 4 only reports PAZO-S. However, the convergence theorems show different upper bounds for PAZO-M, P, and S. Based on these, we would expect different convergence behaviors. Can the authors provide convergence plots for each variant?

**Ethical Concerns:**

["NO or VERY MINOR ethics concerns only"]

**Final Justification:**

In the rebuttal, my major concern about the privacy guarantee was addressed. I raised my score from 3 to 4

**Limitations:**

More interpretation of experimental results is needed regarding robustness and the privacy-utility tradeoff.

**Paper Formatting Concerns:**

The paper is well formatted.

**Quality:**

3

**Strengths And Weaknesses:**

## Strengths
- Proposed three versions of the optimization method to utilize public data in zeroth-order settings.
- The proposed methods often achieve performance comparable to first-order DP methods that use public data, while significantly improve the speed of private learning.
- The privacy-utility trade-off is mitigated, which implies that PAZO can be useful where $\epsilon$ is small.

## Weaknesses
- PAZO improves performance by using public data under the γ-similarity assumption. However, this raises the question: isn't the improvement mostly due to the public data alone? That is, isn’t it possible that the private data contributed very little during training? In fact (1) The performance does not vary much with the privacy budget. (2) The results are not very sensitive to hyperparameters.
In first-order methods, private gradients are clipped, whereas public gradients are not, meaning public gradients may dominate the update. As a result, private gradients might have a negligible impact. To address this, the authors should conduct an experiment using only public data (i.e., no private data, hence no privacy concerns), and explicitly report how much performance improves when private data is included.

- In the vision domain, the authors assume in-distribution public data, while in the language domain, they assume out-of-distribution (OOD) public data. This seems to be done to satisfy the γ-similarity assumption. Perhaps in language tasks, even OOD data still lies in a similar feature space. If that is the case, the paper should discuss domain-specific similarity more clearly or unify the assumption across domains. If in-distribution public data is assumed, this should be clearly stated at the beginning of the paper.


### Minor weaknesses
- Notation should be changed. The notation of the model is $x$, while that of the data is $\xi$. Normally, the use of the notation $x$ refers to data and $\theta$ or $w$ to parameters.
- Comparisons between methods using public data and those that do not (e.g., Figures 2 and 5) are unclear. As done in Tables 1–4, figures should visually differentiate methods that use public data (e.g., using distinct colors or styles).

---

> ### Author Rebuttal · Authors · 2025-07-31
>
> We appreciate the reviewer's time and detailed feedback. We present our responses and address the concerns as follows.
>
> **[Performance of public only]** We thank the reviewer for this question. Indeed, we view the robustness of PAZO across various privacy budgets and hyperparameters as one advantage. Next, we demonstrate that the improvements are not due to overfitting to public data. As suggested, we train on public data alone using SGD with BS, LR, and WD tuned. We denote the optimal accuracy in each dataset as ‘public only’ and pick the best first-order with public data (FO+PUB) and zeroth-order with public data (PAZO) algorithm for each dataset. We present the performance gain when private data is included (i.e., ‘FO+PUB/PAZO’ minus ‘public only’ across $\varepsilon=\\{0.1,0.5,1,2,3\\}$).
>
> |    | CIFAR10 | Tiny-ImgNet | IMDB | MNLI |
> | --- | :---: | :---: | :---: | :--: |
> | Public only  | 66.1 | 27.1 | 68.4 | 60.8 |
> | FO+PUB improvement | -1.3 ~ 6.8   | 3.4 ~ 8.4 | 1.8 ~ 9.5 | -4.3 ~ 12.0 |
> | PAZO improvement | 4.0 ~ 4.5  | 3.8 ~ 4.0 | 5.4 ~ 6.2 | 7.4 ~ 8.2 |
>
> Note that these ‘public only’ accuracies come with severe overfitting due to the small number of public samples. In contrast, the DP accuracies come without overfitting. We observe that 1) FO enjoys up to 12.0 performance gain and ZO enjoys up to 8.2 when public data is included; 2) ZO consistently enjoys performance gain when private data is included, while FO does not, since private gradient can be too noisy when privacy is tight.
>
> **[OOD vision public data]** We thank the reviewer for suggesting this. In our submission, for vision tasks, we did use public data with slight class imbalance (with class_size ratio $[1:...:0.85]$) and treated this as a mild distribution shift from the private data (Section 5.1 line 228). Our $\gamma$ similarity assumption is an upper bound of the distribution gap, which is unified across different domains. To verify that vision OOD data also provide gradients with bounded distance from private gradient (i.e., reasonable $\gamma$-similarity), we create a public dataset with big class imbalance (with class_size ratio $[1.0:0.9:0.8:...:0.2:0.1]$). (Running SGD on this imbalanced public data alone produces a test accuracy of 50.1 on balanced test data.) We report the test accuracies and $\gamma$-similarity of PAZO below ($\varepsilon=1$). We observe that while the range of $\gamma$-similarity depends on specific methods, the values are consistently small. For a fixed PAZO variant, as the public data becomes more in-distribution, performance improves and $\gamma$ decreases (i.e., gradients for public and private data become more similar).
>
> | Pub data \ Method | PAZO-M | PAZO-P| PAZO-S |
> | --- | :---: | :---: | :---: |
> | Slight imbalance  | 70.2 ($\gamma=4.5$) | 69.2 ($\gamma=3.3$) | 70.2 ($\gamma=1.6$) |
> | Half-half | 69.8 ($\gamma=4.9$)   | 67.6 ($\gamma=3.8$) | 67.3 ($\gamma=1.8$) |
> | Big imbalance | 65.5 ($\gamma=6.4$) | 64.1 ($\gamma=4.7$) | 63.0 ($\gamma=2.1$) |
>
> To demonstrate improvements of PAZO under more out-of-distribution public data, we report the performance of PAZO compared with the best first-order methods with public data (Best FO+PUB, a.k.a. DOPE-SGD) under big class imbalance. It shows that 1) OOD vision public data improves the performance of all the methods; 2) ZO+PUB also outperforms FO+PUB under tight privacy budgets ($\varepsilon\leq 1$), even when the public data is OOD.
>
> | Type | Method | $\varepsilon=0.1$ | $\varepsilon=0.5$ | $\varepsilon=1.0$ | $\varepsilon=2.0$ | $\varepsilon=3.0$ |
> | --- | --- | :---: | :---: | :---: | :---: | :---: |
> | FO | DP-SGD  | 46.7 | 49.7 | 50.8 | 54.2 | 54.5|
> | Best FO+PUB | DOPE-SGD|  54.8 | 60.1 | 64.2| **67.6** | **68.3** |
> |ZO | DPZero | 47.0 | 48.1 | 48.2 | 48.2 | 48.1 |
> | (ours) | PAZO-M | **65.3** | **65.6** | **65.5** | 65.9 | 66.3 |
> | (ours) | PAZO-P | 64.3 | 64.0 | 64.1 | 64.3 | 65.1 |
> | (ours) | PAZO-S | 63.2| 62.9 | 63.0 | 64.1 | 63.8 |
>
> **[Adding noise $z$]** Thank you for this question. We’d like to clarify that PAZO-M adds a scalar noise $z$ to the clipped scalar $\text{clip}((f(x+\lambda u)−f(x−\lambda u))/(2\lambda))$ (Algorithm 1 line 8 in our submission), following similar designs as in DPZero ([1] Algorithm 2 line 3). We note that another option of adding noise to the vector $\text{clip}(f(x+\lambda u)−f(x−\lambda u)/(2\lambda) )\cdot u$ ([1] Algorithm 1 line 3) appeared in DPZero as one baseline, not their actual proposed algorithm ([1] Algorithm 2); and they show that adding noise to the scalar derivative is preferred ([1] Section 4 and Appendix E “Privacy guarantee”). The reviewer is correct that $(f(x+\lambda u)−f(x−\lambda u))/(2\lambda) $ is data-dependent, which is exactly why we privatize this term (Line 8) to satisfy differential privacy.
>
> [1] DPZero: Dimension-Independent and Differentially Private Zeroth-Order Optimization. ICML 2024.
>
> **[Convergence speed]** Since figures cannot be uploaded here, we present the accuracy that DPZero converges to and compare the evaluation step index at which each ZO/ZO+PUB method achieves that accuracy ($\varepsilon=1$). We observe that PAZO variants have slightly different convergence behaviors, but they are all consistently faster than ZO without public data.
>
> |    | CIFAR10 (acc 48.2) | Tiny-ImgNet (acc 27.5) | IMDB (acc 62.6) | MNLI (acc 58.2) |
> | --- | :---: | :---: | :---: | :--: |
> | DPZero  | 393 | 272 | 379 | 496 |
> | PAZO-M | 10  | 16 | 80 | 14 |
> | PAZO-P | 7 | 14 | 118 | 16 |
> | PAZO-S | 11  | 9 | 10 | 15 |
>
> We will present the convergence plot of all variants in the revised paper.
>
> **[Notation and figure format]** We appreciate these suggestions and will adopt the new notation in our revised version. We will also update the figures by using more distinct colors and styles to differentiate methods with and without public data.

---

> > ### Comment · Reviewer_jMhd · 2025-08-04
> >
> > Thank you for your rebuttal.
> > In the rebuttal, my major concerns about a public-only setting and the noise addition to a scalar value have been addressed.
> > I'll raise my score.

---

> > > ### Author Response · Authors · 2025-08-04
> > > **Response**
> > >
> > > Thank you again for your valuable time and review! We are glad your major concerns have been addressed.

---

### Official Review · Reviewer_5rKD · 2025-06-30

**Clarity:** 3
**Significance:** 3
**Originality:** 3
**Rating:** 5
**Confidence:** 2

**Summary:**

The authors present a novel zero-order optimization algorithm that is guided by some first-order information (i.e., gradients) of publicly available data. They introduce three different ways of integrating such information in the zero-order optimization process and show that they can achieve faster convergence while still keeping the privacy advantages coming with zero-order optimization.

**Questions:**

- wouldn't a repeated querying of the same function lead to a much higher privacy budget needed to ensure DP?

**Ethical Concerns:**

["NO or VERY MINOR ethics concerns only"]

**Final Justification:**

As I have written in my initial review, I think the authors' work is a significant contribution towards efficient zero-order optimization in privacy restricted regimes. The use of publicly available data is a promising direction to advance optimization algorithms under privacy constraints. In their rebuttal, the reviewers have addressed all my points raised in the initial review. Although I didn't read the other reviews in full detail, it seems the authors have addressed major concnerns in the rebuttal.

**Limitations:**

The work should explicilty discuss current limitations (at least briefly).

**Quality:**

3

**Strengths And Weaknesses:**

**Strengths**

- The idea is interesting and reasonable in distributed environments where data privacy is important
- the paper is written clearly
- a convergence analysis is provided
- empirical results look promising

**Weaknesses**

- in Algo. 1 it is unclear how the batch of public data is obtained. Since it should match with the private batch as good as possible, the authors should describe how they ensure that both batches exhibit similar data characteristics (if they exhibit conflicting characteristics, divergence might even happen)
- in the unnumbered between 141-142 it is not clear to me why with the presented equation, the coefficients $u$ are learned. Please clarify this.
- in the Equation before line 142, why is it $\mathbb{E}_u[\nabla f(x)^T GuGu]$? Isn't it only $Gu$? If so, the subsequent equations are also wrong. A brief derivation would be helpful here.
- The authors claim an equivalence between their projection of the gradient and the expectation of gradient approximations when sampling from a ball induced by $G$ (see lines 143-144). Please provide proof of this equivalence.

Since I am not familiar with the field of DP and zero-order optimization, my review should be used with care for the final decision.

---

> ### Author Rebuttal · Authors · 2025-07-31
>
> We appreciate the reviewer’s valuable feedback and positive assessment of our work. We present our responses below.
>
> **[Choice of public batch]** We thank the reviewer for carefully noticing this. In practice, public data can be obtained via users who opt out of private training or other side information/statistics of the sensitive data [1], which would guarantee certain similarities.
>
> Since there is no prior work that leverages public information in private zeroth-order optimization, we present all three PAZO methods, each with different properties. PAZO-M (Algorithm 1) indeed simply aggregates the private ZO gradient with a randomly sampled public batch gradient, which is fast and memory-efficient. Empirically, we show that if we pick reasonable public data with mild distribution shifts, PAZO-M performs well. Theoretically, we show that if the public gradient is not close to the private gradient, one should reduce $\alpha$ for better convergence.
>
> Due to the need for selecting good public gradients, we have proposed PAZO-P that allows for larger weights on good public gradients and smaller weights on less related public gradients. We also propose PAZO-S, which directly selects the public gradient that incurs the lowest loss. They may offer better performance in some scenarios (Section 5) while having more computational cost.
>
> [1] Private Adaptive Optimization with Side Information. ICML 2022.
>
> **[Learning coefficients]** We thank the reviewer for the careful reading and pointing out the vagueness. We are learning the coefficient in a zeroth-order fashion, i.e., relying on function evaluations. Specifically, when we sample $q$ $u$’s, each sampled $u$ is an aggregation proposal (for how to linearly combine the columns of $G$), and the loss difference is the feedback that indicates how good this proposal is. The update direction in PAZO-P, in essence, is a weighted average of the columns of $G$, with the weights (i.e., coefficients) depending on the loss differences. Since we take the feedback from $f$ to weigh and aggregate the public gradients, we interpret the coefficients to be learned. We will clarify this in the next version.
>
> **[Why GuGu]** A more detailed derivation is as follows: The term $\Delta := [f(x + \lambda Gu) - f(x - \lambda Gu)]/(2\lambda)$ is a central difference approximation to the directional derivative of $f$ at $x$ in the direction $Gu$. As $λ \rightarrow 0$, this converges to $\nabla f(x)^{\top}Gu$. Therefore,
> $$\mathbb{E}[\Delta Gu]=\mathbb{E}[\nabla f(x)^{\top} GuGu]=\mathbb{E}[Gu\nabla f(x)^{\top}Gu]=\mathbb{E}[Guu^{\top}G^{\top}\nabla f(x)]$$ as claimed.
>
> **[Proof of equivalence]** We present the derivation as follows. Note that the expectation of the two-point zeroth-order gradient estimator is the same as that of one-point due to the symmetry of the sampling distribution:
>
> $$ \mathbb{E}_u \left[\frac{f(x+\lambda u) - f(x-\lambda u)}{2\lambda}u \right] = \mathbb{E}_u \left[\frac{f(x+\lambda u) }{2\lambda}u \right] + \mathbb{E}_u \left[\frac{ f(x + \lambda (-u))}{2\lambda}(-u) \right] = \mathbb{E}_u  \left[\frac{f(x+\lambda u) }{\lambda}u \right].$$
>
> Therefore, the goal is to show $\nabla_x \mathbb{E}\_{v\sim\text{Unif}(G(\sqrt{k}B^k))}[f(x+\lambda v)] = \mathbb{E}\_{u\sim\text{Unif}(\sqrt{k}S^{k-1})}[\frac{f(x+\lambda u)}{\lambda}u]$. According to the divergence theorem in higher dimensions, we define the body $\Omega =G(\sqrt{k}B^k) \subset \mathbb{R}^d$ and its boundary $\partial\Omega=G(\sqrt{k}S^{k-1})$, and then we have
>
> $$ \int_{\Omega} \nabla_v f(x+\lambda v) dv = \int_{\partial\Omega} f(x+\lambda u) \frac{u}{\Vert u\Vert} du = \frac{1}{\sqrt{k}} \int_{\partial\Omega} f(x+\lambda u) u du. $$
>
> Note that $\nabla_v f(x+\lambda v) = \lambda \nabla_x f(x+\lambda v)$ and $\text{Area}(r S^{d-1}) = \frac{d}{r} \cdot \text{Vol} (r B^d)$. Since an orthonormal $G$ is area- and volume-preserving, we have $\text{Area}(\partial\Omega) = \sqrt{k} \cdot \text{Vol} (\Omega)$. Combining the above, we obtain
>
> $$ \nabla_x \mathbb{E}\_{v\sim\text{Unif}(\Omega)}[f(x+\lambda v)] = \frac{1}{\lambda \text{Vol}(\Omega)} \int_{\Omega} \nabla_vf(x+\lambda v) dv = \frac{1}{\lambda \sqrt{k}\text{Vol}(\Omega)}  \int_{\partial\Omega} f(x+\lambda u) u du  = \frac{\text{Area}(\partial\Omega)}{\lambda\sqrt{k} \text{Vol}(\Omega)} \mathbb{E}\_{u\sim \text{Unif}(\partial\Omega)}[f(x+\lambda u) u] = \frac{1}{\lambda} \mathbb{E}\_{u\sim \text{Unif}(\partial\Omega)}[f(x+\lambda u) u]$$
> as desired.
>
> **[Privacy implications of $q$]** We thank the reviewer for emphasizing that and would like to clarify that we have already taken that into account by multiplying the variance of noise $z$ in each step by $q$ (Algorithm 1-3 in our submission). We present the privacy guarantees in Section 3.4.
>
> **[Limitations]** In Section 6, we acknowledge the limitation that the current convergence bound can be further sharpened by considering other similarity metrics, e.g., angle-based metrics. Further, a broader and larger-scale benchmark of public and private dataset pairs will enhance the significance of our empirical results.

---

> > ### Comment · Reviewer_5rKD · 2025-08-04
> >
> > I thank the authors for the rebuttal, my points have been addressed accordingly.

---

> > > ### Author Response · Authors · 2025-08-05
> > > **Response**
> > >
> > > Many thanks for your valuable time and review! We are glad that your concerns have been addressed.

---

### Official Review · Reviewer_97FF · 2025-07-02

**Clarity:** 3
**Significance:** 3
**Originality:** 3
**Rating:** 4
**Confidence:** 5

**Summary:**

The paper considers to use public data to assist private zeroth-order optimization. Existing works have shown that zeroth-order methods can be applied to differentially private training or fine-tuning. However, there is still a gap in terms of performance compared with first-order methods. This work proposes a suite of public data assisted zeroth-order optimizers (PAZO), including PAZO-{M,P,S}. PAZO-M simply uses a convex combination of private zeroth-order estimations and public gradient as the descent step. PAZO-P narrows the search space of zeroth-order optimization into a subspace constructed using public gradient. PAZO-S uses private data to select the best public gradient. Convergence analysis shows that all three methods match the worst-case upper bound of existing results. Empirical experiments show that all three methods match or even outperform exising public data assisted first-order methods, while offering speedup in time.

**Questions:**

Major questions are already asked in weakness. Other minor:

1. Typos: should it be "rely on" instead of "reply on" in line 21 and line 132? I guess it should be "combined with these prior methods" in line 88.

2. The paper uses random direction uniformly on the sphere when introducing the algorithm. Random search directions from Gaussian are also extensively used in previous work. Which noise is used in the experiments? How do these two compare in both the theory and experiments?

3. Isn't it costly to sample $k$ mini-batches in PAZO-P and PAZO-S? What is the value of $k$ and batch size $b$ used for each method when comparing time efficiency in Figure 5?

**Ethical Concerns:**

["NO or VERY MINOR ethics concerns only"]

**Final Justification:**

My major concern is on the memory efficiency of the proposed method, as public gradient is required but the main motivation to use zeroth-order methods is to save memory by avoiding gradient computation. The author replied in the rebuttal that the saving in memory is on the private data compared with public data assisted DP first-order methods. This makes some sense but not fully resolved this limitation. However, there might be some settings where the proposed methods provide clear advantage. I think the paper proposes an interesting and new observations that public data can greatly improve the performance of private zeroth-order methods. The reasons to accept outweight reasons to reject, so I keep my score 4 for acceptance.

**Limitations:**

Limitations are already discussed in the weakness, including dimension dependence in theoretical results and memory consumption.

**Paper Formatting Concerns:**

Looks good to me.

**Quality:**

3

**Strengths And Weaknesses:**

The paper is clearly written, and the studied topic is interesting and important for the community. The claims are supported by evidence, and the results look good to me. My concerns are following.

1. Why is $\gamma$ similarity used? A more natural measure to me is the cosine similarity. In all convergence results, there is dependence on $\sigma$. I understand the meaning of $\sigma_1$ and $\sigma_2$, but what is $\sigma$? What is the dependence on $q$ and the dimension $d$ in all theoretical results? The dependence on the dimension $d$ is crucial in zeroth-order methods and should be highlighted and well discussed.

2. In several places of the paper, the authors say existing zeroth-order methods only consider prompt tuning for language models, like [6,7,8,10]. I think this is not a correct statement. Prompt tuning is one efficient way of doing language model fine-tuning, where only prompt embeddings are updated and the base model is frozen. This is not what [6,7,8,10] do. In these related works, all trainable model parameters are updated and thus considered to be full parameter fine-tuning. The authors should correct all these wrong statements.

3. One of the major reasons that previous works propose to use zeroth-order methods is to save memory by avoid gradient computations. In PAZO, public gradients are anyway required, and thus this benefit of saving memory is lost. Is there a comparison in terms of GPU memory for all methods, including vanilla DPZero and DPSGD, PAZO-{M,P,S}, and public-data assisted first-order methods?

4. The authors claim in the contribution that improved convergence can be guaranteed using a proper choice of parameters. It is not clear in the main text how the rates can be improved. Is this improvement only in terms of constants or also in terms of key parameters like $T$ and $d$?

---

> ### Author Rebuttal · Authors · 2025-07-31
>
> We are grateful to the reviewer for the careful reading and valuable comments. We present our responses as follows.
>
> **[$\gamma$- vs. cosine similarity]** We thank the reviewer for this suggestion. Cosine similarity measures angular alignment, while Euclidean distance captures both scale and angle mismatch. Since $\Vert a−b\Vert\leq\gamma$ ensures $b$ stays within a $\gamma$-radius ball around $a$, it more directly controls the absolute error of gradient updates in convergence analysis, which is also adopted by existing literature [2,3]. Empirically, we observe that our Euclidean similarities consistently correlate positively with performance (Table 1-2 in rebuttal). However, we acknowledge that cosine similarity could complement our analysis: If we assume $\Vert a\Vert\leq M_1$ and $\Vert b\Vert\leq M_2$, then the cosine similarity lower bound $\text{cos⁡}(a,b)\geq \varphi$ implies $\Vert a-b\Vert\leq\gamma=\max(M_1, M_2, \sqrt{M_1^2+M_2^2-2M_1M_2\varphi})$, which can be plugged into our current analysis. We leave more geometric-aware similarity measures and their relations to the convergence of our methods and those of others for future work.
>
> **[Clarification on $\sigma$]** We clarify that $\sigma$ is the variance of the noise we add to guarantee differential privacy (Section 3.4 in our submission, line 171).
>
> **[Dependence on $q$, $T$, $d$, and improved convergence]** We achieve either constant improvements or better dependence on $d$, depending on the specific PAZO variant. Specifically, if we choose $T=\frac{(1-\alpha)^2n(r+2)\varepsilon}{4\sqrt{r\log(e+\varepsilon/\delta)}}$, PAZO-M achieves $O((1−\alpha)^2\sqrt{d} \log((1−\alpha)d))$, which has a better dependence on $d$ than DPZero if the errors from public data $O(\alpha^2/(1-\alpha)^2(\gamma^2+\sigma_2^2/b'))$ do not dominate. If we choose $T=\frac{n(r+2)\varepsilon}{4\sqrt{r\log(e+\varepsilon/\delta)}}$, PAZO-P has $d$-independent error rate $O(\sqrt{d} \log k)$ with an additional error $O(\gamma^2 + \sigma_2^2/b')$ where $k \ll d$ is the number of bases of $G$. PAZO-S has an error rate that is $d$-independent since it selects the best public gradient as the update direction.
>
> When $q>1$, the error of zeroth-order optimization reduces by $O(1/q)$ while the variance of the noise $z$ increases by $q$-times. The resultant error term will be of the form $Aq+B/q$ for some $A$ and $B$, so the optimal $q$ can be chosen as $q=\sqrt{B/A}$.
>
> **[Wording on prompt tuning]** We thank the reviewer for pointing it out. We would like to clarify that we use “prompt tuning” to refer to the task of full-parameter fine-tuning with prompts. Our argument is that (which also aligns with [5]) ZO may not work in standard full-model fine-tuning without prompts where 1) the input sequence is not modified with task-specific natural language templates or 2) a task-specific classification head is added on top of the final \<s\> embedding. We will replace “prompt tuning” with “full fine-tuning with prompts” to eliminate the vagueness.
>
> **[Memory cost of PAZO]** While we agree that PAZO requires memory for public gradients, it eliminates the memory/computation/implementation bottlenecks when we consider private settings. PAZO is more efficient (and sometimes more accurate) than differentially-private first-order methods, and is more accurate than vanilla DP zeroth-order methods without public data.
>
> In Table 6 of our submission, we present the number of different operations needed per iteration of each method. Specifically, PAZO-M requires one batch of public gradient, so the memory overhead is $O(d)$, where $d$ is the number of model parameters. PAZO-S is also $O(d)$ since we can compute the $k$ public batch gradients sequentially. Though PAZO-P has an $O(kd)$ memory overhead than DPZero, it is still more memory- and computation-efficient than the first-order DP methods because the latter generally requires $O(bd)$ memory to maintain per-sample gradients, where $b$ is the private batch-size. Our experimental results are obtained using $k=\\{3,6\\}$ while $b=64$. Such dependence on $b$ is also restrictive since larger batch-sizes improve performance [6,7].
>
> Therefore, compared to first-order DP with public data, PAZO 1) enjoys time- and memory-efficiency and 2) performs better under tight privacy; compared to DPZero, PAZO converges faster to better accuracies, at the cost of computing public batch gradients.
>
> **[Perturbation sampling]** We would like to clarify that we use Gaussian perturbations in our experiments (line 570-572 in our submission) as most of the other ZO work does [1,4,5]. Our rationales are two-fold. (1) Theoretically, uniform sampling from the sphere and sampling from Gaussian are closely related: To sample from $\text{Uni}(\sqrt{d}S^{d-1})$, it is equivalent to sample a vector from $\mathcal{N}(0, I_d)$ and then scales it to have norm $\sqrt{d}$. Since the norm of $v\sim\mathcal{N}(0, I_d)$ concentrates on $\sqrt{d}$ with variance $O(1/d)$, we can expect that sampling from these two distributions does not give us much different vectors when $d\gg 1$ with high probability.
>
> (2) Empirically, prior work [1] has shown that Gaussian and sphere perturbations produce very similar results. The following table shows that this also holds for PAZO, where we present the accuracy of sampling from Gaussian minus that of sampling from the sphere ($\varepsilon=1$).
>
> | Method | CIFAR10 |Tiny-ImgNet |IMDB |MNLI |
> | -- | :--: | :--: | :--: | :--: |
> | PAZO-M | 0.9 | 0.2 | 0.5 | -0.3 |
> | PAZO-P | 0.7 | -0.1 | -0.7 | 1.1|
> | PAZO-S | 0.4 | 0.5 | -0.4 | 0.9 |
>
> **[Figure 5 hyperparameter]** We would like to clarify that each run plotted in Figure 5 is the one with the best accuracy of the corresponding method after $\\{k, b^{\prime}\\}$ are tuned. DPZero is occasionally slower than PAZO because we try $q=\\{1,5\\}$ for each method and use $q=1$ if $q=5$ does not bring significant improvements. As presented in Table 7 in our submission, we observe that the optimal $q$ for DPZero is $5$ while public information allows PAZO to use $q=1$ to achieve competitive accuracies.
>
> Though the runs of PEZO-P and PAZO-S in Figure 5 require computing $k$ batch gradients, $k=\\{3,6\\}$ are small. $q=1$ for PAZO and $q=5$ for DPZero. The values of $b^{\prime}$ of each method are presented below.
>
> | Method | CIFAR10 |Tiny-ImgNet |IMDB |MNLI |
> | -- | :--: |:--: |:--: |:--: |
> | DPMD | 8 | 8 | 8 | 8 |
> | DOPE-SGD | 8 | 32 | 8 | 8 |
> | GEP | 32 | 32 | 32 | --|
> | PAZO-M | 8 | 8 | 4 | 32 |
> | PAZO-P | 32 | 16| 64| 32|
> | PAZO-S | 8 | 8 | 8 | 8|
>
> [1] DPZero: Dimension-Independent and Differentially Private Zeroth-Order Optimization. ICML 2024. \
> [2] Train Faster, Generalize Better: Stability of Stochastic Gradient Descent. ICML 2016. \
> [3] Optimization Methods for Large-Scale Machine Learning. SIAM Review 2018. \
> [4] Revisiting Zeroth-Order Optimization for Memory-Efficient LLM Fine-Tuning: A Benchmark. ICML 2024. \
> [5] Fine-Tuning Language Models with Just Forward Passes. NeurIPS 2023. \
> [6] Learning Differentially Private Recurrent Language Models. ICLR 2018. \
> [7] Unlocking High-Accuracy Differentially Private Image Classification through Scale. ICML 2022.

---

> > ### Comment · Reviewer_97FF · 2025-08-04
> >
> > Many thanks for the detailed response! I do not have further questions. I will keep my initial score to recommend acceptance of the paper.
> >
> > There seem to be some typos in the rebuttal. Isn't the rate of PAZO $\sqrt{r}$ instead of $\sqrt{d}$, as the authors said dimension independent. I also checked the response to another reviewer, where the rate is $\sqrt{r}$ under additional assumptions.

---

> > > ### Author Response · Authors · 2025-08-04
> > > **Response**
> > >
> > > Many thanks for your time and feedback. Yes, that is a typo (thanks for catching it!), and we will correct to $O((1-\alpha)^2\sqrt{r}\log((1-\alpha)d))$ for PAZO-M and $O(\sqrt{r}\log k)$ for PAZO-P in our revised paper.

---

### Official Review · Reviewer_bY4V · 2025-07-03

**Clarity:** 3
**Significance:** 2
**Originality:** 3
**Rating:** 4
**Confidence:** 4

**Summary:**

In this paper, the authors proposed combining public first-order gradients with private zeroth-order gradients for differentially private training.  The paper proposes three ways of mixing the gradients: 1. naive weighted averaging; 2. projecting ZO direction to FO gradient subspace; 3.   using zeroth-order evaluation to determine the optimal public gradient direction. The paper provides both convergence and privacy analyses of the proposed algorithms, and presents numerical results on CV and NLP datasets.

**Questions:**

Please address the weakness.

I believe the theoretical analysis is limited and may contain errors. If my concerns about the theoretical analysis can be addressed, I would love to increase my score.

**Ethical Concerns:**

["NO or VERY MINOR ethics concerns only"]

**Final Justification:**

Based on the rebuttal discussion and clarification. I believe the issues in the theoretical analysis are minor and can be fixed during final revision. The numercial experiment details have also been clarified. Therefore, I would raise my score.

**Quality:**

3

**Strengths And Weaknesses:**

Strength:
1. The paper provides novel ways of mixing zeroth and first-order gradients for differentially private optimization.
2. The paper introduces $\gamma$-similarity to measure the dissimilarity between the public and private data distributions.
3. The paper provides a theoretical analysis for the privacy and convergence guarantee of all three proposed algorithms under reasonable assumptions.
4. Numerical results show that the proposed algorithms outperform the baseline algorithm DPZero and are less sensitive to $\epsilon$.

Weakness:
1. For PAZO-P and PAZO-S, how does the memory consumption compare with DP-Zero or PAZO-M? A key benefit of using ZOO is its memory-saving. If the FO gradient matrix G uses too much memory, there might be no point in using the ZO gradient.
2. For PAZO-M, clipping the function value itself does not make sense to me. In the case that $f$ is very large (or very small), clipping it to C does not provide any information. It might be more reasonable to clip the function difference.
3. Is argmin the best operator for PAZO-S? The author fails to investigate other smoothed versions of argmin to avoid directly dropping the other gradient information.

4. For theoretical analysis, the convergence results rely on the clipping being inactive. While in DPZero, the effect of clipping is taken into consideration with the concentration bound. Therefore, the analysis requires further refinement to avoid assuming the clipping is inactive and the bounded gradient Assumption 1.
5. The paper does not provide the final privacy-utility trade-off, i.e., the optimal choice of steps $T$ for convergence.
6. In the proof in B.4 for PAZO-S, in lines 550 and 551, it is unclear to me why the expected value of $g_t - g'(x_t;B'_{*})$ is bounded by $\sigma^2_2$. Since after argmin, $g'$ is no longer unbiased.

7. I didn't find the choice of $b$ in Table 8. Also, for DP-SGD, why is $b$ not an integer?
8. Since the paper introduced $\gamma$, it is good to have a set of experiments demonstrating how the algorithm performs with different $\gamma$'s for one dataset.

---

> ### Author Rebuttal · Authors · 2025-07-31
>
> We appreciate the reviewer’s time and detailed feedback. We address the concerns and clarification questions as follows.
>
> **[Memory cost of PAZO]** While we agree that PAZO requires memory for public gradients, it eliminates the memory/computation/implementation bottleneck of maintaining per-sample gradients in private settings. PAZO is more efficient (and sometimes more accurate) than differentially-private first-order methods, and is more accurate than vanilla DP zeroth-order methods without public data.
>
> In Table 6 of our submission, we present the number of different operations needed per iteration of each method. Specifically, PAZO-M requires one batch of public gradient, so the memory overhead is $O(d)$, where $d$ is the number of model parameters. PAZO-S is also $O(d)$ since we can compute the $k$ public batch gradients sequentially. Though PAZO-P has an $O(kd)$ memory overhead than DPZero, it is still more memory- and computation-efficient than the first-order DP methods because the latter generally require $O(bd)$ memory to maintain per-sample gradients, where $b$ is the private batch-size. Our experimental results are obtained using $k=\\{3,6\\}$ while $b=64$. Such dependence on $b$ is also restrictive since larger batch-sizes improve performance [3,4].
>
> Therefore, compared to first-order DP with public data, PAZO 1) enjoys time- and memory-efficiency and 2) performs better under tight privacy; compared to DPZero, PAZO converges faster to better accuracies, at the cost of computing public batch gradients.
>
> **[Clipping difference]** We’d like to clarify that for PAZO-M, we indeed clip the function difference (Algorithm 1 line 8). Please let us know if the reviewer still has concerns.
>
> **[argmin in PAZO-S]** We motivate the design of argmin as in Eq. (3) where in that particular setting, argmin is optimal. We note that a smoothed version of argmin (linearly combining the public gradients) is explored in PAZO-P, where the final update direction is a linear combination of the public gradients. We acknowledge that this is a promising direction, and we leave it as future work to explore other ways to determine the coefficients.
>
> **[Refined bounds with active clipping and final privacy/utility tradeoffs]** We note that we did not incorporate effective clipping in the original submission as we were using weaker assumptions. In particular, DPZero uses our Assumption 1 ([1] page 37, the equation above Eq. 21), in addition to a stronger per-sample Lipschitz assumption (Assumption 3.1 in [1]). Our submission assumed inactive clipping also because 1) it is a common assumption in DP literature [2] and 2) our novelty is not proposing/analyzing clipping methods but exploring leveraging public information in zeroth-order optimization both theoretically and empirically.
>
> However, for a clearer comparison with DPZero, we thank the reviewer for suggesting this and provide the refined error bounds considering active clipping and privacy/utility tradeoffs under exactly the same assumptions as DPZero:
>
> **PAZO-M.** For any $\varepsilon>0$ and $\delta\in(0,1)$, PAZO-M is $(\varepsilon,\delta)$-DP. Under Assumption 3.5 in [1], suppose $\max_{0\leq t\leq T} |F_S(x_t)|\leq B$, the output $x_{\tau}$ satisfies
> $$  \mathbb{E}[\Vert\nabla F_S(x_{\tau})\Vert^2] \leq \frac{\sqrt{r\log(e + \varepsilon/\delta)}}{n\varepsilon} \left[ 64((F_S(x_0) - F_S^*)l + (1-\alpha)^2\tilde{L}^2)+ 2L^2 \right] + 4(\alpha(\gamma^2+\sigma_2^2/b^{\prime}) ) + \frac{\alpha^2(\gamma^2+\sigma_2^2/(2b'))}{4(1-\alpha)^2(r+2)}$$
>
> where we define $\tilde{L}^2 = L^2\log\left(\frac{2\sqrt{2\pi}n^3\varepsilon^2(r+2)((1-\alpha)d+\alpha+8(1-\alpha)^2lB(r+2)/L^2)}{r\log(e+\varepsilon/\delta)}\right)$
>
> and choose the parameters to be
> $$\eta = \frac{1}{4(1-\alpha)^2l(r+2)}, \quad T=\frac{(1-\alpha)^2n(r+2)\varepsilon}{4\sqrt{r\log(e+\varepsilon/\delta)}}, \quad \lambda\leq \frac{L}{ld^{\frac{3}{2}}}\left(\frac{\sqrt{r\log(e+\varepsilon/\delta)}}{n\varepsilon}\right)^{\frac{1}{2}}, \quad C=4\tilde{L}.$$
>
> This shows that our dependence on $d$ and $r$ becomes $O((1-\alpha)^2\sqrt{r}\log((1-\alpha)d))$. Compared to DPZero's $O(\sqrt{r}\log d)$, if $\alpha>0$, our error has a smaller dependence on $d$ but additional errors $O(\alpha^2/(1-\alpha)^2(\gamma^2+\sigma_2^2/b^{\prime}))$ due to using biased public gradients.
>
> **PAZO-P.** For any $\varepsilon>0$ and $\delta\in(0,1)$, PAZO-P is $(\varepsilon,\delta)$-DP. Under the same assumptions as stated above, the output $x_{\tau}$ satisfies
> $$ \mathbb{E}[\Vert\nabla F_S(x_{\tau})\Vert^2] \leq \frac{\sqrt{r\log(e + \varepsilon/\delta)}}{n\varepsilon} \left[ 64((F_S(x_0) - F_S^*)l + \tilde{L}^2)+ 3L^2 \right] + 8(\gamma^2+\sigma_2^2/b^{\prime}) $$
> where we define $\tilde{L}^2 = L^2\log\left(\frac{2\sqrt{2\pi}n^3\varepsilon^2(r+2)(k+8lB(r+2)/L^2)}{r\log(e+\varepsilon/\delta)}\right)$ and choose $C=4\tilde{L}$ and $\eta, T$, and $\lambda$ the same as in DPZero Theorem 3 [1]. This shows that we achieve a $d$-independent error bound $O(\sqrt{r}\log k) + O(\gamma^2+\sigma_2^2/b^{\prime}), k\ll d$.
>
> **[Proof in B.4]** We thank the reviewer for pointing out the typo—the upper bound should be $2\sigma_2^2$, and all the remaining analysis still holds (differing by a constant). We provide step-by-step derivation for the claim $\mathbb{E}[\Vert g_t^{\prime}−g^{\prime}(x_t;B_∗^{\prime})\Vert^2] \leq 2\sigma_2^2/b^{\prime}$ as follows:
>
> Recall that $b'$ is the public batch-size and $g_t^{\prime}$ is the full public gradient. For a sampled public batch index $i\in[k+1]$, its gradient is $g(x_t;B_{t,i}^{\prime}) = g_t^{\prime} + \frac{1}{b^{\prime}} \sum_{j=1}^{b^{\prime}} \zeta_{t,j}^{(i)}$ where $\zeta_{t,j}^{(i)}$ is the stochastic gradient noise for the public sample $j$ in the $i$-th batch. We denote the selected best batch as $I$ and thus
>
> $$ \mathbb{E}\_{B_t^{\prime}}[\Vert g_t^{\prime}-g^*(x_t;B_t^{\prime})\Vert^2]  = \mathbb{E}\_{B_t^{\prime}} \left[\Vert\frac{1}{b^{\prime}}\sum_{j=1}^{b^{\prime}}\zeta_{t,j}^{(I)}\Vert^2\right]  = \frac{1}{b^{\prime}} \mathbb{E}\_{B_t^{\prime}} \left[\Vert\zeta_{t}^{(I)}\Vert^2\right] = \frac{1}{b^{\prime}} \left( \Vert\mathbb{E}\_{B_t^{\prime}} \left[\zeta_{t}^{(I)}\right]\Vert^2 + \mathbb{E}\_{B_t^{\prime}} \left[\Vert\zeta_t^{(I)} - \mathbb{E}\_{B_t^{\prime}}[\zeta_{t}^{(I)}]\Vert^2\right] \right).$$
>
> By assumption, $\mathbb{E}\_{B_t^{\prime}} \left[\Vert\zeta_{t}^{(i)}\Vert^2\right] \leq \sigma_2^2$ for any batch $i$. The choice of $I$ is dependent on the private data batch $B_t$. Therefore, by the law of total expectation,
>
> $$\Vert\mathbb{E}\_{B_t^{\prime}} \left[\zeta_{t}^{(I)}\right]\Vert^2 = \Vert\mathbb{E}\_i \left[ \mathbb{E}\_{B_t^{\prime}} [\zeta_{t}^{(I)}|I=i]\right]\Vert^2 \leq \mathbb{E}\_i \left[ \Vert\mathbb{E}\_{B_t^{\prime}} [\zeta_{t}^{(I)}|I=i] \Vert^2 \right] \leq \mathbb{E}\_i [\sigma_2^2] = \sigma_2^2.$$
>
> For the variance term,
> $$ \mathbb{E}\_{B_t^{\prime}} \left[\Vert\zeta_{t}^{(I)} - \mathbb{E}\_{B_t^{\prime}}[\zeta_{t}^{(I)}]\Vert^2\right] \leq \mathbb{E}\_{B_t^{\prime}} \left[\Vert\zeta_{t}^{(I)}\Vert^2\right] = \mathbb{E}\_i \left[ \mathbb{E}\_{B_t^{\prime}} \left[\Vert\zeta_{t}^{(I)}\Vert^2\right] |I=i \right] \leq \sigma_2^2.$$
>
> Substituting the bias and variance bounds, we obtain $\mathbb{E}\_{B_t^{\prime}}[\Vert g_t^{\prime}-g^*(x_t;B_t^{\prime})\Vert^2] \leq 2\sigma_2^2/b^{\prime}.$
>
> **[Choice of b in Table 8]** In hyperparameter tuning, $b$ is tuned only for non-private methods (MeZO and SGD). We fixed the private batch-size to 64 for all private methods, including zeroth-order and first-order, with and without public data. We will make it clear in our appendix. We thank the reviewer for finding the typo in the line for DP-SGD in Table 8—the $b$ should have been $\eta$.
>
> **[Performance under various $\gamma$]** To create public data of different extents of distribution shifts (different $\gamma$), we mix ID public data (MNLI) and OOD public data (SNLI) with different proportions. We present the performance and $\gamma$ of PAZO ($\varepsilon=1.0$) under these scenarios. We observe that 1) the range of $\gamma$ is method-dependent and 2) for any fixed PAZO variant, the accuracy increases as the data become more similar (smaller $\gamma$).
> | Pub data \ Method  | PAZO-M | PAZO-P | PAZO-S |
> | -- | -- | -- | -- |
> | MNLI only | 75.9 ($\gamma=38$) | 73.2 ($\gamma=43$) | 74.0 ($\gamma=49$) |
> | Half-half | 73.5 ($\gamma=49$) | 72.7 ($\gamma=45$) | 73.1 ($\gamma=67$) |
> | SNLI only | 67.7 ($\gamma=72$) | 68.8 ($\gamma=68$) | 68.9 ($\gamma=81$) |
>
> We also present results on CIFAR10 with public data of different extents of distribution shifts, where we observe similar trends. The public data with slight class imbalance has class_size ratios $[1:...:0.85]$. To create public data with more distribution shift, we enforce big class imbalance (with class_size ratios $[1.0:0.9:0.8:...:0.2:0.1]$).
> | Pub data \ Method  | PAZO-M | PAZO-P | PAZO-S |
> | -- | -- | -- | -- |
> | Slight imbalance | 70.2 ($\gamma=4.5$) | 69.2 ($\gamma=3.3$) | 70.2 ($\gamma=1.6$) |
> | Half-half | 69.8 ($\gamma=4.9$) | 67.6 ($\gamma=3.8$) | 67.3 ($\gamma=1.8$) |
> | Big imbalance | 65.5 ($\gamma=6.4$) | 64.1 ($\gamma=4.7$) | 63.0 ($\gamma=2.1$) |
>
> [1] DPZero: Dimension-Independent and Differentially Private Zeroth-Order Optimization. ICML 2024. \
> [2] Do Not Let Privacy Overbill Utility: Gradient Embedding Perturbation for Private Learning. ICLR 2021. \
> [3] Learning Differentially Private Recurrent Language Models. ICLR 2018. \
> [4] Unlocking High-Accuracy Differentially Private Image Classification through Scale. ICML 2022.

---

> > ### Comment · Reviewer_bY4V · 2025-08-04
> >
> > My concerns have been addrerssed by the authors. The correction and update in the theoretical analysis, and clarification on the settings in the experiments are clear. I would raise my score.
> >
> > Still, for private first order gradient computation, it is worth mentioning that exisitng ghost-clipping and bookkeeping methods avoids using O(bd) memory for storing per-sample DP gradients.

---

> > > ### Author Response · Authors · 2025-08-05
> > > **Response**
> > >
> > > Thank you again for your valuable time and review! We are glad to know that the concerns have been addressed.
> > >
> > > We appreciate the suggestion of mentioning ghost-clipping and bookkeeping methods, which could mitigate the memory cost of vanilla DP-SGD. While we briefly mention efficient DP-SGD implementations in the introduction (first paragraph), we will explicitly highlight the benefits of those methods in our revised paper. For instance, we note that ghost-clipping/bookkeeping can improve the layer-wise gradient storage to $\min\\{2bp^2, bd\\}$ where $b$ is private batch-size, $d$ is the number of model parameters of a specific layer, and  $p$ is the feature dimension of this layer, i.e., sequence length for text data [5]. It may introduce additional computations, and the improvements may be limited when $p$ is large, which is usually the case for LM with long context windows [5]. We totally agree that there exist relatively efficient DP-SGD variants, and we will properly cite and discuss them.
> > >
> > > [5] Differentially Private Optimization on Large Model at Small Cost. ICML 2023.

---

### Note · Authors · 2025-08-12

Dear reviewers and area chair(s),

We truly appreciate your valuable time and review of our submission. Our rebuttal addressed reviewers’ clarification questions and concerns, summarized as follows:

**[Memory cost]** Compared to first-order DP with public data, PAZO (1) performs better under tight privacy, and (2) is both time- and memory-efficient. It reduces memory overhead from DP-SGD’s $O(bd)$ to $O(kd)$, $b \gg k$. Additionally, existing memory-efficient DP methods reduce dependence on $d$ (not $b$), making PAZO complementary. Compared to zeroth-order DP, PAZO converges faster to higher accuracies, requiring only a small number of public-batch gradients.

**[Privacy/utility tradeoffs]** Our submission proves convergence under weaker assumptions than DPZero; in the rebuttal, we additionally show convergence under identical assumptions: PAZO-M has constant-factor improvements in $d$-dependence, and PAZO-{P,S} have $d$-independent error rates.

**[Experiment diagnostics]** Improvements of public-data-assisted methods, both first-order (FO) and zeroth-order (ZO), are not due to overfitting to public data. Training on public data alone performs worse, and adding private data yields substantial gains for both.

**[Various OOD public data]** We evaluated PAZO with public data of varying distribution shifts and computed their $\gamma$ values. For any PAZO variant, accuracy increases as public data becomes more similar to private data (smaller $\gamma$), consistent with theory.

In summary, our contributions are:

**Algorithm design.** PAZO-{M,P,S} are the first private zeroth-order optimizers augmented with public data to construct better gradient estimates, making ZO competitive in settings where ZO traditionally underperforms.

**Memory efficiency & convergence guarantees.** PAZO is more memory-efficient than private FO methods with public data, and converges faster than private ZO methods. We present the convergence of each PAZO variant and compare its memory consumption with baselines.

**Empirical validation.** Across image/text domains and pre-training/fine-tuning, we show that PAZO consistently offers superior privacy/utility tradeoffs, outperforming the best public-augmented FO in high-privacy regimes, with up to 16$\times$ speedup over traditional FO.

After the rebuttal, all reviewers expressed that their concerns have been addressed, and we will incorporate new results/discussions in the revision. We thank reviewers and area chair(s) again.

---

### Decision · Program_Chairs · 2025-09-17

**Decision:**

Accept (poster)

**Comment:**

After the discussion phase all reviewers agree that the submission is of high enough quality for publication to NeurIPS. As Area Chair, I agree and recommend acceptance.